# A solution for constraining past marine Polar Amplification

A. Morley [1,2] ✉, E. de la Vega[1], M. Raitzsch[3], J. Bijma [4], U. Ninnemann[5], G. L. Foster[6], T. B. Chalk[7], J. Meiland [8], R. R. Cave[9], J. V. Büscher[9,10] & M. Kucera [8]

Most climate proxies of sea surface temperatures suffer from severe limitations when applied to cold temperatures that characterize Arctic environments. These limitations prevent us from constraining uncertainties for some of the most sensitive climate tipping points that can trigger rapid and dramatic global climate change such as Arctic/Polar Amplification, the disruption of the Atlantic Meridional Overturning Circulation, sea ice loss, and permafrost melting. Here, we present an approach to reconstructing sea surface temperatures globally using paired Mg/Ca · $\delta^{18}O_c$ recorded in tests of the polar to subpolar planktonic foraminifera *Neogloboquadrina pachyderma*. We show that the fidelity of Mg/Ca-based paleoclimate reconstructions is compromised by variations in seawater carbonate chemistry which can be successfully quantified and isolated from paleotemperature reconstructions using a multiproxy approach. By applying the calibration to the last glacial maximum, we show that marine polar amplification has been underestimated by up to $3.0 \pm 1.0$ °C in model-based estimates.

One of the key drivers of Arctic climate and Arctic Amplification is the ocean[1,2] because of its role in transporting heat to high latitudes, which influences not only marine temperatures but also terrestrial climates and sea ice extent[3,4]. As a result, temperature changes in the high-latitude oceans have global implications, impacting planetary albedo, atmosphere-ocean carbon dioxide exchange, atmospheric and ocean circulation, deep-water formation, freshwater budgets, productivity, and biological diversity[5–8]. However, large uncertainties remain about the sensitivity of Arctic climate to natural[9] and anthropogenic forcing[10] owing to the limited observational and historical network in this area[11] and uncertainties linked to the strength of cloud feedback in model simulations[12]. In the absence of a direct proxy for clouds, large-scale surface temperature patterns are useful analogues[13] and critical for constraining equilibrium climate sensitivity in climate models[12]. Efforts to assess the magnitude of past polar surface temperature patterns using climate proxies are thus crucial to further our understanding of how the Arctic will respond to, and influence continued global warming[4,14]. Ice-core records provide accurate and high-resolution surface air temperatures over high-altitude central Greenland for the past 120 thousand years (ka), however, the reconstruction of sea surface temperatures (SSTs) from high-latitude oceans has been challenging. This is because planktonic foraminifera species diversity collapses to one dominant species (*N. pachyderma*) at cold SSTs (< 4°), leading to large uncertainties associated with reconstructed SSTs based on assemblage counts[15,16]. Other proxies are affected by their own set of limitations in high-latitude oceans, for example, dinoflagellate cyst

[1]University of Galway, School of Geography, Archaeology, and Irish Studies, and Ryan Institute, Galway, Ireland. [2]iCRAG – Irish Centre for Research in Applied Geosciences, Belfield, Dublin 4, Ireland. [3]Dettmer Group GmbH & Co. KG., Bremen, Germany. [4]Alfred-Wegener-Institut, Helmholtz-Zentrum für Polar- und Meeresforschung, Bremerhaven, Germany. [5]University of Bergen, Department of Earth Science and Bjerknes Centre for Climate Research, Bergen, Norway. [6]University of Southampton, School of Ocean and Earth Science, National Oceanography Centre Southampton, Southampton, UK. [7]Centre Européen de Recherche et d'enseignement des géosciences de l'environnement (CEREGE), Aix-en-Provence, France. [8]MARUM – Center for Marine Environmental Sciences, University of Bremen, Bremen, Germany. [9]University of Galway, School of Natural Sciences, Galway, Ireland. [10]Ulster University, School of Geography and Environmental Sciences, Coleraine, UK. ✉e-mail: audrey.morley@universityofgalway.ie

assemblages at high latitudes are controlled by multiple variables, and it is often difficult to ascertain which is responsible for the observed assemblage change[17]. In addition, standard geochemical proxies such as oxygen isotopes ($\delta^{18}O$), Mg/Ca-thermometry and Alkenone-based SST reconstructions cannot accurately constrain cold temperatures < 4 °C because secondary controls besides temperature influence all these proxies at low temperatures[18–20].

The Mg/Ca ratio of the $CaCO_3$ tests of foraminifera is an established and powerful tool for the reconstruction of past SSTs[21]. It is based on the principle that the partition of magnesium in calcite is endothermic and, therefore, leads to higher Mg/Ca values at higher temperatures[22]. In fact, the first Mg/Ca-temperature calibration, which ultimately led to the 'acceptance' of this temperature proxy within the scientific community, was based on measurements of *N. pachyderma*[23]. However, this seminal work also excluded samples collected from "cold sites" annually covered by sea ice from which the Mg content appeared disconnected from temperature. This makes it difficult to use the otherwise powerful Mg/Ca proxy in the polar realm. Several attempts have been made since to calibrate the Mg/Ca – thermometer for *N. pachyderma*[20,23–27], however, to date, calibration efforts have not been able to resolve SSTs below 4 °C. The difficulty in solving cold SSTs is exacerbated by the low sensitivity of the exponential Mg/Ca temperature equation and because it appears that secondary controls have a dominant influence on Mg/Ca at low temperatures. These non-thermal controls lead to unrealistically high temperatures at high latitude sites causing reconstructed glacial maxima SSTs at polar and subpolar latitudes to be too warm by 6-7 °C, especially in the Greenland and Iceland Seas[20] relative to modelled SSTs (Supplementary Fig. S5). Importantly, these warm glacial maximum SSTs are at odds with paleoclimate simulations, causing significant data-model mismatch[28]. To resolve this problem, calibration equations for *N. pachyderma* have been modified to ever more insensitive calibration equations[27,29,30], reducing the actual temperature estimates and the corresponding amplitude of glacial-interglacial SST change. Subsequently, these low-SST and low-amplitude records have been interpreted as 'winter' or subsurface (below 100–150 m) temperature signals. The latter interpretation is based on the introduction of a cold bias in the geochemical signature (both Mg/Ca and oxygen isotopes) recorded in the gametogenic crust *N. pachyderma* deposits at the end of its lifecycle that is frequently assumed to be formed in colder and deeper water[31] (see also Methods). However, recent advances in our understanding of *N. pachyderma* ecology challenge the assumption of systematic onto-genic vertical migration to deeper or colder waters associated with reproduction or "crusting"[32,33]. Moreover, culture-grown *Neogloboquadrina incompta* and *N. pachyderma* recovered from sediment traps have lower crust Mg/Ca ratios than ontogenetic calcite formed at the same temperature, suggesting that a lower temperature is not responsible for the low Mg/Ca ratio of neogloboquadrinid crusts[25,34].

The departure from the predicted Mg/Ca-temperature relationship at low temperatures has led to the hypothesis that, in addition to temperature, seawater carbonate ion concentration ($[CO_3^{2-}]$) may exert a strong secondary control on Mg/Ca especially at lower concentrations often associated with cold SSTs[20,24]. Culture experiments where planktonic foraminifera grew under controlled conditions[35–37] and core-top calibration studies, for which modern hydrographic parameters are available, support this hypothesis, especially when $[CO_3^{2-}]$ values are < 200 $\mu$mol.kg$^{-1}$[38,39], which is common at polar and Arctic latitudes. To account for and correct for a possible $[CO_3^{2-}]$ influence on Mg/Ca-thermometry, the species-specific sensitivity must be fully resolved[38,39]. Once the sensitivity is determined, mathematical correction schemes can isolate temperature from $[CO_3^{2-}]$ within the Mg/Ca ratios[39–41]. The resulting tool would allow for accurate SST reconstructions at high latitudes and thereby provide the means to quantify polar amplification on multiple timescales under various boundary conditions. Furthermore, accurate estimates of past SSTs

are essential to constrain the oceanic carbonate system via boron isotope ($\delta^{11}B$) geochemistry[42], currently leaving a gap in our understanding of high-latitude ocean-atmosphere carbon fluxes.

Here, we define the Mg/Ca - $[CO_3^{2-}]$ sensitivity for *N. pachyderma* by measuring Mg/Ca ratios on specimens collected live (as determined by the presence of active cytoplasm) from plankton tows from the Labrador and Nordic Seas that can be paired directly with hydrographic conditions of the calcification environment (Supplementary Fig. S1 and Supplementary Tables 1–3). Plankton tows via multinet sampling allow for an accurate analysis of the hydrographic conditions of the calcification environment when assuming that the plankton tow depth is representative of the mean calcification depth of living specimens (for details on hydrographic data used, see Supplementary Tables 1–3). This approach allows us to circumvent the fundamental limitations and assumptions stemming from the traditional core-top approach to proxy establishment e.g., our analysis does not rely on assumptions that are required to link tests found in core-top sediments to hydrographic conditions during their formation. This is particularly important for Polar and Arctic Oceans, considering the scale of recent warming and low sedimentation/accumulation rates that result in the mixing of pre-industrial with modern samples in the top 0.5 cm of sediment.

## Results and discussion

Like previous core top calibrations for *N. pachyderma*, our plankton tow-based results show a departure from the expected Mg/Ca-temperature relationship (Fig. 1a). To estimate the influence of seawater $[CO_3^{2-}]$ on Mg/Ca values, we isolate the contribution of $[CO_3^{2-}]$ on measured Mg/Ca values using water-column data for temperature and carbonate-ion concentration, following a similar approach as described by Evans et al.[35] and Morley et al.[39]. First, we normalised measured Mg/Ca values against predicted Mg/Ca values (Mg/Ca$_{[pred]}$) to reveal by how much the measured Mg/Ca is an overprediction of the expected. Mg/Ca$_{[pred]}$ represents the Mg/Ca sensitive to temperature only and is determined as Mg/Ca$_{[pred]}$=0.4exp$^{(0.09T)}$. This equation was derived for *N. incompta* calcifying at $[CO_3^{2-}]$ values <200 $\mu$mol.kg$^{-1}$ in Morley et al.[39]. Given that *N. incompta* and *N. pachyderma* are genetical sister species[43] and share ecological preferences[44], we assume that the temperature sensitivity is shared between both species. We then derive a Mg/Ca-$[CO_3^{2-}]$ sensitivity for all samples included in this study by comparing normalised Mg/Ca$_{[Norm]}$ against measured $[CO_3^{2-}]$ (Fig. 1b). Mg/Ca$_{[Corr]}$ (Fig. 1c) then reveals the temperature-only component and is calculated by dividing Mg/Ca$_{[meas]}$ over the Mg/Ca-$[CO_3^{2-}]$ sensitivity derived in Fig. 1b. We further complement the tow dataset with new core tops and paired hydrographic profiles including temperature, salinity, Dissolved Inorganic Carbon (DIC) and Alkalinity (ALK) from the Nordic Seas, and also include the published Kozdon et al.[27] core top dataset in our analysis using depth estimate from foraminifera $\delta^{18}O_c$ projected on equilibrium $\delta^{18}O_c$ derived from hydrographic profiles (see Methods). The comparisons of plankton tows with core tops allow us to test if the tow-derived Mg/Ca-$[CO_3^{2-}]$ sensitivity can be applied to climate archives.

The Mg/Ca-$[CO_3^{2-}]$ sensitivity for *N. pachyderma* from plankton tows (Fig. 1b) is consistent with calibrations for the closely related planktonic foraminifera *N. incompta*[39] but is more pronounced than in temperate planktonic foraminifera[35]. This can be explained by a non-linear response of $[CO_3^{2-}]$ on Mg incorporation at different temperatures. Specifically, the steeper response for subpolar and polar for-aminifera would imply that the $[CO_3^{2-}]$ effect is greater at low temperatures indicating that the high sensitivity of Mg to low $[CO_3^{2-}]$ may be linked to the decreasing efficiency of Mg exclusion under conditions less favourable for calcification. Along this line of argument, Evans et al.[45] hypothesised that for low-Mg foraminifera like *N. pachyderma*, the mechanism by which Mg is removed during calcification competes energetically with the need to modify the carbonate chemistry of seawater for calcification at low $[CO_3^{2-}]$. They propose

that it may be more effective for foraminifera to prioritise the removal of H$^+$ over Mg$^{2+}$ at low [CO$_3^{2-}$] (this is also supported by ref. 46). Also, low metabolic/respiration rates hypothesised at low temperatures may further reduce the energy available (required) to exclude Mg$^{2+}$ and modify the carbonate chemistry for calcification[27], leading to the negative relationship between Mg/Ca and [CO$_3^{2-}$] at low values.

Our Mg/Ca data from plankton samples provide an opportunity to directly quantify the carbonate ion concentration effect on Mg/Ca measured in Arctic *N. pachyderma*. This is because the carbonate ion concentration for the calcification environment can be determined reliably, and our sampled environments cover the entire range of low carbonate ion conditions from 200 to less than 50 μmol.kgSW$^{-1}$ (Fig. 1b). The results reveal that the departure of the measured Mg/Ca$_{[Meas]}$ from the theoretical Mg/Ca$_{[Pred]}$, calculated assuming they reflect only ambient temperature, is exponentially correlated with [CO$_3^{2-}$]. When deriving [CO$_3^{2-}$] values at the apparent calcification depths for the new and previously published core top values included in this study (see Methods), we find that the Mg/Ca sensitivity to

[CO$_3^{2-}$] for core top samples is within the 95% confidence intervals of the Mg/Ca sensitivity to [CO$_3^{2-}$] derived for plankton tow samples (Fig. 1b). This is important, as it allows us to apply the Mg/Ca sensitivity to [CO$_3^{2-}$] derived from plankton tows to *N. pachyderma* preserved at the seafloor and in climate archives.

Before we compared datasets, we took into account the difference in Mg/Ca between the crust and ontogenetic calcite for *N. pachyderma*[25,26,47]. Recent findings show that the Mg/Ca ratios of the outer test of *N. pachyderma* are between 8.6 to 15% higher in living (uncrusted) specimens when compared to the outer test of dead specimens recovered from the same plankton tows[48]. Similarly, the analysis of intra-test banding of Mg/Ca ratios in crusted *N. pachyderma* tests from North Atlantic sediment traps reveals that the addition of crust lowers overall Mg/Ca ratios of a specimen by 15%[34] when assuming an average crust volume of 50%. To compare Mg/Ca values from tows and core tops we therefore adjust uncorrected Mg/Ca values from plankton tow samples down by 15% to account for the difference in Mg between ontogenetic and crust calcite.

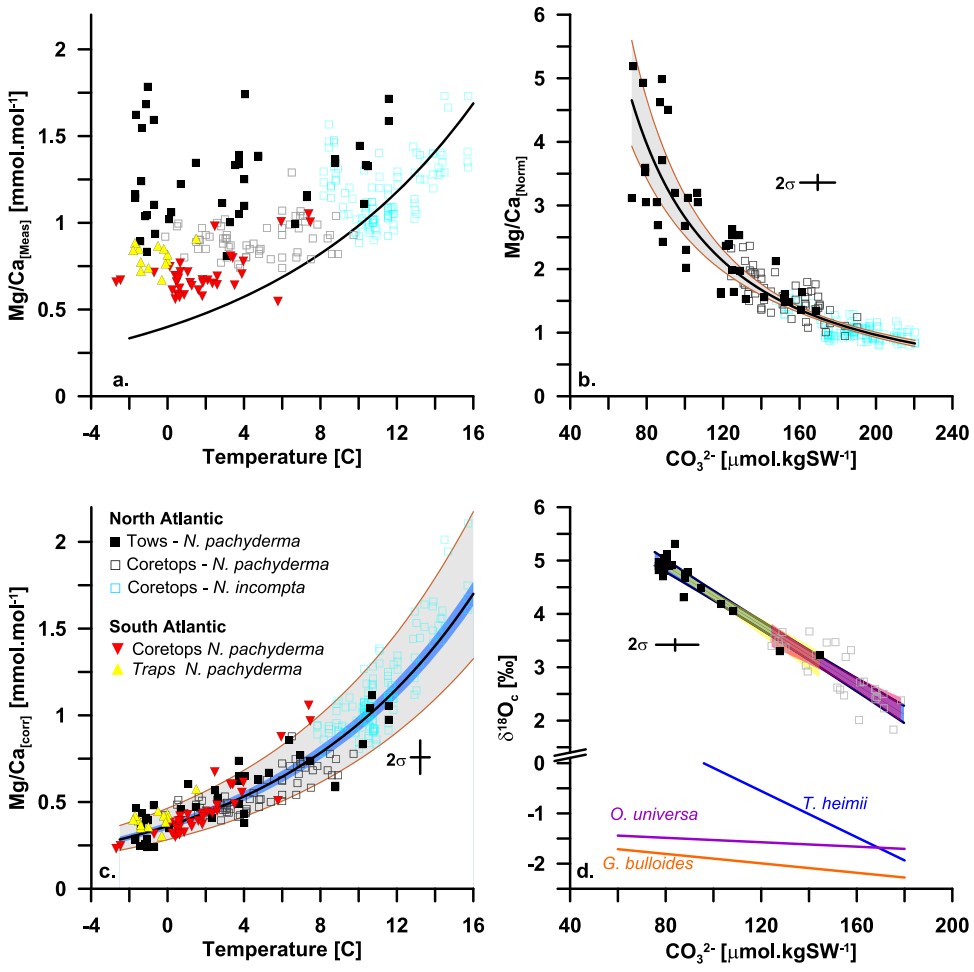

**Fig. 1 | Calibration.** In all panels, closed black squares refer to *N. pachyderma* collected from plankton tow samples, open dark grey squares represent *N. pachyderma* from core tops[27], and open turquoise squares represent *N. incompta* from core tops[39]. Inverted red triangles refer to *N. pachyderma* core top data from the South Atlantic and the Antarctic continental shelf[59], while yellow triangles refer to *N. pachyderma* sediment trap data from the Antarctic Peninsular[24]. **a** shows measured Mg/Ca values. **b** Shows normalised Mg/Ca values against water-column [CO$_3^{2-}$] for tows and δ$^{18}$O$_c$ derived [CO$_3^{2-}$] values for core tops. The grey shading connotes the 95% confidence intervals associated with the regression. **c** shows Mg/Ca values corrected for [CO$_3^{2-}$] seawater carbonate-ion concentration. The blue shading connotes the 95% confident intervals of fully propagated uncertainties for

the slope of the calibration relationship, while the grey shading connotes the fully predicted uncertainties which would apply to the full uncertainty on a single data point. **d** shows δ$^{18}$O$_c$ values for *N. pachyderma* vs water column [CO$_3^{2-}$]. *N. pachyderma* from plankton tows are corrected for a + 2‰ offset to account for the difference between ontogenetic and crust calcite. The blue shading connotes the 95% confidence intervals for the regression combining both tow and core top data, while the yellow and red shading connotes the 95% confidence intervals for the tows and core top regressions, respectively. In addition, function plots are shown for other unicellular planktonic calcifying organisms[53,54]. Source data for all panels are provided as a Source Data file.

With plankton tows, new, and previously published North Atlantic core top samples combined (e.g., ref. [20,27,39],), Mg/Ca$_{[Corr]}$ describes the following equation: Mg/Ca$_{[Corr]}$ = 0.372 ± 0.014exp$^{(0.0936±0.0045T)}$, $n = 235$, $r^2 = 0.90$, which shares the same sensitivity (within error) of the predicted temperature calibration equation, showing that [$CO_3^{2-}$] is the sole or at least the primary non-thermal control on the predicted Mg/Ca – temperature relationship (Fig. 1c). Our results also demonstrate that the influence of [$CO_3^{2-}$] on Mg/Ca measured on *N. pachyderma* can be mathematically quantified and isolated from temperature as for other planktonic foraminifera species (e.g., ref. [40],), when [$CO_3^{2-}$] is known. Thus, for applications of the Mg/Ca paleothermometer on fossil material, an independent [$CO_3^{2-}$] proxy is required. Here, we investigate the influence of [$CO_3^{2-}$] on the fractionation of oxygen isotopes measured in *N. pachyderma* tests and evaluate its potential to serve as a proxy for the Mg/Ca correction scheme. $\delta^{18}O_c$ is a traditional palaeoceanographic tracer combining the influence of $\delta^{18}O_{sw}$ of seawater and temperature[49–51]. However, in addition to temperature, a dependence of oxygen isotope fractionation on the carbonate chemistry of seawater has previously been demonstrated in planktonic foraminifera[52,53] and other unicellular planktonic calcifying organisms[54], suggesting the existence of a universal carbonate ion effect on stable oxygen isotope ratios measured in unicellular planktonic calcifying organisms.

To evaluate whether past seawater [$CO_3^{2-}$] values can be derived from $\delta^{18}O_c$ in the past, we first tested if the $\delta^{18}O_c$/[$CO_3^{2-}$] sensitivity observed in material from plankton tows and core tops (Fig. 1d) is compromised by interactions with $\delta^{18}O_{sw}$ and/or temperature in a generalised linear model including $\delta^{18}O_c$, $\delta^{18}O_{sw}$, [$CO_3^{2-}$], and temperature. We found that $\delta^{18}O_{sw}$ does not significantly contribute to the total variance ($n = 52$, $p > 0.01$, Supplementary Table 5) and therefore excluded it from considerations. Furthermore, we found that temperature only weakly contributes to the total variance ($p = 0.017$) observed in the model. Importantly, the interaction term between [$CO_3^{-2}$] and temperature is not significant ($n = 52$, $p = 0.14$), suggesting that while temperature contributes to the total variance in $\delta^{18}O_c$, it does not influence or interact with the $\delta^{18}O_c$/[$CO_3^{2-}$] relationship. Nevertheless, we evaluate the contribution of temperature and the potential of a non-linear effect of temperature on $\delta^{18}O_c$ further, using a regression tree (Supplementary Fig. S2). As expected, the regression tree shows that [$CO_3^{2-}$] explains most of the variance and that temperature contributes to the variance in the $\delta^{18}O_c$/[$CO_3^{2-}$] linear model only for values above 5 °C. When analysing the relationships between temperature, [$CO_3^{2-}$], and $\delta^{18}O_c$ separately, we find that temperature and [$CO_3^{2-}$] only covary above 5 °C or 150 μmol.kg$^{-1}$ but not at colder temperatures or lower concentrations (Supplementary Fig. S3). Similarly, the covariance between temperature and $\delta^{18}O_c$ breaks down below 4-5 °C (also observed in ref. [55]), while the relationship between $\delta^{18}O_c$ and [$CO_3^{2-}$] remains unchanged regardless of concentrations (Supplementary Fig. S3). In summary, while temperature contributes weakly to the variance in $\delta^{18}O_c$ above 5 °C, it does not influence the interaction between $\delta^{18}O_c$ and [$CO_3^{2-}$]. Furthermore, these results confirm that the $\delta^{18}O_c$/[$CO_3^{2-}$] sensitivity does not rely on covariance with temperature, which varies over space and time.

Following, we posit that the sensitivity or slope of $\delta^{18}O_c$/[$CO_3^{2-}$] is regulated by the calcite saturation product ($K_{sp}$), which in turn is determined by the conversion between (isotopically heavier) bicarbonate and (isotopically lighter) carbonate ions in the calcification environment, that is a function of pH, salinity, and to a lesser extent temperature[54]. We observe that the slopes of $\delta^{18}O_c$/[$CO_3^{2-}$] for *N. pachyderma* from tows and core tops are statistically the same at −0.0279 but offset by ~2‰ (Fig. 1d). The slopes in our dataset are also steeper than reported for *Orbulina universa* (−0.002) or *Globigerina bulloides* (e.g., −0.004)[53], but not outside of the slopes observed previously for other unicellular planktonic calcifying organisms[54] (Fig. 1d). The steeper slope for *N. pachyderma* suggests that the relative

contribution of the isotopically light [$CO_3^{2-}$] to the calcifying fluid is low and that comparatively less bicarbonate is converted into carbonate ions in the calcification environment when compared to *O. universa* or *G. bulloides*. This hypothesis agrees well with the biomineralization model controlling the exclusion/uptake of Mg described earlier in *N. pachyderma*, where low cell energy due to unfavourable calcification environments and low metabolic/respiration rates at cold temperatures leads to reduced [$CO_3^{2-}$] available for calcification[45,46,56]. In addition, the release of $CO_2$ during respiration and the absence of photosynthetic activity from symbionts, further decrease the relative contribution of isotopically light [$CO_3^{2-}$] in favour of the isotopically heavier bicarbonate and potentially further enhances the sensitivity of oxygen isotope fractionation to varying [$CO_3^{2-}$] in low [$CO_3^{2-}$] settings. The observed 2‰ offset in the intercepts between tows and core tops agrees well with the average 2.1‰ offset observed between ontogenetic calcite and a fully crusted *N. pachyderma*[57]. A caveat of this offset is that specimens of a similar degree of encrustation must be analysed together to avoid the influence of varying degrees of encrustation on the $\delta^{18}O_c$/[$CO_3^{2-}$] relationship. Accordingly, the larger scatter observed for the core top versus plankton tow dataset could potentially be explained by different degrees of crust formation[57].

To explore the application potential of the $\delta^{18}O_c$/[$CO_3^{2-}$] proxy to fossil material, we compared [$CO_3^{2-}$] derived from $\delta^{18}O_c$ with CARINA[58] derived [$CO_3^{2-}$] values for the Kozdon core top dataset (see Methods). Both methods provide [$CO_3^{2-}$] values that are within a root mean square error (RMSE) of 8.44 μmol.kg$^{-1}$ of each other (adjusted $r^2 = 0.95$ $n = 52$, $p < 0.001$). Residuals are symmetrically distributed around zero (Supplementary Fig. S4). When $\delta^{18}O_c$ derived [$CO_3^{2-}$] values are used to estimate the Mg/Ca-[$CO_3^{2-}$] sensitivity, the scatter is much lower (when compared to CARINA), resulting in more robust correlation statistics (adjusted $r^2 = 0.81$ vs $r^2 = 0.58$) and lower uncertainties. The reduced scatter is most likely due to the uncertainties linked to assigning modern water column hydrography from CARINA to pre-industrial/late Holocene core tops. By using $\delta^{18}O_c$ to derive [$CO_3^{2-}$], these uncertainties are removed. To further test the validity of this approach, we also derived [$CO_3^{2-}$] from $\delta^{18}O_c$ values published in the core top dataset (paired Mg/Ca - $\delta^{18}O_c$) by Meland et al.[20]. and find that the Mg/Ca sensitivity to [$CO_3^{2-}$] is statistically identical with the tows and the Kozdon datasets (Fig. 1b).

Finally, we combine all *N. pachyderma* datasets (e.g., this study,[20,27]) with a previously published dataset for *N. incompta*[39]. When all datasets are combined, we derive the following Mg/Ca - [$CO_3^{2-}$] sensitivity: Mg/Ca[$CO_3^{2}$] = 3227.95*[$CO_3^{2-}$]$^{-1.53}$ ($n = 235$, $r^2 = 0.87$, $p < 0.01$). All uncertainties (including calibration uncertainties accounting for the covariance of error in calibration coefficients) are propagated through Monte Carlo simulation [e.g., 40]. Briefly, to account for dependent and independent uncertainties, 1000 potential calibration slopes are created from selecting random environmental values within the 2σ uncertainty ranges for individual parameters ($\delta^{18}O_c$, [$CO_3^{2-}$], Mg/Ca, and Temperature, see Methods). The median of these slopes is taken as our central calibration estimate with the 2σ described by the 95th percentile of potential slopes. We also include the prediction interval for the Monte Carlo Ensemble (Fig. 1c, see Methods for full details). Typical uncertainty in calibrated temperatures, combining calibration uncertainty, and [$CO_3^{2-}$] uncertainty, is ±1.0 °C (±1.2 °C at 0 °C and ±0.9 °C at 15 °C, 2σ) if $\delta^{18}O_c$ are used to correct for [$CO_3^{2-}$].

To evaluate the suitability of the proposed equations to other regions and genotypes of *N. pachyderma,* we applied the correction scheme to the comprehensive core top dataset published in Vazquez-Riveiros et al.[59]. that includes 38 core top samples from the subpolar South Atlantic to the Antarctic continental shelf and the sediment trap series from the Antarctic Peninsula published in Hendry et al.[24]. These datasets are dominated by genotype IV, which is the only *N.*

*pachyderma* genotype adapted to overwintering in sea ice[60]. Nineteen of the 38 core top sites, as well as the sediment trap series are situated on the Antarctic margin and are therefore subject to the possible influence of sea ice on Mg/Ca ratios[24,59]. When Mg/Ca are corrected for $\delta^{18}O_c$-derived $[CO_3^{2-}]$ values, the Mg/Ca-temperature correlation statistics for the South Atlantic core top dataset are much more robust (e.g., $r^2 = 0.87$ vs. $r^2 = 0.30$, $n = 36$, see also data plotted in Fig. 1a, c) and within the uncertainties of the North Atlantic derived dataset. Similarly, the sediment traps from Hendry et al.[24]. agree well with the North Atlantic dataset (Fig. 1a, c). These applications, therefore, suggest that the proposed correction scheme can be applied to both hemispheres and that genotypes do not influence the geochemical climate signature recorded in *N. pachyderma* specimens as previously hypothesised[60].

The combination of the Mg/Ca correction scheme and $\delta^{18}O_c$ derived $[CO_3^{2-}]$ values presents a unique opportunity to apply the correction to previously published Mg/Ca - $\delta^{18}O_c$ datasets and estimate SSTs in polar environments. In this way, it allows us to reevaluate the magnitude and spatial expression of marine polar amplification to glacial-interglacial climate forcing as recorded by Mg/Ca and $\delta^{18}O_c$ values. Using previously published Mg/Ca - $\delta^{18}O_c$ datasets[29,30,61], we show that uncorrected Mg/Ca-based SSTs systematically underestimate maximum glacial cooling by 2 to 3° ± 1.0 °C for four previous glacial maxima (Fig. 2 and Supplementary Table 6). Furthermore, the magnitude of warming from glacial to interglacial SSTs in the Labrador Sea at IODP Site U1305 is significantly higher based on Mg/Ca[corr] values for the past five glacial-interglacial cycles increasing from 2.3-4.2 °C for uncorrected Mg/Ca-derived values to 7.9-11.0 ± 1.0 °C

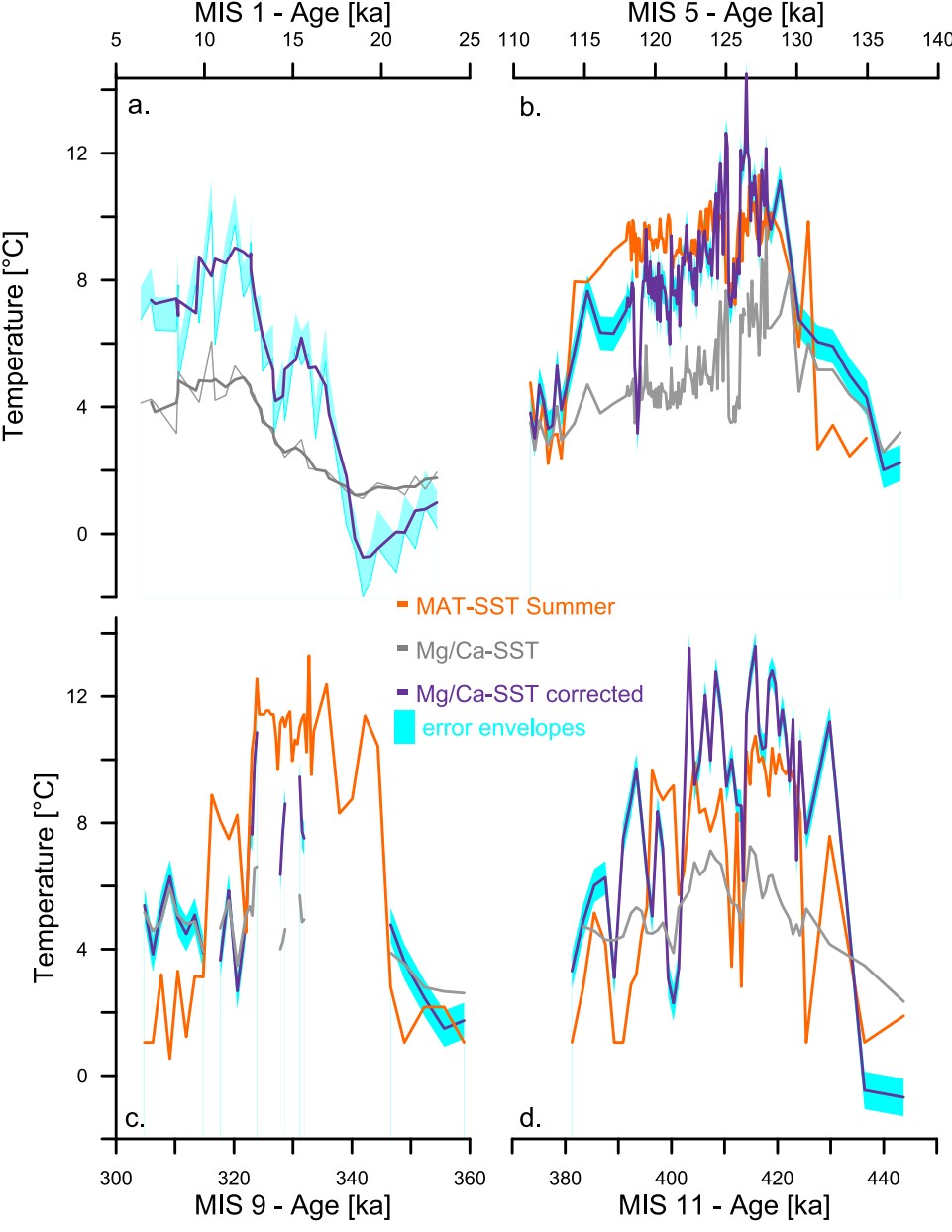

**Fig. 2 | Paleo application of Mg/Ca correction scheme to four glacial-interglacial periods including Marine Isotope Stage 1 (MIS1), MIS5, MIS9 and MIS11**[29,30,61]**.** Datasets originate from Eirik Drift, Southeast of Greenland, from IODP site U1305, and sister site MD99-2227. Purple lines show three-point running averages of corrected Mg/Ca-based sea surface temperatures (SSTs), while grey lines show three-point running averages of uncorrected Mg/Ca-based SSTs reconstructions. The blue shading shows the 95% confidence interval associated with Mg/Ca[Corr] SST reconstructions. In orange are Modern Analogue Technique (MAT)-based SSTs reconstructions as shown in Irvali et al.[30]. for MIS 5 (panel b.), 9 (panel c.), and 11(panel d.). Note that MAT-based SSTs are not available for the last glacial-interglacial transition (panel a.). Source data for all panels are provided as a Source Data file.

once corrected. This increase in gradient reflects both colder SSTs during glacial periods and warmer SSTs during interglacials, highlighting the importance of applying the correction to both glacial and interglacial periods. Assemblage-based (e.g., modern analogue technique; MAT[62]) glacial-interglacial SST gradients for the same site range between 7–9 °C. MAT-based glacial-interglacial SST gradients are, on average, 1.5 °C smaller than our corrected Mg/Ca-based estimates, which is likely the result of low species diversity below 4 °C leading to higher SST values during glacial maxima[63].

A paleoceanographic map of corrected *N. pachyderma* Mg/Ca records covering the LGM and Late Holocene demonstrates that the underestimated cooling (e.g., ΔSST) is most pronounced across mid-latitudes where our estimates suggest cooling between 6–9 ± 1.0 °C instead of milder 4-6° C cooling (Fig. 3a). Our re-assessment suggests that polar amplification or high-latitude LGM cooling was of a greater magnitude and characterized by a more extensive spatial expression than previously estimated, which raises questions about our understanding of heat transport under glacial boundary conditions. Furthermore, absolute Holocene and LGM SSTs values based on corrected *N. pachyderma* Mg/Ca records (Fig. 3b, c) are much closer to model-based reconstructions of Holocene and LGM SSTs than previous estimates based on this species (Supplementary Fig. S5). Considering these examples, we believe that a systematic application of the calibration approach, as presented in our study, provides, an opportunity to directly estimate past temperatures in Polar Oceans, where low diversity of planktonic foraminifera prevented the use of transfer functions and where dinocysts are rare.

Furthermore, these corrected datasets demonstrate that paired Mg/Ca · δ$^{18}$O$_c$ records, including reanalysis of published data, can now answer the call by the modelling community to reduce uncertainties linked to high latitude SST reconstructions and offer a more robust means of data-model comparison. In particular, Mg/Ca-based SST reconstructions in *N. pachyderma* can now evaluate the ability of climate models to simulate polar amplified warming and cooling, which is especially important as climate model simulations targeting warmer than present climates have historically not captured the full extent of polar amplified warming[12,64]. Further, more accurate high-latitude paleoclimate data will inform the latest generation of climate models that are using paleoclimate-calibrated Community Earth System Models (CESM2)[12,65] to simulate multiple paleoclimate states—warmer and colder than the present. Finally, the corrected polar paleoclimate datasets will be a critical tool to refine estimates on the sensitivity of ice sheets to melting (e.g., sea level rise during warm climates[66]), and to quantify how AMOC strength affects ocean heat transport on orbital to millennial timescales through glacial-interglacial cycles[67,68].

## Methods

### Sample selection & preparation
*N. pachyderma* specimens were picked from multinet (HydroBios, type Midi) plankton tows collected during four separate oceanographic cruises, MSM09 (2008), MSM44 (2015), and MSM66 (2017) to the Labrador Sea/Baffin Bay and CE20009 (2020) to the Nordic Seas (see Supplementary Tables 1–3). Foraminifera were either immediately picked and placed into slides on board and subsequently frozen or directly frozen at − 80 °C to prevent degradation of the cytoplasm before picking. This also allowed for clear separation of specimens that were alive when collected (cytoplasm bearing) from specimens that were dead (empty tests) for subsequent geochemical analysis. Specimen were divided into four size fractions where possible: 100–150 µm, 150–200 µm, 200–250 µm and 250–300 µm. Weights per individual were determined using a Sartorius XM1000P microbalance (nominal resolution of 1 µg) at the University of Bremen and a Sartorius Semi-microbalance SECURA (nominal resolution of 2 µg) at the University of Galway before δ$^{18}$O$_c$ and Mg/Ca analysis. Given the low average weights of tow specimens (e.g., 2.12 µg per individual), at least 20 specimens per size fraction were required for δ$^{18}$O$_c$ and, on average, ~ 320 individuals for Trace Element (TE) analysis. To determine individual test weights, the total number of individuals measured was divided by the final weight per sample. Core tops analysed in this study were washed and dried to separate fine from coarse fractions. Between 250 and 350 specimens were picked within the size fraction 200–250 µm. While we selected specimens from this narrow size fraction, we did not perform a systematic analysis of the degree of encrustation of *N. pachyderma* in the core top samples.

### Hydrographic data & processing
Water column carbonate chemistry for each tow station was derived from hydrographic surveys carried out as close to the date and station as possible. For example, for stations from MSM 09, the CLIVAR and Davies Strait Time Series were collected sometimes within days (at most 1 month) and within 1 degree latitude and longitude of tow samples. When more than one station was available within a 1-degree radius, average values were calculated. Supplementary Table 1 provides a detailed list of datasets and stations used for carbonate chemistry, including coordinates and dates. For some stations, carbonate chemistry data from the same year and area was not available; in these cases, only profiles collected during the same season were considered and evaluated against available CTD data. For CE20009, paired hydrographic data and water samples, including ALK and DIC, were collected. ALK and DIC were analysed simultaneously from the same sample at the University of Galway using a VINDTA-3C[69] following the methods outlined in Dickson et al.[70]. The VINDTA-3C was calibrated by duplicate measurement of Certified Reference Materials (A. Dickson, Scripps Institute of Oceanography, USA) with every batch of samples. The remaining carbonate system parameters were calculated using the Seacarb package in R.

For the core top datasets, calcification depth and hydrographic parameters were estimated by comparing the measured foraminiferal δ$^{18}$O$_c$ values to the expected equilibrium values of calcite (δ$^{18}$O$_c$) precipitated from seawater. For the Kozdon dataset, we note that specimens for Mg/Ca and δ$^{18}$O$_c$ analysis were picked from the same sample but are not paired i.e., they constitute two separate sets of specimens. To determine calcification depth, we selected temperature, salinity, and carbonate chemistry using CARINA[58], while paired hydrographic profiles were used for CE20009 core tops. Since the dominant flux of *N. pachyderma* occurs in the summer, we only selected profiles collected between June and October from stations as close to the core sites as possible for the Kozdon dataset. Specific stations, locations and dates of profiles are detailed in the Supplementary Table 8. To obtain δ$^{18}$O$_{sw}$ depth profiles, we used modelled water-column data from LeGrande and Schmidt[71]. We retrieved data from both the closest station to each core site and within a 0.5° radius to provide locally averaged δ$^{18}$O$_{sw}$ values. Despite these constraints, it is possible that the hydrographic profiles used here do not represent the calcification environment of specimens recovered from core tops. For example, unrealistically deep calcification habitat estimates below 300 m may result from modern hydrographic profiles that are "too warm" for samples that predate modern global warming. For the CE20009 core tops, paired water column δ$^{18}$O$_{sw}$ values are available. We then calculated site-specific δ$^{18}$O$_{sw}$ - salinity linear relationships for each core. Using the δ$^{18}$O - paleotemperature equation developed by O'Neil et al.[72]. we determined expected δ$^{18}$O$_c$ depth profiles (0–400 m) based on seasonal temperature, salinity, and δ$^{18}$O$_{sw}$ data. Calculations were completed by subtracting 0.27‰ to compare measured δ$^{18}$O values of CO$_2$ produced by the reaction of calcite with H$_3$PO$_4$ and CO$_2$ equilibrated with water[73]. The depth at which the expected equilibrium δ$^{18}$O$_c$ values matched the measured foraminiferal δ$^{18}$O$_c$ values (determined by linear interpolation) is the calcification depth, and the calcification temperature, salinity, and carbonate chemistry were determined based on respective values recorded at this calcification depth.

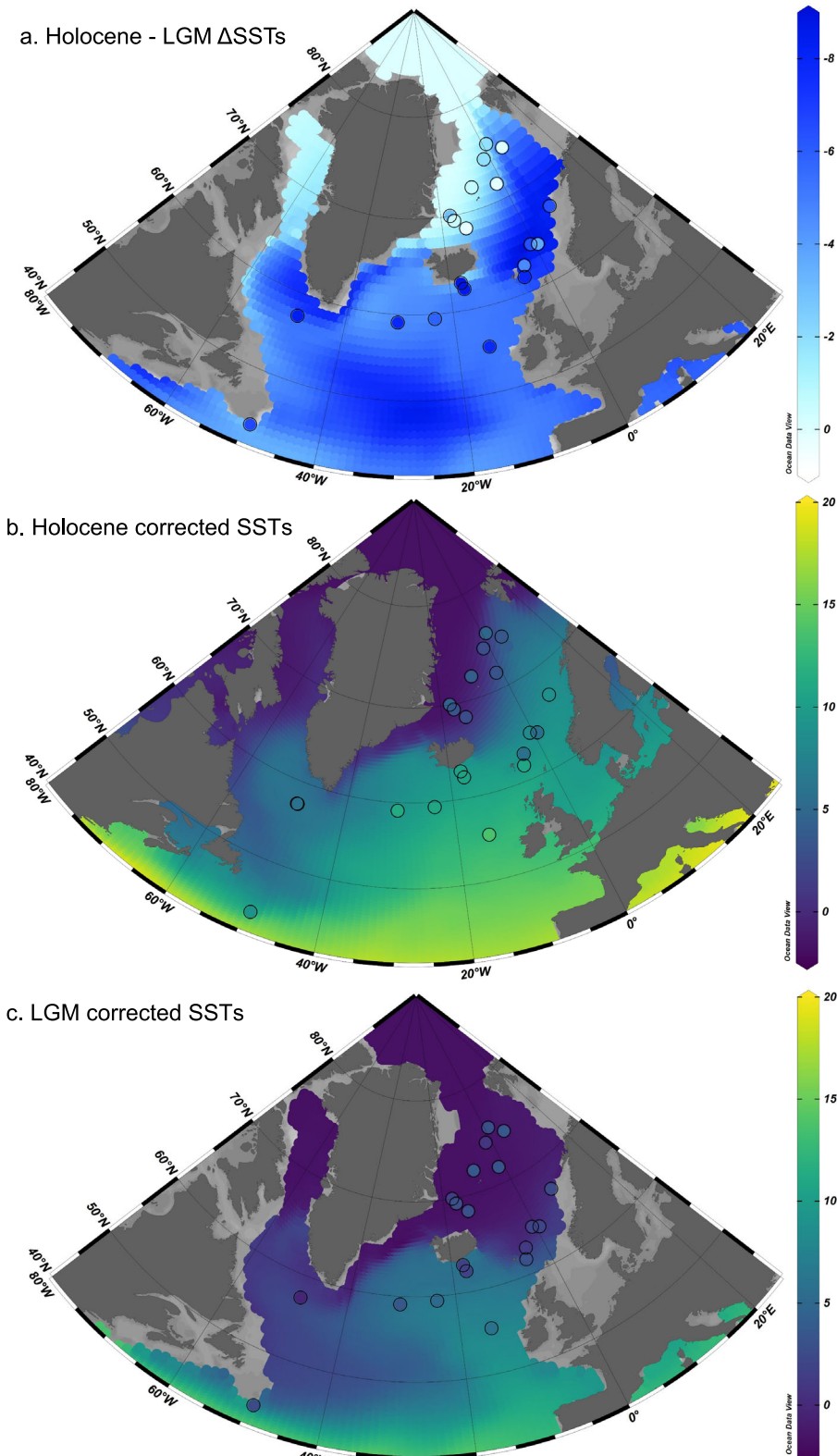

**Fig. 3 | Data model inter-comparison.** Panel (**a**) shows the difference in sea surface temperature cooling (ΔSST) between the Holocene and the Last Glacial Maximum calculated using the isotope-enabled Community Earth System Model (iCESM) as shown in Tierney et al. [28]. In the background together with ΔSST calculated using corrected Mg/Ca values measured on *N. pachyderma* (symbols). For a list of all datasets used, please see Supplementary Table 7. Panels (**b**) and (**c**) show Holocene and LGM SSTs, respectively, with the modelled (iCESM) SSTs in the background and

corrected Mg/Ca-based SSTs shown as symbols. Maps were produced with Ocean Data View[93]. Please note that modelled annual SSTs are likely too cold for a direct comparison with *N. pachyderma*-based SSTs, given their seasonal reproductive cycle. Maps displaying uncorrected Mg/Ca values against modelled paleo data are shown in Supplementary Fig. S5. Source data for all panels are provided as a Source Data file.

We did not apply a systematic correction for a "vital effect" on $\delta^{18}O_c$ values because the application of such a vital effect when estimating calcification depth for core top *N. pachyderma* is complicated by: (1) the $\delta^{18}O_c$ offset from equilibrium measured in *N. pachyderma* collected from plankton tows is highly variable within a region, across regions, seasons, and on interannual timescales[26,74–76]; (2) the addition of crust, typical for specimens recovered from core tops, increases the $\delta^{18}O_c$ signal because crust is isotopically heavier than ontogenetic calcite[57,77]. Variations in the degree of crusting can, therefore, either "lower", completely "mask", or shift the vital effect towards positive values depending on the degree of crusting[57]. As a result, there remains considerable uncertainty in the exact value to use when correcting for the competing signals of vital effects and crust in *N. pachyderma*, with studies reporting both calcification in equilibrium[77–80] and out of equilibrium with seawater[55,81–83].

Since it is not possible to isolate the vital effect from crust in core top specimens the application of a fixed offset leads to assumptions of calcification habitats that are not grounded in ecological observations. Specifically, it has long been assumed that the isotopically heavier crust calcite results from systematic ontogenic vertical migration to deeper and colder waters associated with reproduction[31]. However, from comprehensive, cross-regional plankton tow studies, we know that heavily crusted specimens can be found at any depth, including the top 0–50 m[32] suggesting that there is no systematic ontogenetic vertical migration associated with reproduction or "crusting" for *N. pachyderma*[32,33]. Furthermore, culture-grown crust calcite in *N. incompta* has a lower Mg/Ca ratio than ontogenetic calcite formed at the same temperature, suggesting that a lower temperature is not responsible for the low Mg/Ca ratio of neogloboquadrinid crusts[25]. Finally, cross-sections of *N. pachyderma* tests from North Atlantic sediment traps also show that tests growing in a well-mixed water column with constant temperatures exhibit lower Mg/Ca ratios in their crust[34], suggesting that alongside temperature, other factors control the difference between crust and ontogenetic Mg/Ca.

By not applying a correction for vital effects or the contribution of crust calcite we avoid biasing calcification depth estimates towards deeper depth. This approach is also consistent with sediment trap studies from both the North and South Atlantic, where $\delta^{18}O_c$ values measured on crusted *N. pachyderma* are in equilibrium with measured water column $\delta^{18}O_c$[77,78]. Except for two stations that yield anomalously deep estimates, all calcification depths fall between 7 and $67 \pm 22$ m $2\sigma$ water depth (Supplementary Table 8). Compared to the existing literature, these are at the shallow end of previously reported depth estimates[82] but are in line with a seasonal calcification depth of 50 m[80] and observations that place the main seasonal bloom of *N. pachyderma* at the base of the chlorophyll max following the breakup of sea-ice[84].

For South Atlantic sediment traps included here, we note that crusted specimens were specifically excluded from the analysis to avoid the contribution of secondary or gametogenic crust calcite[24]. We, therefore, applied the vital effect as derived in Hendry et al[24]. when estimating $[CO_3^{2-}]$ values and lowered Mg/Ca ratios by 15% to account for the difference in Mg between ontogenetic and crust calcite[34,48]. No vital effect or correction on Mg/Ca values was applied to the core top dataset from the South Atlantic[59] consistent with the processing of North Atlantic core top datasets analysed here.

## Analytical methods

To prepare foraminifera from plankton tows from MSM cruises for TE analysis, tests were gently crushed between two glass plates and transferred into acid-leached vials. The crushed foraminiferal tests were cleaned using a cleaning protocol focussing on the removal of organic matter (i.e., the oxidative step was repeated twice) followed by a final dissolution in dilute $HNO_3$. We did not pre-rinse samples with Milli-Q or include a reductive cleaning step since clay or metal oxide

removal is not required for tow samples. Samples from MSM cruises were analysed using a Nu AttoM high-resolution double-focusing inductively coupled plasma mass spectrometer at the Alfred Wegner Institute in Bremerhaven, Germany. Five replications were carried out for each sample, where the average repeatability for Mg/Ca among the five replicates was $0.95 \pm 0.47\%$ (relative standard deviation). Long-term reproducibility of Mg/Ca analyses is monitored by regular analysis of the carbonate standard JCp-1, yielding $3.94 \pm 0.15$ ($2\sigma$) mmol.mol$^{-1}$ over two years. Samples from cruise CE20009 were processed for elemental analysis at the University of Southampton. They were cleaned using established methods following Barker et al.[85], clay was removed via multiple Milli-Q rinses (for core top samples only) and the tests were oxidatively cleaned. Elemental analysis was performed on a Thermo Element Inductively Couple Plasma Mass Spectrometer (ICPMS) following Henehan et al.[86]. Stable isotope analysis on MSM samples was performed on a Finnigan MAT 252 mass spectrometer equipped with an automated carbonate preparation device at MARUM, University of Bremen. Isotopic results were calibrated relative to the Vienna Pee Dee Belemnite (VPDB) using the NBS19 standard. The standard deviation of the laboratory standard was better than 0.07‰ for $\delta^{18}O_c$ over the measuring period. Samples from CE20009 were analysed at FARLAB, using a Thermo Scientific MAT 253 IRMS coupled to a Kiel IV automated carbonate preparation device. Results are reported on the VPDB scale calibrated with NBS19, together with NBS18, and in-house synthetic carbonate standards CM12 and Riedal used in daily operations at FARLAB, University of Bergen. Long-term external precision (1 σ SD) is better than 0.08‰ for $\delta^{18}O_c$ based on replicate analysis of the in-house standard CM12.

## Paleodata correction

For all previously published datasets, $\delta^{18}O_c$ data were corrected for glacial-interglacial sea-level changes using sea-level reconstructions by Spratt and Lisiecki[87]. Here we use the commonly cited value of 0.11‰/ 10 m SL change for the correction of $\delta^{18}O_c$[88–91]. We also applied a $+10$–15% offset to Mg/Ca$_{[meas]}$ values that underwent a reductive cleaning step (e.g., ref. 29,30,) or repeated acid leaches[92] since these additional steps have been shown to lower Mg/Ca values in planktonic foraminifera[85]. We then derived paleo $[CO_3^{2-}]$ values from $\delta^{18}O_c$ and corrected Mg/Ca$_{[meas]}$ using the correction scheme presented here. For the Late Holocene – LGM compilation, we included as many available records from literature as possible. Supplementary Table 7 details core sites, time intervals and references used for each estimate. For the calculation of mean glacial and interglacial values covering the last five glacial-interglacial cycles from IODP site U1305, please see Supplementary Table 6.

## Calibration uncertainty calculations

For the *N. pachyderma* data, first the relationship between $\delta^{18}O$ and carbonate ion concentration is assessed (Fig. 1d). We performed a Monte Carlo simulation ($n = 1000$) on the slope of this relationship using the analytical uncertainty on the oxygen isotope measurements (typically 0.2 at $2\sigma$) and carbonate ion concentration from hydrographic data (GLODAP) with a conservative prescribed uncertainty of $\pm 20$ μmol/kg ($2\sigma$). This gives a range of potential slopes which can be used to determine a central estimate and uncertainties defined by the distribution of potential slopes (inside 95% of the potential slopes). The central coefficients are used to calculate the $CO_3^{2-}$:Mg/Ca$_{[norm]}$ relationship alongside the error on this slope to determine the uncertainties on the *x*-axis (Fig. 1b). These uncertainties are then propagated into the $CO_3^{2-}$:Mg/Ca$_{[norm]}$ relationship, where $CO_3^{2-}$ as given by $\delta^{18}O$ is plotted against normalised Mg/Ca (Mg/Ca$_{[meas]}$/Mg/Ca$_{[pred]}$) to obtain a Mg/Ca$_{[corr]}$ for non-thermal effects alongside its uncertainties. Based on the previous relationships (Fig. 1a, b), Mg/Ca$_{[corr]}$ has an uncertainty of $\pm 0.2$ ($2\sigma$), which we then calibrate against the in situ temperatures obtained from CTDs and/or the CARINA database.

These errors are related to the depth estimation of the foraminifera from the isotope data but typically range from $0 - 3\,°C$ (minimum $0\,°C$, maximum $10\,°C$, average $\pm 2.8\,°C$ with a standard deviation of $0.9\,°C$). Note, for the data from Meland et al.[20] we are unable to calculate uncertainty, so we use the average of the uncertainties from Kozdon et al.[27] which is 1.6 °C at $2\sigma$). For the *N. incompta* data, all the steps to calculate the uncertainties are identical to the *N. pachyderma* dataset except, as there are no $\delta^{18}O$ data, we assigned a temperature uncertainty of $\pm 0.8\,°C$ ($2\sigma$)[39]. The Mg/Ca$_{[corr]}$ uncertainty is then assumed to be the same as for *N. pachyderma*. Both species are combined to make the final calibration but using one or the other only gives consistent relationships within error, strengthening the argument against a species-specific vital effect.

The calibration Monte Carlo simulation finally is conducted on 1000 iterations of a slope using the propagated Mg/Ca$_{[corr]}$ and the individual temperature uncertainties (as above). We take the median slope of this curve to define the Mg/Ca$_{[corr]}$ to temperature relationship, and the 68 and 95 % confidence intervals (1 and $2\sigma$ respectively) to describe its uncertainties. These uncertainties represent the uncertainty in the defined relationship, assuming it to be consistent and are associated with a temperature reconstruction uncertainty of $\pm 1.2\,°C$ at temperatures of around $0\,°C$ and $\pm 0.9\,°C$ at temperatures of around $15\,°C$. These confidence intervals give a conservative uncertainty on the relationship between Mg/Ca$_{[corr]}$ and temperature and are used for any relative change calculations presented in this study. We also plot a prediction uncertainty, which can be considered the uncertainty from a single future or additional data point but is overly conservative for the whole population of data.

## Data availability
All new data and recalculated datasets shown here are available in the Supplementary Information/Source Data file. Source data are provided in this paper.

## Code availability
The fully commented code for the error propagation calculation and input files have been deposited in the Zenodo database under DOI: 10.5281/zenodo.13833954.

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

## Acknowledgements

This research was funded by MSCA-IF Project ARCTICO [838529], the Marine Institute of Ireland Research Programme 2014-2020 (PDOC/19/05/02), and Science Foundation Ireland Frontiers for the Future Project 21/FFP-P/10261, awarded to A.M. In addition, A.M. acknowledges Grant in Aid funding from the Marine Institute for research expedition CE20009 on the RV Celtic Explorer with special thanks to the crew sailing under Master Anthony Hobin and Research Assistant Aedin McAlear for the analysis of Ocean ALK and DIC samples from the CE20009 survey. M.R. acknowledges DFG (German Research Foundation) for funding through research grant no. RA 2068/4-1. R.C. acknowledges funding for Project CE2COAST, funded by ANR (FR), BELSPO (BE), FCT (PT), IZM (LV), MI (IE), MIUR (IT), Rannis (IS) and RCN (NO) through the 2019 "Joint Transnational Call on Next Generation Climate Science in Europe for Oceans" initiated by JPI Climate and JPI Oceans.

## Author contributions

A.M. designed the project, carried out palaeoceanographic sample preparation of plankton tows from the Labrador Sea, compiled and analysed all datasets, wrote the manuscript, and coordinated the input of all co-authors. E.D.L.V. carried out palaeoceanographic sample preparation and geochemical analysis of plankton tows and core tops from CE20009 and contributed to the writing of the final manuscript. M.R. performed trace element and boron isotope analysis on *N. pachyderma* plankton tow samples from the Labrador Sea. J.B. supervised A.M. at the Alfred Wegner Institute and oversaw trace element and boron isotope analysis. U.N. oversaw all stable isotope analyses of water and foraminifera samples from CE20009 and contributed to the final version of this manuscript. G.F. oversaw trace element analysis on *N. pachyderma* plankton tow samples from CE20009 and contributed to the final version of this manuscript. T.C. performed uncertainty calculations for the Mg/Ca correction and contributed to the final version of this manuscript. J.M. collected all plankton tows during CE20009 and contributed to the final version of this manuscript. R.C. oversaw Alkalinity and DIC analysis on seawater samples collected during CE20009 and contributed to the final version of this manuscript. J.B. performed Alkalinity and DIC analysis on seawater samples collected during CE20009 and contributed to the final version of this manuscript. M.K. advised A.M. during project design, supervised her during her MSCA-IF at MARUM, University of Bremen, and contributed to the final version of this manuscript.

## Competing interests

The authors declare no competing interests.
