## [Transparent Peer Review file · Nature Communications]

A solution for constraining past marine Polar Amplification

Corresponding Author: Dr Audrey Morley

Version 0:

Reviewer comments:

Reviewer #1

(Remarks to the Author)

Review: A solution for constraining past marine Polar Amplification

Submitted to Nature Communications by Morley et al.

Morley et al. submitted a very interesting manuscript to Nature Communications: Since the seminal study of Nuernberg (1995) that is widely regarded as the starting point to measure the Mg/Ca ratio in foraminiferal shells as a temperature proxy, it is known that Mg/Ca ratios in shells of the polar to subpolar foraminifera *N. pachyderma* are “too high” in water temperatures colder than about 4°C. These aberrant Mg/Ca ratios have been somehow puzzling and were confirmed by several follow-up studies.

The performance of other proxies that work well in warmer waters such as the d18O composition of foraminifera shells is also limited in the polar to subpolar ocean. Thus, the fidelity of available paleoclimate reconstructions from the northern North Atlantic is relatively poor, even though this region plays an important role in modulating the climate of the northern hemisphere. Therefore, robust paleoclimate data from the polar to subpolar realm are essential and sorely needed to test and improve climate model simulations.

In this study, Morley et al. presents an explanation for the aberrant (‘too high’) Mg/Ca ratios in *N. pachyderma* from high latitude oceans: elevated Mg/Ca is linked to a low carbonate ion concentration. The authors developed a new approach to correct for the secondary control of the carbonate ion concentration on Mg/Ca with the ultimate goal to obtain much more robust Mg/Ca-based paleotemperatures. Morley et al. furthermore present a technique that allows for the reconstruction of the past carbon ion concentration based on the oxygen isotope ratio measured in foraminiferal shells. Thus, existing records with paired Mg/Ca – d18O analyses can be corrected for the carbonate ion effect on Mg/Ca. The authors apply this approach on a downcore record and found much higher glacial-interglacial temperature swings than inferred from previous (uncorrected) data.

This study and the results are timely, important, and transformative. We definitely need better paleoclimate records from high latitude oceans, and this study is a welcomed contribution to solve long-standing challenges. However, in its current form, the manuscript needs moderate to major revisions – I made many comments in the attached PDF (and Excel file) and hope that they are helpful to further improve the manuscript.

My main concern is centered on the calcification depth: Morley et al. suggests relatively shallow average calcification depths for *N. pachyderma* (of course, the habitat depth changes during the ontogenetic development of the foraminifera, and at the end, we get an average value). Many older studies (e.g. Stangeew, Jensen, Simstich ... - details are in the annotated PDF) using plankton tows are reporting calcification depths of *N. pachyderma* that may reach well below >150 m in some areas of the Nordic Seas (however, the calcification depth is variable and may be shallower in other areas). Other studies try to assign the calcification depth based on a match between foraminiferal d18O and the depth in the water column where we would expect formation of that shell-d18O value in equilibrium with d18O seawater and ambient temperature. These studies also suggest larger calcification depth (e.g., Simstich et al.). In addition, many *N. pachyderma* studies are reporting a vital effect in d18O that directly affects depth reconstructions.

Thus, in a nutshell, I believe that the average calcification depths modeled/calculated by Morley et al., are quite shallow and conflict with some of the well-established literature. I am not suggesting that they are wrong – we have to accept that many aspects of the life cycle of *N. pachyderma* are not well known, and we can also challenge some of the existing literature. However, I would expect a much more detailed discussion about the depth habitat and prior studies.

What about vital effects in d18O? What about the formation of the crust (relevant for core tops)? What is the water depth of

crust formation? If the crust contributes more than 50% to the total shell mass (suggested by some studies), what will be the 'recorded' water depth of that shell? I believe that a new paragraph discussing past studies assessing the depth habitat of *N. pachyderma* and a justification of the depth habitat suggested by Morley et al. would be very helpful. In addition, I believe that the figures can be improved. At least on my computer, they appeared to be in low-resolution. Again, I made my detailed comments in the attached PDF.

In summary, this is a very important paper that requires moderate to major revision before I consider it ready for publication. However, the findings are of great importance, and I believe that this manuscript is -in general- suitable for Nature Communications. I am looking forward to seeing it published.

Reinhard Kozdon

Reviewer #2

(Remarks to the Author)

A solution for constraining past marine Polar Amplification - Morley, et al. 2024

This study is an important contribution to improving Arctic and sub-Arctic Pleistocene sea surface temperature (SST) estimation. It provides extensive new geochemical data (paired Mg/Ca - oxygen isotopes), from plankton samples, of the dominant modern and Pleistocene planktic foraminifera in these regions. Pairing this geochemistry data with seawater chemistry, the study improves the understanding of, and correction for, carbonate ion effects on both the Mg/Ca and oxygen isotope composition of foraminiferal calcite. The study then uses these new empirical correction factors to review published paired Mg/Ca - oxygen isotope datasets for a number of glacial - interglacial transitions, and demonstrates a larger magnitude of SST change, which is more in line with climate model and foraminiferal modern analogue techniques. I would recommend publication with minor revisions - the points to consider are listed below.

Figure 1b – isn't this normalised Mg/Ca[norm] dimensionless? A ratio of Mg/Ca / Mg/Ca? The values of this are not "Mg/Ca" values, but how much the measured Mg/Ca is an overprediction of expected – the scale is 1x, 2x, 3x – not a scale in "mmol.mol⁻¹".

This then makes sense of panel 1c, which is back in mmol.mol⁻¹ – as it's the measured value divided by the correction factor.

Figure 1c – I can't see any pink crosses for the uncorrected *N. pachyderma*.

- What are the two "uncertainty" metrics on this plot? – not clear what the "95% CIs of "fully propagated uncertainties" is from / refers to. Please clarify.

Lines 173 – 178 – if this crust v. ontogenetic calcite Mg/Ca is applied prior to the plotting of the data in figure 1, then I would urge them to include this text higher up (before Figure 1) – it's not a "discussion point" it's a fundamental component of how they are treating their data prior to the assessment of other correlations, and should be placed in that order.

Line 278 – 1.5°C cooler rather than warmer? Consider combining with previous sentence to make clearer "These..." is slightly ambiguous.

Figure 2a – why doesn't the uncorrected Mg/Ca record extend as far back in time as the corrected one (before 20ka)?

Units on the y-axis

Figure 2 - It would be helpful to have some specification on the sites from which these data are drawn – in the text or in the figure caption, rather than just in methods / SI.

Figure 2 – the mismatch with the Mg/Ca "uncorrected" SST data in this figure is mostly in the interglacials – with the MAT and the corrected Mg/Ca SSTs ~4°C warmer than the "uncorrected" values. Given that the main thrust of the correction presented here is for the glacials (at low SST and low carbonate ion concentrations), I'm presuming these "uncorrected" Mg/Ca SSTs are ones that have been "adjusted" to give a better fit to colder glacial SSTs, but then underestimate glacial – interglacial warming? If so, this would be worth stating clearly - that these Mg/Ca SSTs have already been somewhat adjusted to try and address the issue you are tackling; i.e they are trying to improve the glacial temperature estimates, but then miss the interglacial warming - whereas your method does a better job on both.

Figure 3 – would be useful to have these plots labelled with the two time intervals and as the difference plot – to save going back and forth to the caption.

I think it would be better to have the same temperature scales for the LGM and the Holocene – at the moment it looks like there is no / little change in SST between the two and makes the general "reading" of these plots less intuitive.

Reviewer #3

(Remarks to the Author)

This manuscript by Morley et al. tackles a long-standing problem in paleoceanography: at high latitudes, one of the only available foraminifera species is *N. pachyderma*, yet Mg/Ca derived temperature (a mainstay of paleothermometry) seem not to “work” in this species, especially at the low temperatures where it would be most relevant. Here the authors take a very clever approach to correcting this paleothermometer for variability in carbonate chemistry using oxygen isotopes. The result is a new equation for deriving temperature from paired Mg/Ca and $\delta^{18}\text{O}$ measurements on the shells of *N. pachyderma*. This is then applied to several glacial cycles of data to present Arctic temperature reconstructions which agree more readily with model results. Overall, I think this is a really exciting paper. I have some minor comments below and one major suggestion, which should be taken as just that.

One question that has been raised as the community has attempted to solve the problem of Mg/Ca in *N. pachyderma* is that of the different populations or genotypes. Thus, I think it would substantially strengthen this paper to test the efficacy of the new equation more widely than just in the North Atlantic, and thus expand the utility of this proxy from just the Arctic to high latitudes globally. Paired records from the Pleistocene exist for both the North Pacific and the Antarctic, as do model runs, and polar amplification is likely not localized to the North Atlantic. Including a reanalysis and data-model comparison of some records from other regions would, if successful, increase confidence in the utility of this method. And, if unsuccessful, provide important insights into limitations.

Minor suggestions:

93: Remove either “furthermore” or “also”

174: Consider rephrasing. The paper referenced does not compare crusts in living vs dead foraminifera, but the outer portion of the shell under the assumption that they will include variable amounts of crust in living (less/no crust) vs dead (more crust) foraminifera.

176-178: needs citation or data.

234-236: Ontogenetic growth likely takes weeks or months in this species (e.g., Davis et al., 2020; Westgard et al., 2023), whereas crusting likely occurs over a much shorter period of time at the end of the life cycle (Davis et al., 2017, but also evidenced by the low number of crusted individuals in the water column). Unless there is a large, undescribed discrepancy between precipitation rate and growth rate, that the difference in Mg/Ca between the ontogenetic and crust calcite is rate dependent seems very unlikely.

Figure 1: Please label figure panels. I also don't see the pink crosses described in c.

Figure 2: I'd consider adding a key to this figure rather than the stacked axis titles to make it more readily interpretable. The axes titles also need units.

Version 1:

Reviewer comments:

Reviewer #1

(Remarks to the Author)

Morley et al. satisfactory addressed the comments and concerns I have raised for the initially submitted version of this manuscript. There are just two minor issues that can be easily addressed:

- (1) The authors used both the terms "lamellar calcite" and "ontogenetic calcite". In my understanding, these two terms describe exactly the same portion of shell calcite. Thus, the authors should decide on the terminology they would like to use and not alternate between these two terms.
- (2) The pink color chosen for Fig. 2 is difficult to read (at least on my screen). I strongly suggest to use a color that offers more contrast against a white background.

I have attached an annotated PDF with some very minor comments, mostly grammar. I recommend that the authors also carefully check the final copyedited version.

I am looking forward to seeing this paper published, as it offers a promising solution to improve the fidelity of paleoclimate records in high latitudes - a long standing challenge in paleoclimate.

Reviewer #2

(Remarks to the Author)

The authors have addressed all of my comments and I'd recommend publication.

Reviewer #3

(Remarks to the Author)

This is my second review of the manuscript by Morley et al. and I'm satisfied that most points raised have been addressed.

One minor point remains the assertion on line 249 that ontogenetic calcite is formed more rapidly than gametogenic. Please remove this line. In the author's response to Reviewer 1 on this issue they cite a study (Spero et al., 1988) where initial chamber formation was observed over 1-3 hours. However, calcification actually continues for many days after initial chamber formation. This can be seen in the presence of diurnal banding in *O. universa* (the same species observed in the Spero et al. study) (e.g., Eggins et al., 2015) which suggests days to over a week of calcification in the final "orb" chamber. An even more apt example is *Neogloboquadrina dutertrei*, which also calcifies for several days over a single chamber as evidenced banding in its ontogenetic calcite (Fehrenbacher et al., 2017), prior to crust formation. Thus this assertion runs counter to the currently available evidence and is moreover not terribly relevant to the story here.

Dear Reviewers,

We would like to thank you for helpful comments on our manuscript. They have helped to improve our submission, and we are grateful for the critical assessment of our work. In our response, we have addressed each of your concerns and questions in the following format: Each question or comment is re-stated as in the original review of the manuscript in black 'Calibri font'. Our response to each comment/question is indented and written in blue 'Calibri font'.

REVIEWER COMMENTS

Reinhard Kozdon (RK) (Remarks to the Author):

Review: A solution for constraining past marine Polar Amplification

Submitted to Nature Communications by Morley et al.

Morley et al. submitted a very interesting manuscript to Nature Communications: Since the seminal study of Nuernberg (1995) that is widely regarded as the starting point to measure the Mg/Ca ratio in foraminiferal shells as a temperature proxy, it is known that Mg/Ca ratios in shells of the polar to subpolar foraminifera *N. pachyderma* are "too high" in water temperatures colder than about 4°C. These aberrant Mg/Ca ratios have been somehow puzzling and were confirmed by several follow-up studies. The performance of other proxies that work well in warmer waters such as the $\delta^{18}\text{O}$ composition of foraminifera shells is also limited in the polar to subpolar ocean. Thus, the fidelity of available paleoclimate reconstructions from the northern North Atlantic is relatively poor, even though this region plays an important role in modulating the climate of the northern hemisphere. Therefore, robust paleoclimate data from the polar to subpolar realm are essential and sorely needed to test and improve climate model simulations.

In this study, Morley et al. presents an explanation for the aberrant ('too high') Mg/Ca ratios in *N. pachyderma* from high latitude oceans: elevated Mg/Ca is linked to a low carbonate ion concentration. The authors developed a new approach to correct for the secondary control of the carbonate ion concentration on Mg/Ca with the ultimate goal to obtain much more robust Mg/Ca-based paleotemperatures. Morley et al. furthermore present a technique that allows for the reconstruction of the past carbon ion concentration based on the oxygen isotope ratio measured in foraminiferal shells. Thus, existing records with paired Mg/Ca – $\delta^{18}\text{O}$ analyses can be corrected for the carbonate ion effect on Mg/Ca. The authors apply this approach on a downcore record and found much higher glacial-interglacial temperature swings than inferred from previous (uncorrected) data. This study and the results are timely, important, and transformative. We definitely need better paleoclimate records from high latitude oceans, and this study is a welcomed contribution to solve long-standing challenges. However, in its current form, the manuscript needs moderate to major revisions – I made many comments in the attached PDF (and Excel file) and hope that they are helpful to further improve the manuscript.

We thank Prof Reinhard Kozdon (RK) for his positive evaluation of the significance of our study and for his constructive criticism, which helped us very much to clarify our approach, especially with respect to the ecology of the proxy carrier.

My main concern is centered on the calcification depth: Morley et al. suggests relatively shallow average calcification depths for *N. pachyderma* (of course, the habitat depth changes during the ontogenetic development of the foraminifera, and at the end, we get an average value). Many older

studies (e.g. Stangeew, Jensen, Simstich ... - details are in the annotated PDF) using plankton tows are reporting calcification depths of *N. pachyderma* that may reach well below >150 m in some areas of the Nordic Seas (however, the calcification depth is variable and may be shallower in other areas). Other studies try to assign the calcification depth based on a match between foraminiferal d18O and the depth in the water column where we would expect formation of that shell-d18O value in equilibrium with d18O seawater and ambient temperature. These studies also suggest larger calcification depth (e.g., Simstich et al.). In addition, many *N. pachyderma* studies are reporting a vital effect in d18O that directly affects depth reconstructions.

In a nutshell, I believe that the average calcification depths modeled/calculated by Morley et al., are quite shallow and conflict with some of the well-established literature. I am not suggesting that they are wrong – we have to accept that many aspects of the life cycle of *N. pachyderma* are not well known, and we can also challenge some of the existing literature. However, I would expect a much more detailed discussion about the depth habitat and prior studies.

What about vital **effects in d18O**? What about **the formation of the crust** (relevant for core tops)? What is the **water depth of crust formation**? If the crust contributes more than 50% to the total shell mass (suggested by some studies), what will be the ‘recorded’ water depth of that shell? I believe that a new paragraph discussing past studies assessing the depth habitat of *N. pachyderma* and a justification of the depth habitat suggested by Morley et al. would be very helpful.

We have added two sections on depth habitat for *N. pachyderma* to address this major comment in the revised manuscript. The first is early in the revised manuscript (following ll. 77-78) and the second is in the Methods where we address d18O_c vital effects and crust formation. Here we specifically address the questions raised above by RK.

What about vital **effects in d18O**? What about **the formation of the crust**?

We did not apply a systematic correction for a “vital effect” on $\delta^{18}\text{O}_c$ values because the application of such a vital effect when estimating calcification depth for core top *N. pachyderma* is complicated by; 1.) The $\delta^{18}\text{O}_c$ offset from equilibrium measured in *N. pachyderma* collected from plankton tows is highly variable within a region, across regions, seasons, and on interannual timescales (Stangeew 2001, Volkmann and Mensch 2001, Pados, Spielhagen et al. 2015, Livsey, Kozdon et al. 2020); 2.) The addition of crust, typical for specimens recovered from core tops, increases the $\delta^{18}\text{O}_c$ signal because crust is isotopically heavier than ontogenetic calcite (Kozdon, Ushikubo et al. 2009, Mikis, Hendry et al. 2019). Variations in the degree of crusting can therefore either “lower”, completely “mask”, or shift the vital effect towards positive values depending on the degree of crusting (Kozdon, Ushikubo et al. 2009). As a result, there remains considerable uncertainty in the exact value to use when correcting for the competing signals of vital effects and crust in *N. pachyderma*, with studies reporting both calcification in equilibrium (Jonkers, Brummer et al. 2010, Jonkers, Heuven et al. 2013, Mikis, Hendry et al. 2019, Jonkers, Brummer et al. 2022) and out of equilibrium with seawater (Kohfeld, Fairbanks et al. 1996, Bauch, Carstens et al. 1997, Simstich, Sarnthein et al. 2003, King and Howard 2005).

What is the **water depth of crust formation**?

It has long been assumed that the isotopically heavier crust calcite, results from systematic ontogenic vertical migration to deeper and colder waters associated with reproduction (Emiliani 1954). However, from comprehensive, cross-regional plankton tow studies we know that heavily crusted specimens can be found at any depth including the top 0-50m (Tell, Jonkers et al. 2022) suggesting that there is no systematic ontogenic vertical migration associated with reproduction or “crusting” for *N. pachyderma* (Manno and Pavlov 2014, Tell,

Jonkers et al. 2022). Furthermore, culture-grown crust calcite in *N. incompta* have lower Mg/Ca ratios than ontogenetic calcite formed at the same temperature, suggesting that a lower temperature is not responsible for the low-Mg/Ca ratio of neogloboquadrinid crusts (Davis, Fehrenbacher et al. 2017). Finally, cross-sections of *N. pachyderma* tests from North Atlantic sediment traps also show that tests growing in a well-mixed water column with constant temperatures exhibit lower Mg/Ca ratios in their crust (Jonkers, Buse et al. 2016) suggesting that alongside temperature other factors control crust Mg/Ca.

By not applying a correction for vital effects or the contribution of crust calcite we avoid biasing calcification depth estimates towards deeper depth. This approach is also consistent with sediment trap studies from both the North and South Atlantic where $\delta^{18}\text{O}_c$ values measured on crusted *N. pachyderma* are in equilibrium with measured water column $\delta^{18}\text{O}_c$ (Jonkers, Heuven et al. 2013, Mikis, Hendry et al. 2019).

In addition, I believe that the figures can be improved. At least on my computer, they appeared to be in low-resolution. Again, I made my detailed comments in the attached PDF.

All suggestions to improve figures were made.

In summary, this is a very important paper that requires moderate to major revision before I consider it ready for publication. However, the findings are of great importance, and I believe that this manuscript is -in general- suitable for Nature Communications. I am looking forward to seeing it published.

Reinhard Kozdon (RK)

Editorial suggestions and in-text comments from RK

- I. 21 are references allowed in the abstract for Nature Communications? Please confirm
We consulted the guide for authors and found that references are not allowed. We removed them from the abstract.
- I. 23 SSTs
The acronym SST has been changed to SSTs throughout the manuscript
- I. 24 Please add the following reference, as it summarizes nicely the challenges at the 'cold end' of calibrations: Weinelt et al., 2001. Paleoclimatographic Proxies in the Northern North Atlantic. In: P. Schäfer, W. Ritzrau, M. Schlüter and J. Thiede (Editors), The Northern North Atlantic. Springer Berlin Heidelberg, pp. 319-352.
We have added this reference as suggested
- L. 28 SSTs
changed
please use either 'shell' or 'test'. Test is used more frequently in this manuscript, thus, I recommend to change it to 'test'
we have changed "shells" to "tests" throughout the revised manuscript
- I. 28 okay, but I prefer "polar to subpolar" instead of "Arctic". *N. pachyderma* can also be found outside the Arctic realm
changed as suggested by RK
- I. 29 what exactly is meant by "this proxy system"? The polar regions? $\delta^{18}\text{O}$ - Mg/Ca? Please clarify and reword accordingly.
We have clarified this sentence and referred to Mg/Ca-based paleoclimate reconstructions specifically
suggestion: The fidelity of Mg/Ca - based paleoclimate reconstructions is affected by...
changed as suggested by RK

- I suggest to reword this sentence, as as the correction is not for the variations in seawater carbonate chemistry. It is the correction for the impact of seawater carbonate chemistry on Mg/Ca that needs to be corrected
reworded as suggested by RK
- I. 31 it is not removed from the equations. The study proposes a method to correct for the impact of variations in seawater carbonate chemistry on foraminiferal Mg/Ca
we replaced "removed" with "isolated to address this point
for a reassessment
We reworded the sentence as suggested by RK
- I. 32 response?
changed as suggested by RK
temperatures
changed as suggested by RK
- I. 33 Has only the high latitude cooling been underestimated? What about high latitude warming during past interglacials? Are there also some indications of under or overestimates of SSTs during past 'warm' periods?
Yes, we believe that Mg/Ca-based SSTs derived from NP have underestimated past interglacial warmth, primarily because of the very insensitive calibration equations that were used to derive SSTs (e.g. Irvani, Ninnemann et al. 2016, Irvani, Galaasen et al. 2020) These resulted in muted glacial-interglacial variability. To reflect this point, we reworded this sentence to include warm periods as well.
- I. 37 of the high latitude.
changed as suggested by RK
suggestion: allowing for
changed as suggested by RK
carbon dioxide, to be consistent with line 43
changed as suggested by RK
- I. 42 "changes" are is very unspecific. Do you mean temperature changes?
Yes, we added temperature before changes to clarify
- I. 54 suggestion: challenging
changed as suggested by RK
- I. 56 add comma
added as suggested by RK
SSTs (?)
changed as suggested by RK
- I. 57 please also cite the excellent summary paper of Weinelt et al 2001. Paleooceanographic Proxies in the Northern North Atlantic. In: P. Schäfer, W. Ritzrau, M. Schlüter and J. Thiede (Editors), The Northern North Atlantic. Springer Berlin Heidelberg, pp. 319-352."
We have added this reference as suggested
Suggestion for a transition: Other proxies are affected by their on set of limitations in high latitude oceans. For example, dinoflagellate....
changed as suggested by RK
- I. 59 oxygen
changed as suggested by RK
- I. 60 no comma
changed as suggested by RK
- I. 66 I think it is important to mention that the first Mg/Ca-temperature calibration, that ultimately lead to the 'acceptance' of this temperature proxy within the scientific community, is based on measurements of *N. pachyderma* (Nuernberg 1995).

The calibration of Nuernberg was successful as he excluded shells from the 'cold end' (in his paper, he reported that he excluded shells from sampling sites that are annually covered by sea ice).

We have added two sentences to the revised manuscript to address this comment

- I. 69 Suggestion: Mention that the Mg/Ca temperature calibration is exponential. Thus, the sensitivity is low at cold temperatures, which further contributes to the challenges (besides the effects of the seawater carbonate chemistry on Mg/Ca discussed here) mentioned as suggested by RK
plural?
changed as suggested by RK
- I. 72 Please cite a reference for these temperatures. Any published paper discussing these extremely warm Mg/Ca-based glacial temperatures?
We have added Meland et al. 2006 to address this comment. They show in their results section for Mg/Ca-based SSTs reconstructions of the LGM that *"In the Greenland and Iceland Seas, temperatures approach 6–10°C, higher than the surface temperatures today."*
- I. 73 shouldn't it be plural?
changed as suggested by RK
- I. 74 plural?
changed as suggested by RK
- I. 77 "There are several aspects that should be discussed in more detail:
- (1) The average habitat depth of *N. pachyderma* may be quite variable in the Nordic Seas. Please see older studies using traps (e.g. Stangeew (https://macau.uni-kiel.de/receive/diss_mods_00000464)). 100-150 m may not be unrealistic for some sites. Also Simstich et al., 2003 and references therein is a great source (Simstich, J., Sarnthein, M., Erlenkeuser, H., 2003. Paired d18O signals of *Neogloboquadrina pachyderma* (s) and *Turborotalita quinqueloba* show thermal stratification structure in Nordic Seas. *Mar. Micropaleontol.*, 48: 107-125.
In the revised manuscript we have included three new paragraphs (Methods) that review the habitat range of *N. pachyderma* based on recent ecological observations
 - (2) Fossil shells (even relatively young fossil shells from core tops) feature a thick crust. Although a crust may be found in plankton tow or trap samples, a very thick crust, resembling those from fossil shells, is rarely found on shells collected in the water column. I may be missing some recent literature, but as far as I remember, the formation of the thick crust is still a bit enigmatic. Thus, if the thick crust, that may contribute >50% of the total shell mass (e.g., Stangeew), is formed in deeper waters, the average chemical and isotopic composition of that shell may reflect deeper waters (although the habitat depth was much shallower in earlier life stages). Therefore, I think it is possible that calibrations based on fossil shells are biased towards deeper water masses.
In the revised manuscript (Methods) we have included a discussion on this point specifically, outlining our argument that crust is not indicative of a deeper calcification depth.
 - (3) The comparison of measured d18O values of *N. pachyderma* with water column profiles also support crust formation in deeper waters (Hupp and Fehrenbacher 2023 and references therein)"
Hupp and Fehrenbacher 2023 show that crusted specimens collected from the upper 0-200m are isotopically heavier than uncrusted specimens collected from the same depth interval. Unfortunately, this sampling strategy (e.g. one net from 0-200m) does not allow us to infer ontogenic vertical migration or crust formation at deeper depth which is assumed in the paper but not demonstrated.

- I. 78 there are other studies suggesting greater habitat depth - please see prior comment. This is still an active field of research. See Simstich et al., 2003, Bauch et al., 1997, Stangeev 2001, Hupp and Fehrenbacher, 2023..."
In the revised manuscript (Methods) we have included an extensive section on this point.
- I. 85 plural?
changed as suggested by RK
- I. 89 comma
changed as suggested by RK
- I. 93 plural?
changed as suggested by RK
- I. 94 this sentence is unnecessarily complicated. Please reword
We rephrased this sentence to ease readability as suggested by RK
- I. 99 present tense
changed as suggested by RK
- I. 108 please use either 'shell' or 'test'. Test is used more frequently in this manuscript, thus, I recommend to change it to 'test'
changed as suggested by RK
- I. 134 ...for the closely...
changed as suggested by RK
 suggestion: "more pronounced" instead of "steeper"
changed as suggested by RK
- I. 139 et al. (period after al)
changed as suggested by RK
- I. 140 comma
changed as suggested by RK
- I. 144 Please discuss this in more detail. Fig. 2 in reference [48] (Lombard et al., 2009) suggests that *N. pachyderma* features lowest growth rates at about 12 degree C, whereas at colder temperatures, growth rates are increasing. I would expect that the metabolic/respiration rates are lowest at 12 degree C, not at the 'cold end'
It is important not to mix respiration/metabolic rates with calcification rates. At the extreme warm end of a species' optimal range, it is to be expected that respiration rates are high (e.g. (Lombard, Erez et al. 2009) while growth rates are low (Lombard, Labeyrie et al. 2009) due to the temperature stress. In line 144 though the confusion possibly comes from the fact that we used the wrong reference. We apologize for this oversight and correct the reference in the revised manuscript.
- I. 145 this was also hypothesized by [5]
reference added as suggested by RK
- I. 147 Figure 1 panel b. I don't understand this equation. $Mg/Ca_{[norm]} = Mg/Ca_{[meas]} / Mg/Ca_{[pred]}$.
We have rephrased this section to clarify the equations and Figure 1. The revised text now reads as follows: First, we normalized measured Mg/Ca values against predicted Mg/Ca values (\$Mg/Ca_{[pred]}\$ ) to reveal by how much the measured Mg/Ca is an overprediction of the expected. We then derive a \$Mg/Ca-[CO_3^{2-}]\$ sensitivity for all samples included in this study by comparing \$Mg/Ca_{[Norm]}\$ against measured \$[CO_3^{2-}]\$ (Fig. 1b). \$Mg/Ca_{[Corr]}\$ (Fig. 1c) then reveals the temperature-only component and is calculated by dividing \$Mg/Ca_{[meas]}\$ over the \$Mg/Ca-[CO_3^{2-}]\$ sensitivity derived in Fig 1b.
 What does the equation shown in Panel (b) calculate? n = 235. What has been sampled/measured 235 times?
Here we show all datasets included in our analysis which include previously published core tops from Kozdon et al. 2009, and Meland et al. 2006 (Grey), our Plankton tows and new core top data (Black) and all data shown in Morley et al 2019 for *N incompta* (Turquoise).

The number of data points in this plot is much lower

No that is not the case. It may appear that way but it is the same amount of data as shown in panels a. and c.

The four plots are somehow pixelated when zoomed in. Please compile a higher-res version for publication.

A high-resolution plot will be submitted with the revised manuscript

Figure 1 panel d. thus, there are no O-isotope data from *N. incompta* as this species is excluded from panel (d)? Please mention this in the caption

changed as suggested by RK

Figure 1 panel d. please do not extend the range of the regression outside the range of available data

changed as suggested by RK

I. 149 seems to be turquoise on my screen

changed as suggested by RK

I. 151 reword? yellow shading connotes...

changed as suggested by RK

I. 152 I am not able to see pink crosses

We apologize, the pink crosses referred to a previous version of Figure 1. They have since been removed. We have revised the figure caption accordingly.

I. 153 reword? blue shading connotes...

changed as suggested by RK

I. 154 please reword.

changed as suggested by RK

I. 157 it this relevant, as we discuss here Mg/Ca data from cold waters. *O. universa* and *G. bulloides* prefer warmer waters, and the effect of the carbonate ion concentration on Mg/Ca is less pronounced

Yes, we believe that the addition of these data is relevant, especially since these data are also mentioned/discussed in the text. We therefore prefer to leave the data in Fig. 1.

L. 162 unit is missing!

changed as suggested by RK

L. 164 delete systematically

changed as suggested by RK

I. 167 please mention that this is the sensitivity with respect to the carbonate ion concentration

changed as suggested by RK

I. 168 Please reword. you have developed a new methodology that allows for a significant improvement of the fidelity of climate reconstructions. (We knew before that the climate signal is recorded and preserved, but we were not able to compile robust records from the 'cold end'.)

reworded as suggested by RK

I. 171 in my opinion, this is conflicting with the statement in lines 77-80.

We have reworded the sentence in lines 77-80 and the relevant section in line 171

I. 173 Mg/Ca ratios (add ratios)

changed as suggested by RK

I. 174 Why? Does this mean the Mg/Ca of the crust changes quickly (Mg-removal) after the death of the organisms?

Observations from culture suggest that thickening of test walls or crusting in *N. pachyderma* is always linked to reproduction and specifically to gametogenesis as opposed to asexual reproduction (Westgard et al. 2023, Meilland et al. 2023, Davis et al. 2017). The process of crusting can occur over 24 h and always happens before gametogenesis and death. It is this so-called "gam crust" that results in the 8.6-15% difference in Mg/Ca between living and dead specimens recovered from the same plankton tows (Hupp and Fehrenbacher 2023).

- I. 176 Is it known why dead specimens from core tops feature a thicker crust than dead specimens from tows? Does inorganic precipitation takes place after the shell is deposited on the seafloor? Why doesn't this happen to shells in the trap?
 We do not have a definitive answer to these questions. We cannot exclude that post-mortem processes including the precipitation of inorganic calcite, dissolution, or recrystallization impact the thickness of the crust as the tests sink through the water column and settle at the seafloor. With regards to sediment traps, when deployed in the deep ocean (Jonkers et al. 2010, 2022) authors state that *"in the many years of work on this time series we have never come across a crust-free specimen."* Similarly, empty and strongly encrusted shells have been found at the whole depth range of the productive zone in plankton tows (Tell et al. 2023) suggesting that the formation of crust may be complete in the upper ocean rather than a feature of post-depositional processes. The fact that sediments are devoid of thin-walled specimens may therefore be linked to their low preservation potential. We have revised this section of the manuscript accordingly.
- I. 182 please reword, as this could be understood as the "least dominant" (negligible) control
 reworded as suggested by RK
- I. 185 not perfectly correct. You adjust the Mg/Ca ratio for the effect of the carbonate ion concentration, and then concert the adjusted Mg/Ca to (corrected) temperatures
 reworded to address this point made by RK
- I. 187 on the oxygen...
 changed as suggested by RK
- I. 190 I suggest to cite the seminal work by Epstein/Emiliani
 We added both references Emiliani, C., 1955. Pleistocene temperatures. *The Journal of geology*, 63(6), pp.538-578 and Epstein, S., Buchsbaum, R., Lowenstam, H. and Urey, H.C., 1951. Carbonate-water isotopic temperature scale. *Geological Society of America Bulletin*, 62(4), pp.417-426 to the revised manuscript to address this point
- I. 193 delete "stable"
 changed as suggested by RK
- I. 196 ...past seawater [CO₃²⁻]...
 changed as suggested by RK
 delete "in the past"
 changed as suggested by RK
- I. 200 suggestion: excluded
 changed as suggested by RK
- I. 201 found (to be consistent with line 199)
 changed as suggested by RK
- I. 202 This is somehow conflicting with the statement in lines 209-210. Is there no interaction between CO₃²⁻ and temperature, or is there only an interaction at temperatures above 5 degree C (line 209)?
 We have added a clarifying sentence in the revised manuscript stating that while temperature contributes to the variance in $\delta^{18}\text{O}_c$ above 5C it does not influence the interaction between $\delta^{18}\text{O}_c$ and [CO₃²⁻].
- I. 218 ions (?)
 changed as suggested by RK
- I. 229 suggestion: add the relevant references at the end of the sentence, even if they were mentioned earlier in the text
 we added the relevant references as suggested by RK
- I. 232 add "the"
 changed as suggested by RK
 "sensitivity of d18Oc to varying CO₃²⁻ ... in a low [CO₃²⁻] setting. The way it is written, it is not clear what the ""sensitivity"" refers to"

- changed as suggested by RK
- I. 235 "I am not sure if this is correct. Please verify. I always thought that the crust forms quite fast at the end of the lifecycle. The shell mass is increasing significantly in a short amount of time (associated with gametogenesis in many species, I don't know if the same was observed for pachyderma). Please verify and include a reference"
- To the best of our knowledge, this statement is correct. It is important to differentiate between the total ontogenetic growth of a specimen and the calcification rate of individual chambers ($\mu\text{g/h}$). Westgard et al. 2023 described long periods of dormancy between chamber additions, which does not mean that chamber growth took weeks. Specifically: "*Specimens were observed to go for periods of several days to weeks without adding chambers and/or growing several chambers over a few days*". The exact duration of calcification from the moment the primary organic sheet (Erez 2003, de Nooijer et al. 2014) forms to the retreat of cytoplasm to reveal a new translucent chamber is difficult to determine in culture for *N. pachyderma* as temperature control at low temperatures prevents long periods of observations at a time. However, most observations on chamber calcification in other species show that a thin-walled chamber is produced within 1–3 h (Spero 1988). While these observations were made for a different species (e.g. *Orbulina Universa*), they compare to a much longer pregametogenic period of 1-3 days observed for *N. pachyderma* during which wall-thickening occurs (Davis et al. 2017, Davis et al. 2020).
- I. 236 "Absolutely! However, please mention that this approach (i.e. monitor the degree of encrustation) is also important as the crust is formed in deeper waters, which are typically colder. If the degree of encrustation changes from sample to sample, one sample may be biased towards colder (deeper) waters, and the other sample towards shallower (warmer) waters, thereby potentially attenuating the magnitude of climate swings.
- We have included a comprehensive section on calcification depth in the revised manuscript (Methods) where we outline why we do not attribute the geochemical signature preserved in crust calcite to precipitation at a deeper depth.
- I was able to estimate the degree of encrustation in carefully split the shells, and inspect the cross section with a good binocular microscope. Would the ratio of shell-diameter/shell weight also work? I never tried this"
- We don't believe that this method would result in accurate estimates of crusting, because the shell diameter does not take into account the number of chambers precipitated before gametogenesis. In other words, the ratio between Lamellar calcite and shell diameter is not constant.
- add 1-2 sentences about the approach to select shells with ~similar degree of encrustation
- While we selected specimens from a very narrow size fraction, we did not perform a systematic analysis of the degree of encrustation of *N. pachyderma* in the core top samples. We have added a statement addressing this point in the Method section.
- I. 242 I think it should be mentioned that Kozdon et al. was dissolving shells for paired Mg/Ca and $\delta^{44}\text{Ca}/40\text{Ca}$ measurements. The $\delta^{18}\text{O}$ data are from the same core samples, but from a different set of picked and analyzed foraminifera.
- We have included a statement in the Methods clarifying that specimens for Mg/Ca and $\delta^{18}\text{O}_c$ analysis were picked from the same sample but are not paired i.e. they constitute two separate sets of specimens.
- I. 246 add "the"
- changed as suggested by RK
- I. 247 shouldn't it be 0.81 vs 0.58?
- changed as suggested by RK
- I. 256 ADD ":"
- changed as suggested by RK
- present tense?

- changed as suggested by RK
- L. 259 Are there other references that directly discuss Monte-Carlo simulations in more detail?
The Monte Carlo simulation was specifically designed for this study, so there are no more specific references for it, however, we have expanded the explanation in the main text and the Methods and will attach the fully commented code to the final version of the paper.
how was the uncertainty range determined?
We clarified the text as follows: *“Briefly, to account for dependent and independent uncertainties 1000 potential calibration slopes are created from selecting random environmental values within the 2σ uncertainty ranges for individual parameters ($\delta^{18}\text{O}_c$, $[\text{CO}_3^{2-}]$, Mg/Ca, and Temperature, see Methods) and a calibration found for each distribution.”*
In addition, we have added more information to the methods
- L. 261 delete “the” before 2σ
changed as suggested by RK
- I. 263 delete parenthesis
changed as suggested by RK
- I. 268 SSTs (?)
changed as suggested by RK
- I. 273 suggestion: the magnitude of warming
changed as suggested by RK
- I. 274 suggestion: significantly higher
changed as suggested by RK
- I. 276 Could you please reword this sentence so that it is clear that the temperatures reflect the magnitude of warming and not absolute temperatures? I suggested a small change above.
changed as suggested by RK
Could you please add a reference for MAT (e.g. Pflaumann...)
Reference added as suggested by RK
- I. 278 Word order
changed as suggested by RK
- I. 279 This is not quite clear. Please reword. In my understanding, the reconstructed MAT temperatures for the glacial interval are higher than Mg/Ca-based temperatures due to the low species diversity. However, interglacial MAT temperatures are more in agreement with Mg/Ca-based temperatures. Is this correct? Please reword to add this detail.
We have added this detail as suggested by RK
- I. 280 Figure 2 Suggestion: also label the MIS in the graph
changed as suggested by RK -MIS stages are now in the x-axis for each time series
- I. 284 on my screen, 'pink' looks more like orange
The official name of the chosen colour is “faded Pink”- we have provided higher resolution figures in this resubmission which will hopefully address this point.
- I. 289 no comma
changed as suggested by RK
- I. 290 no comma
changed as suggested by RK
- I. 292 based on corrected N. pachyderma Mg/Ca records
changed as suggested by RK
- I. 293 replace N pachyderma with “this species”
changed as suggested by RK
- I. 295 no comma
changed as suggested by RK
no comma
changed as suggested by RK
- I. 306 please add reference for ODV

reference added as suggested by RK

in addition, *N. pachyderma* reflects sub-surface temperatures, models calculate surface temperatures

We have added a comprehensive section on calcification depth in the revised manuscript and do not believe that SSTs based on *N. pachyderma* exclusively record subsurface temperatures but instead provide an integrated climate signal that is representative of the top 70m of the water column

I. 313 Does this conflict with the statement in lines 267-279? I thought that the main challenge is the reconstruction of glacial temperatures, not the warmer temperatures during interglacials. In the revised manuscript we have clarified the importance of the correction scheme to both glacial and interglacial climates.

I. 326 comma

changed as suggested by RK

I. 331 please use either 'shell' or 'test'. Test is used more frequently in this manuscript, thus, I recommend to change it to 'test'

changed as suggested by RK

I. 342 delete "was"

changed as suggested by RK

I. 347 add semicolon

changed as suggested by RK

add comma

changed as suggested by RK

I. 356 add comma

changed as suggested by RK

L. 369 Most readers know about this correction, however, I prefer that more information is added.

We have clarified that calculations were completed by subtracting 0.27‰ "in order to compare measured $\delta^{18}\text{O}$ values of CO_2 produced by the reaction of calcite with H_3PO_4 and CO_2 equilibrated with water (Friedman and O'Neil 1977)."

Thus, you assume no vital effect for d18O. Okay. However, many studies from the 1990s and 2000s report a vital effect for d18O in *N. pachyderma* (e.g. see Simstich et al., 2003, and references therein).

We have added a paragraph on this point as mentioned above

In addition, there are some indications that *N. pachyderma* may live in marine snow featuring its own microenvironment that is different from seawater.

We acknowledge studies have suggested *N. pachyderma* live in marine snow, and tow observations show that this species is sometimes embedded in marine aggregates. However, there is high variability in marine snow types, leading the foram microenvironment to be dominated by either photosynthesis or organic matter degradation, which results in uncertainty in the signal recorded. Furthermore, the abundance of marine snow is variable in time and proportional to biological productivity (bloom events). In the absence of further constraints, we assume that marine snow habitat does not significantly impact the d18O signal recorded.

With respect to these studies, I would like to see a short discussion why - in your opinion - the assumption that *N. pachyderma* calcifies in equilibrium with ambient temperature and seawater d18O_{sw} is justified.

We added three paragraphs in the revised manuscript justifying why we did not apply a vital effect and why we believe our "shallow" calcification depths are justified.

L. 372 any hypothesis what could cause this?

Anomalously deep calcification depth estimates can result from hydrographic profiles that do not represent the calcification environment (either temperature or $\delta^{18}\text{O}_{\text{sw}}$) of the core top specimen analysed. For example, unrealistically deep calcification habitat estimates below

300m may result from modern hydrographic profiles that are “too warm” for samples that grew before modern global warming occurred. We have added a sentence on this in the revised manuscript.

- I. 373 "Please discuss these values. I believe that they are too shallow for encrusted core tops and are in conflict with other studies (Simstich, Stangeew, Jenses... see <https://oceanrep.geomar.de/id/eprint/45820/>; <https://d-nb.info/972248404/34>; <https://www.sciencedirect.com/science/article/abs/pii/S0377839802001652>
We explained why we believe these shallow estimates are correct in the revised manuscript (please see Methods).
- I. 376 add comma and delete “ foraminifera
changed as suggested by RK
- I. 379 remove period, subscript “3” in HNO₃”
changed as suggested by RK
- I. 380 reduced compared to which other cleaning protocol? Please provide reference
We have rephrased this sentence to clarify the cleaning procedure employed here.
- I. 385 remove period
changed as suggested by RK
- I. 388 how?
We clarified that clays were removed using multiple Milli-Q rinses.
- I. 393 Why is the cleaning procedure written so detailed for the samples analyzed at Southampton, but very brief for the samples analyzed in Bremerhaven?
We have shortened the cleaning procedure for samples measured in Southampton for the sake of consistency.
- I. 399 Belemnite
changed as suggested by RK
- I. 400 replace “lower” with “better”
changed as suggested by RK
- I. 403 what is Riedal? I have never heard of it. Please add some details
Riedal is an internal working synthetic carbonate standard used in daily operations at FARLAB, University of Bergen – we have clarified this in the manuscript.
replace “lower” with “better”
changed as suggested by RK
- I. 407 add comma
changed as suggested by RK
- I. 419 add “concentration”
changed as suggested by RK
- I. 424 suggestion: to obtain
changed as suggested by RK
- I. 425 not quite clear. What previous relationships?
We have clarified this by referring back to the relationships in figures 1a and b.
- L. 426 acquired during the cruise from
We revised this sentence to clarify
Suggestion. Start new sentence. ...errors related to uncertainties with respect to the estimated habitat depth of the foraminifera...
changed as suggested by RK
- L. 428 10 degree C is a significant deviation. Why? Please explain in a few words
Nine is the maximum value out of the whole data set (as defined by the difference between max and min temperature at 2SD), but it is not commonplace. Such rare values occur where there is a high thermal gradient in the water column at the calibration point, therefore, a small change in depth estimate gives a large difference in temperature. We used 10 to round up and be even more conservative. We choose to keep data such as this in the calibration to reflect

- the true state of affairs, which may include significant uncertainty on individual data points (as shown by the large envelope in Fig. 1c).
present tense
changed as suggested by RK
- L. 429 replace “an” with “the”
the N. incompta data...
changed as suggested by RK
- I. 430 add comma
changed as suggested by RK
suggestion: ...the steps to calculate the uncertainties are identical to those...
changed as suggested by RK
- I. 431 I believe this sentence is not correct - please reword - e.g., ...we assigned a temperature uncertainty of...
changed as suggested by RK
- I. 438 suggestion: are associated with...
changed as suggested by RK
- I. 652 I believe this is not the final competing interest statement
Yes, that is correct. We have revised the competing interest statement in the resubmission
-

Reviewer #2 (Remarks to the Author):

A solution for constraining past marine Polar Amplification - Morley, et al. 2024

This study is an important contribution to improving Arctic and sub-Arctic Pleistocene sea surface temperature (SST) estimation. It provides extensive new geochemical data (paired Mg/Ca - oxygen isotopes), from plankton samples, of the dominant modern and Pleistocene planktic foraminifera in these regions. Pairing this geochemistry data with seawater chemistry, the study improves the understanding of, and correction for, carbonate ion effects on both the Mg/Ca and oxygen isotope composition of foraminiferal calcite. The study then uses these new empirical correction factors to review published paired Mg/Ca - oxygen isotope datasets for a number of glacial - interglacial transitions, and demonstrates a larger magnitude of SST change, which is more in line with climate model and foraminiferal modern analogue techniques. I would recommend publication with minor revisions - the points to consider are listed below.

We thank Reviewer 2 for their positive evaluation of our manuscript and the constructive feedback provided.

Fig 1b - isn't this normalised Mg/Ca[norm] dimensionless? A ratio of Mg/Ca / Mg/Ca? The values of this are not “Mg/Ca” values, but how much the measured Mg/Ca is an overprediction of expected – the scale is 1x, 2x, 3x – not a scale in “mmol.mol⁻¹”. This then makes sense of panel 1c, which is back in mmol.mol⁻¹ – as it's the measured value divided by the correction factor.

The reviewer is correct. We have corrected the axis caption accordingly

Figure 1c – I can't see any pink crosses for the uncorrected N. pachyderma.

We apologize, the pink crosses referred to a previous version of Figure 1. They have since been removed. We have revised the figure caption accordingly.

- What are the two “uncertainty” metrics on this plot? – not clear what the “95% CIs of “fully propagated uncertainties” is from / refers to. Please clarify.

This is explained in the Methods, which was an oversight on our part. It is now also expanded on in the figure caption.

Lines 173 – 178 – if this crust v. ontogenetic calcite Mg/Ca is applied prior to the plotting of the data in figure 1, then I would urge them to include this text higher up (before Figure 1) – it's not a “discussion point” it's a fundamental component of how they are treating their data prior to the assessment of other correlations, and should be placed in that order.

We have moved the figure down a paragraph as suggested by the reviewer

Line 278 – 1.5°C cooler rather than warmer? Consider combining with previous sentence to make clearer “These...” is slightly ambiguous.

changed as suggested by R2

Figure 2a – why doesn't the uncorrected Mg/Ca record extend as far back in time as the corrected one (before 20ka)?

We have corrected this mistake in the revised manuscript

Units on the y-axis

We have added units (e.g. °C) on the y-axis

Figure 2 - It would be helpful to have some specification on the sites from which these data are drawn – in the text or in the figure caption, rather than just in methods / SI.

We have added this information in the figure caption as suggested by R2

Figure 2 – the mismatch with the Mg/Ca “uncorrected” SST data in this figure is mostly in the interglacials – with the MAT and the corrected Mg/Ca SSTs ~4°C warmer than the “uncorrected” values. Given that the main thrust of the correction presented here is for the glacials (at low SST and low carbonate ion concentrations), I'm presuming these “uncorrected” Mg/Ca SSTs are ones that have been “adjusted” to give a better fit to colder glacial SSTs, but then underestimate glacial – interglacial warming? If so, this would be worth stating clearly - that these Mg/Ca SSTs have already been somewhat adjusted to try and address the issue you are tackling; i.e they are trying to improve the glacial temperature estimates, but then miss the interglacial warming - whereas your method does a better job on both.

We agree with R2 and have revised the manuscript accordingly also echoing comments by RK

Figure 3 – would be useful to have these plots labelled with the two time intervals and as the difference plot – to save going back and forth to the caption.

We have labelled each panel as suggested by R2

I think it would be better to have the same temperature scales for the LGM and the Holocene – at the moment it looks like there is no / little change in SST between the two and makes the general “reading” of these plots less intuitive.

We have revised the figure as suggested by R2

Reviewer #3 (Remarks to the Author):

This manuscript by Morley et al. tackles a long-standing problem in paleoceanography: at high latitudes, one of the only available foraminifera species is *N. pachyderma*, yet Mg/Ca derived temperature (a mainstay of paleothermometry) seem not to “work” in this species, especially at the low temperatures where it would be most relevant. Here the authors take a very clever approach to correcting this paleothermometer for variability in carbonate chemistry using oxygen isotopes. The result is a new equation for deriving temperature from paired Mg/Ca and $\delta^{18}O$ measurements on the shells of *N. pachyderma*. This is then applied to several glacial cycles of data to present Arctic temperature reconstructions which agree more readily with model results. Overall, I think this is a really exciting paper. I have some minor comments below and one major suggestion, which should be taken as just that.

One question that has been raised as the community has attempted to solve the problem of Mg/Ca in *N. pachyderma* is that of the different populations or genotypes. Thus, I think it would substantially strengthen this paper to test the efficacy of the new equation more widely than just in the North Atlantic, and thus expand the utility of this proxy from just the Arctic to high latitudes globally. Paired records from the Pleistocene exist for both the North Pacific and the Antarctic, as do model runs, and polar amplification is likely not localized to the North Atlantic. Including a reanalysis and data-model comparison of some records from other regions would, if successful, increase

confidence in the utility of this method. And, if unsuccessful, provide important insights into limitations.

We thank Reviewer 3 for their positive feedback and for the suggestion to apply the correction scheme developed for the North Atlantic to different regions/morphotypes. In the revised manuscript we have extended the application of the correction scheme to the Southern Hemisphere by including two previously published calibration datasets on *N. Pachyderma*.

Specifically, to evaluate the suitability of the proposed equations to other genotypes of *N pachyderma* we applied the correction scheme to the comprehensive core top dataset published in Vazquez Riveiros et al. (2016) that includes 38 core top samples from the subpolar South Atlantic to the Antarctic continental shelf (red upside down triangles) and the sediment trap series from the Antarctic Peninsula (yellow upside down triangles) published in Hendry et al. (2009). These datasets are dominated by genotype IV, which is the only *N pachyderma* genotype adapted to overwintering in sea ice (Darling, Kucera et al. 2004). Nineteen of the 38 core tops sites as well as the sediment trap series are situated on the Antarctic margin and are therefore subject to the possible influence of sea ice on Mg/Ca ratios (Hendry, Rickaby et al. 2009, Vázquez Riveiros, Govin et al. 2016). When

corrected for $\delta^{18}O_c$ – derived $[CO_3^{2-}]$ correlation statistics for the South Atlantic core top dataset are much more robust (e.g. $r^2=0.87$ vs. $r^2=0.30$, $n=36$, see also data plotted in Fig 1a and 1c) and within the uncertainties of the North Atlantic derived dataset. Similarly, the sediment traps from Hendry et al. (Hendry, Rickaby et al. 2009) agree well with the North Atlantic dataset (Fig. 1a and 1c). These applications therefore suggest that the proposed correction scheme can be applied globally and that genotypes do not influence the geochemical climate signature recorded in *N. pachyderma* specimens as previously hypothesised (Darling, Kucera et al. 2004).

In the revised manuscript we also specified in the methods that samples from the South Atlantic sediment trap series did not exhibit a secondary calcite crust and were “glassy” in appearance. In fact, Hendry et al. (2009) state that crusted specimens were specifically excluded from the analysis to avoid the contribution of secondary or gametogenic crust calcite (Hendry, Rickaby et al. 2009). We therefore applied the vital effect as derived in Hendry et al. (2009) when estimating $[CO_3^{2-}]$ values and lowered Mg/Ca ratios by 15% to account for the difference in Mg between ontogenetic and crust calcite as for the plankton tows from the North Atlantic (Jonkers, Buse et al. 2016, Hupp and Fehrenbacher 2023). No vital effect or correction on Mg/ca values was applied to the core top dataset from the South Atlantic (Vázquez Riveiros, Govin et al. 2016) consistent with the processing of North Atlantic core top datasets analysed here.

R3 also suggested a global application of the application scheme to Pleistocene climate records including data-model comparisons, which we agree are needed but may distract from the main focus

of the current manuscript which is centred around the first description of our proposed correction scheme. We hope that the inclusion of the South Atlantic calibration datasets in the revised manuscript demonstrates that the correction scheme can indeed be applied globally.

Minor suggestions:

93: Remove either “furthermore” or “also”

We have deleted “also” as suggested by R3

174: Consider rephrasing. The paper referenced does not compare crusts in living vs dead foraminifera, but the outer portion of the shell under the assumption that they will include variable amounts of crust in living (less/no crust) vs dead (more crust) foraminifera.

We have revised this sentence as suggested by R3

176-178: needs citation or data.

We have rephrased this sentence and added (Jonkers, Buse et al. 2016) next to (Hupp and Fehrenbacher 2023) to support this statement.

234-236: Ontogenetic growth likely takes weeks or months in this species (e.g., Davis et al., 2020; Westgard et al., 2023), whereas crusting likely occurs over a much shorter period of time at the end of the life cycle (Davis et al., 2017, but also evidenced by the low number of crusted individuals in the water column). Unless there is a large, undescribed discrepancy between precipitation rate and growth rate, that the difference in Mg/Ca between the ontogenetic and crust calcite is rate dependent seems very unlikely.

Yes, it is important here to differentiate between the total ontogenetic growth of a specimen and the calcification rate ($\mu\text{g}/\text{h}$). We would like to refer the reviewer to the response given to RK to comment in l. 235.

Figure 1: Please label figure panels. I also don't see the pink crosses described in c.

We apologize, the pink crosses referred to a previous version of Figure 1. They have since been removed. We have revised the figure caption accordingly. Panels in Figure 1 are labelled from a. to d.

Figure 2: I'd consider adding a key to this figure rather than the stacked axis titles to make it more readily interpretable. The axes titles also need units.

We have revised the axis titles as suggested by R3 and added units to the y-axis

References used in this Response

Bauch, D., et al. (1997). "Oxygen isotope composition of living *Neogloboquadrina pachyderma* (sin.) in the Arctic Ocean." Earth and Planetary Science Letters **146**(1-2): 47-58.

Darling, K. F., et al. (2004). "Molecular evidence links cryptic diversification in polar planktonic protists to Quaternary climate dynamics." Proceedings of the National Academy of Sciences **101**(20): 7657-7662.

Davis, C. V., et al. (2017). "Relationships Between Temperature, pH, and Crusting on Mg/Ca Ratios in Laboratory-Grown *Neogloboquadrina* Foraminifera." Paleoceanography **32**(11): 1137-1152.

Davis, C. V., et al. (2020). "Extensive morphological variability in asexually produced planktic foraminifera." Science advances **6**(28): eabb8930.

De Nooijer, L. d., et al. (2014). "Biomineralization in perforate foraminifera." Earth-Science Reviews **135**: 48-58.

Emiliani, C. (1954). "Depth habitats of some species of pelagic foraminifera as indicated by oxygen isotope ratios." American Journal of Science **252**(3): 149-158.

Erez, J. (2003). "The source of ions for biomineralization in foraminifera and their implications for paleoceanographic proxies." Reviews in mineralogy and geochemistry **54**(1): 115-149.

Friedman, I. and J. R. O'Neil (1977). Data of geochemistry: Compilation of stable isotope fractionation factors of geochemical interest, US Government Printing Office.

Greco, M., et al. (2019). "Depth habitat of the planktonic foraminifera *Neogloboquadrina pachyderma* in the northern high latitudes explained by sea-ice and chlorophyll concentrations." Biogeosciences **16**(17): 3425-3437.

Hendry, K. R., et al. (2009). "Controls on stable isotope and trace metal uptake in *Neogloboquadrina pachyderma* (sinistral) from an Antarctic sea-ice environment." Earth and Planetary Science Letters **278**(1): 67-77.

Hupp, B. N. and J. S. Fehrenbacher (2023). "Geochemical differences between alive, uncrusted and dead, crusted shells of *Neogloboquadrina pachyderma*: Implications for paleoreconstruction." Paleoceanography and Paleoclimatology **38**(10): e2023PA004638.

Irvali, N., et al. (2020). "A low climate threshold for south Greenland Ice Sheet demise during the Late Pleistocene." Proceedings of the National Academy of Sciences **117**(1): 190-195.

Irvali, N., et al. (2016). "Evidence for regional cooling, frontal advances, and East Greenland Ice Sheet changes during the demise of the last interglacial." Quaternary Science Reviews **150**: 184-199.

Jonkers, L., et al. (2022). "Variability in *Neogloboquadrina pachyderma* stable isotope ratios from isothermal conditions: implications for individual foraminifera analysis." Climate of the Past **18**(1): 89-101.

Jonkers, L., et al. (2010). "Seasonal stratification, shell flux, and oxygen isotope dynamics of left-coiling *N. pachyderma* and *T. quinqueloba* in the western subpolar North Atlantic." Paleoceanography **25**(2).

Jonkers, L., et al. (2016). "Chamber formation leads to Mg/Ca banding in the planktonic foraminifer *Neogloboquadrina pachyderma*." Earth and Planetary Science Letters **451**: 177-184.

Jonkers, L., et al. (2013). "Seasonal patterns of shell flux, $\delta^{18}\text{O}$ and $\delta^{13}\text{C}$ of small and large *N. pachyderma* (s) and *G. bulloides* in the subpolar North Atlantic." Paleoceanography **28**(1): 164-174.

King, A. L. and W. R. Howard (2005). " $\delta^{18}\text{O}$ seasonality of planktonic foraminifera from Southern Ocean sediment traps: Latitudinal gradients and implications for paleoclimate reconstructions." Marine Micropaleontology **56**(1-2): 1-24.

Kohfeld, K. E., et al. (1996). "Neogloboquadrina pachyderma (sinistral coiling) as paleoceanographic tracers in polar oceans: Evidence from Northeast Water Polynya plankton tows, sediment traps, and surface sediments." Paleoceanography **11**(6): 679-699.

Kozdon, R., et al. (2009). "Intratest oxygen isotope variability in the planktonic foraminifer *N. pachyderma*: Real vs. apparent vital effects by ion microprobe." Chemical Geology **258**(3-4): 327-337.

Livsey, C. M., et al. (2020). "High-resolution Mg/Ca and $\delta^{18}\text{O}$ patterns in modern Neogloboquadrina pachyderma from the Fram Strait and Irminger Sea." Paleoceanography and Paleoclimatology **35**(9): e2020PA003969.

Lombard, F., et al. (2009). "Temperature effect on respiration and photosynthesis of the symbiont-bearing planktonic foraminifera *Globigerinoides ruber*, *Orbulina universa*, and *Globigerinella siphonifera*." Limnology and Oceanography **54**(1): 210-218.

Lombard, F., et al. (2009). "Modelling the temperature dependent growth rates of planktic foraminifera." Marine Micropaleontology **70**(1): 1-7.

Manno, C. and A. Pavlov (2014). "Living planktonic foraminifera in the Fram Strait (Arctic): absence of diel vertical migration during the midnight sun." Hydrobiologia **721**: 285-295.

Mikis, A., et al. (2019). "Temporal variability in foraminiferal morphology and geochemistry at the West Antarctic Peninsula: a sediment trap study." Biogeosciences **16**(16): 3267-3282.

Pados, T., et al. (2015). "Oxygen and carbon isotope composition of modern planktic foraminifera and near-surface waters in the Fram Strait (Arctic Ocean)—a case study." Biogeosciences **12**(6): 1733-1752.

Simstich, J., et al. (2003). "Paired $\delta^{18}\text{O}$ signals of *Neogloboquadrina pachyderma* (s) and *Turborotalita quinqueloba* show thermal stratification structure in Nordic Seas." Marine Micropaleontology **48**(1): 107-125.

Spero, H. J. (1988). "Ultrastructural examination of chamber morphogenesis and biomineralization in the planktonic foraminifer *Orbulina universa*." Marine Biology **99**: 9-20.

Stangeew, E. (2001). Distribution and Isotopic Composition of Living Planktonic Foraminifera *N. pachyderma* (sinistral) and *T. quinqueloba* in the High Latitude North Atlantic, Christian-Albrechts Universität Kiel.

Tell, F., et al. (2022). "Upper-ocean flux of biogenic calcite produced by the Arctic planktonic foraminifera *Neogloboquadrina pachyderma*." *Biogeosciences* **19**(20): 4903-4927.

Vázquez Riveiros, N., et al. (2016). "Mg/Ca thermometry in planktic foraminifera: Improving paleotemperature estimations for *G. bulloides* and *N. pachyderma* left." *Geochemistry, Geophysics, Geosystems* **17**(4): 1249-1264.

Volkman, R. and M. Mensch (2001). "Stable isotope composition ($\delta^{18}\text{O}$, $\delta^{13}\text{C}$) of living planktic foraminifers in the outer Laptev Sea and the Fram Strait." *Marine Micropaleontology* **42**(3-4): 163-188.

Westgård, A., et al. (2023). "Large-scale culturing of *Neogloboquadrina pachyderma*, its growth in, and tolerance of, variable environmental conditions." *Journal of Plankton Research* **45**(5): 732-745.

RESPONSE TO THE REVIEWERS' COMMENTS

Reviewer #1 (Remarks to the Author):

Morley et al. satisfactorily addressed the comments and concerns I have raised for the initially submitted version of this manuscript. There are just two minor issues that can be easily addressed: (1) The authors used both the terms "lamellar calcite" and "ontogenetic calcite". In my understanding, these two terms describe exactly the same portion of shell calcite. Thus, the authors should decide on the terminology they would like to use and not alternate between these two terms.

All changed as suggested to "ontogenetic calcite"

(2) The pink color chosen for Fig. 2 is difficult to read (at least on my screen). I strongly suggest to use a color that offers more contrast against a white background.

We have changed the pink colour to orange which is much more visible. The figure caption has been revised accordingly.

I have attached an annotated PDF with some very minor comments, mostly grammar. I recommend that the authors also carefully check the final copyedited version.

All 34 editorial comments made in the annotated pdf file were changed as suggested. We have also tried our best to carefully check for grammar and other issues before this final resubmission.

I am looking forward to seeing this paper published, as it offers a promising solution to improve the fidelity of paleoclimate records in high latitudes - a long-standing challenge in paleoclimate.

Thank you!

Reviewer #2 (Remarks to the Author):

The authors have addressed all of my comments and I'd recommend publication.

Thank you!

Reviewer #3 (Remarks to the Author):

This is my second review of the manuscript by Morley et al. and I'm satisfied that most points raised have been addressed.

One minor point remains the assertion on line 249 that ontogenetic calcite is formed more rapidly than gametogenic. **Please remove this line.** In the author's response to Reviewer 1 on this issue they cite a study (Spero et al., 1988) where initial chamber formation was observed over 1-3 hours. However, calcification actually continues for many days after initial chamber formation. This can be seen in the presence of diurnal banding in *O. universa* (the same species observed in the Spero et al. study) (e.g., Eggins et al., 2015) which suggests days to over a week of calcification in the final "orb" chamber. An even more apt example is *Neogloboquadrina dutertrei*, which also calcifies for several days over a single chamber as evidenced by banding in its ontogenetic calcite (Fehrenbacher et al., 2017), prior to crust formation. Thus this assertion runs counter to the currently available evidence and is moreover not terribly relevant to the story here.

We have deleted the line as suggested by Reviewer 3.

A solution for constraining past marine Polar Amplification

Morley, A.^{1,2}, de la Vega, E.¹, Raitzsch, M.³, Bijma, J.⁴, Ninnemann, U.⁵, Foster, G.L.⁶, Chalk, T.B.⁷,
Meilland, J.⁸, Cave, R.R.⁹, Büscher, J. V.^{9,10}, and Kucera, M.⁸.

¹ University of Galway, School of Geography, Archaeology and Irish Studies, H91TK33 Galway, Ireland.

² ICIRAG – Irish Centre for Research in Applied Geosciences, Belfield, Dublin 4, Ireland.

³ Dettmer Group KG, 28195 Bremen, Germany

⁴ Alfred-Wegener-Institut, Helmholtz-Zentrum für Polar- und Meeresforschung, Am Handelshafen 12, 27570, Bremerhaven,
Germany

⁵ University of Bergen, Department of Earth Science and Bjerknes Centre for Climate Research, 5007, Bergen, Norway

⁶ University of Southampton, School of Ocean and Earth Science, University of Southampton, National Oceanography Centre
Southampton, Southampton, SO14 3ZH, UK

⁷ Centre Européen de Recherche et d'enseignement des géosciences de l'environnement (CEREGE), Aix-en-Provence, France

⁸ MARUM – Center for Marine Environmental Sciences, University of Bremen, Bremen, Germany

⁹ University of Galway, School of Natural Sciences, H91TK33 Galway, Ireland

¹⁰ Ulster University, School of Geography and Environmental Sciences, BT52 1SA, Coleraine, UK

**Abstract:** Observation-based reconstructions of global sea surface temperatures in response to
changing climate boundary conditions are critical to constrain climate sensitivity and evaluate the
uncertainties of model simulations [1]. On long and pre-instrumental timescales, this is only possible
by employing climate proxies. Yet, most proxies of essential climate variables, such as sea surface
temperatures (SST), suffer from limitations when applied to cold temperatures that characterize Arctic
environments [2-5]. These limitations prevent us from constraining uncertainties for some of the most
sensitive climate tipping points that can trigger rapid and dramatic global climate change such as
Arctic/Polar Amplification [6], the disruption of AMOC, sea ice loss [7], and permafrost melting [8] that
are intrinsic to the polar regions. Here, we present a new approach to reconstructing sea surface
temperatures (SST) using paired Mg/Ca - $\delta^{18}\text{O}_c$ recorded in shells of the Arctic planktonic foraminifera
*Neogloboquadrina pachyderma*. We show that in this proxy system, the Mg/Ca – palaeothermometry
is affected by variations in seawater carbonate chemistry, which can be successfully quantified and
removed from paleotemperature reconstructions allowing reassessment of the absolute
temperature and the magnitude of marine polar amplification to climate forcing on glacial-interglacial
timescales. By applying this novel approach to existing records, we show that the magnitude of high
latitude SST cooling during glacial periods has been underestimated and that the new estimate of SST
change between the Late Holocene and the LGM exceeds model-based estimates [9] of marine polar
amplification by up to $3.0 \pm 1.0^\circ\text{C}$. Our findings open up opportunities to better constrain the oceanic

carbonate system enabling a quantification of high-latitude ocean-atmosphere carbon exchange as
well as to benchmark the performance of CMIP6 and future generations of climate models [1].

One of the key drivers of Arctic climate and Arctic Amplification is the ocean [10, 11] because of its
role in transporting heat to high latitudes, which influences not only marine temperatures but also
terrestrial climates and sea ice extent [12, 13]. As a result, changes to the high-latitude oceans have
global implications, impacting planetary albedo, atmosphere-ocean carbon dioxide exchange,
atmospheric and ocean circulation, deep-water formation, freshwater budgets, productivity, and
biological diversity [14-17]. However, large uncertainties remain about the sensitivity of Arctic climate
to natural [18] and anthropogenic forcing [19] owing to the limited observational and historical
network in this area [20] and uncertainties linked to the strength of cloud feedback in model
simulations [21]. In the absence of a direct proxy for clouds, large scale surface temperature patterns
have been shown to be useful analogues [22] and critical for constraining equilibrium climate
sensitivity in climate models [21]. Efforts to assess the magnitude of past polar surface temperature
patterns using climate proxies are thus crucial to further our understanding of how the Arctic will
respond to and influence continued global warming [13, 23]. Ice-core records provide accurate and
high-resolution surface air temperatures over high-altitude central Greenland for the past 120 ka,
however the reconstruction of SSTs from high-latitude oceans has been difficult. This is because,
planktonic foraminifera species diversity collapses to one dominant species (*N. pachyderma*) at cold
SST (< 4°) leading to large uncertainties associated with reconstructed SSTs based on assemblage
counts [2]. Furthermore, dinoflagellate cyst assemblages at high latitudes are controlled by multiple
variables and it is often difficult to ascertain which is responsible for the observed assemblage change
[24]. In addition, standard geochemical proxies such as Oxygen Isotopes ($\delta^{18}\text{O}$), Mg/Ca-thermometry
and Alkenone-based SST reconstructions cannot accurately constrain cold temperatures <4°C, because
secondary controls besides temperature influence all these proxies at low temperatures [3, 4, 25].

The Mg/Ca ratio of the CaCO_3 tests of foraminifera is an established and powerful tool for the
reconstruction of past SST [26]. It is based on the principle that the partition of magnesium in calcite
is endothermic and therefore leads to higher Mg/Ca values at higher temperatures [27]. Unlike most
foraminifera species in low and mid latitudes, the Mg content in *N. pachyderma* appears disconnected
from temperature, making it difficult to use the otherwise powerful Mg/Ca proxy in the polar realm.

Several attempts have been made to calibrate the Mg/Ca – thermometer for *N. pachyderma* [4, 5, 28-
31], however, to date calibration efforts are not able to resolve SST below 4°C. This difficulty arises
because it appears secondary controls have a dominant influence on Mg/Ca at low temperatures and

these lead to unrealistically high temperatures at high latitude sites causing reconstructed glacial
maxima SST at polar and subpolar latitudes to be too warm (e.g., 6-7°C). Importantly, these warm
glacial maximum SST are at odds with paleoclimate simulations, causing significant data-model
mismatch [9]. To resolve this problem, calibration equations for *N. pachyderma* have been modified
to ever more insensitive calibration equations [5, 32, 33], reducing the actual temperature estimates
and also the amplitude of glacial-interglacial change. Subsequently, these low-SST and low-amplitude
records have been interpreted as 'winter' or subsurface (below 100-150m) temperature signals, an
interpretation that improves data-model agreement but one that is not based on the observed habitat
depth of the main seasonal population [34] or on observations of seasonal flux data of *N. pachyderma*
[35].

The departure from the predicted Mg/Ca-temperature relationship at low temperatures also led to
the hypothesis that, in addition to temperature, seawater carbonate ion concentration ($[CO_3^{2-}]$) may
exert a strong secondary control on Mg/Ca especially at lower concentrations often associated with
cold SST [4, 29]. Culture experiments where planktonic foraminifera grew under controlled conditions
[36-38] and core-top calibration studies, for which modern hydrographic parameters are available,
support this hypothesis, especially when $[CO_3^{2-}]$ values are $<200\mu\text{mol.kg}^{-1}$ [39, 40], which is common
at polar and Arctic latitudes. To account and correct for a possible $[CO_3^{2-}]$ influence on Mg/Ca-
thermometry  species-specific sensitivity must be fully determined [39, 40]. Once the sensitivity is
determined, mathematical correction schemes can isolate temperature from $[CO_3^{2-}]$ within the Mg/Ca
ratios [40-42]. The resulting tool would allow for accurate SST reconstructions at high latitudes and
thereby provide the means to quantify polar amplification on multiple timescales under various
boundary conditions. Furthermore, accurate estimates of past SST are also essential to constrain the
oceanic carbonate system via boron isotope ($\delta^{11}\text{B}$) geochemistry because of the temperature
sensitivity of the dissociation constants of carbonic acid and boron species, and the solubility constant
of CO_2 [43], currently leaving a gap in our understanding of high-latitude ocean-atmosphere carbon
fluxes.

Here, we defined the Mg/Ca - $[CO_3^{2-}]$ sensitivity for *N. pachyderma* for the first time by measuring
100 Mg/Ca ratios on specimens collected live (as determined by the presence of active cytoplasm) from
101 plankton tows from the Labrador and Nordic Seas that can be paired directly with hydrographic
conditions of the calcification environment (Extended Data Figure 1, Extended Data Tables 1-3).
Plankton tows via multinet sampling allow for an accurate analysis of the hydrographic conditions of
the calcification environment when assuming that the plankton tow depth is representative of the

mean calcification depth of living specimens (for details on hydrographic data used see Extended Data
Tables 1-3). This approach allows us to circumvent the fundamental limitations and assumptions
stemming from the traditional core top approach to proxy establishment e.g., our analysis does not
rely on assumptions that are required to link signals found in core-top sediments to hydrographic
conditions during their formation. This is particularly important for Polar and Arctic Oceans considering
the scale of recent warming and low sedimentation/accumulation rates that result in mixing of
preindustrial with modern samples in the top 0.5cm of sediment.

Like previous core top calibrations for *N. pachyderma*, our plankton tow-based results show a
departure from the expected Mg/Ca-temperature relationship (Fig. 1a). To estimate the influence of
seawater $[\text{CO}_3^{2-}]$ on Mg/Ca values, we isolate the contribution of $[\text{CO}_3^{2-}]$ on measured Mg/Ca values
using water-column data for temperature and carbonate-ion concentration, following a similar
approach as described by Evans et al. [36] and Morley *et al.* [40]. First, we normalized measured Mg/Ca
values against predicted Mg/Ca values ($\text{Mg/Ca}_{\text{pred}}$) to derive a $\text{Mg/Ca}-[\text{CO}_3^{2-}]$ sensitivity for all
samples included in this study. $\text{Mg/Ca}_{\text{pred}}$ represents the Mg/Ca sensitive to temperature only, and is
determined as $\text{Mg/Ca}_{\text{pred}} = 0.4 \exp^{(0.09T)}$. This equation was derived for *Neogloboquadrina incompta*
calcifying at $[\text{CO}_3^{2-}]$ values $< 200 \mu\text{mol.kg}^{-1}$ in Morley et al. [40] Given that *N. incompta* and *N*
*pachyderma* are genetical sister species [44] and share ecological preferences [45] we assume that
the sensitivity to temperature is shared between both species. We then normalise $\text{Mg/Ca}_{\text{meas}}$ to
reveal any non-thermal effects, by dividing it by the temperature component such that: $\text{Mg/Ca}_{\text{norm}} =$
$\text{Mg/Ca}_{\text{meas}} / \text{Mg/Ca}_{\text{pred}}$ (Fig. 1b). $\text{Mg/Ca}_{\text{corr}}$ then reveals the temperature only component and is
defined by $\text{Mg/Ca}_{\text{corr}} = \text{Mg/Ca}_{\text{meas}} / \text{Mg/Ca}_{\text{norm}}$. We further complement the tow dataset with new
core tops and paired hydrographic profiles including temperature, salinity, Dissolved Inorganic Carbon
(DIC) and Alkalinity (ALK) from the Nordic Seas, and also include the published Kozdon *et al.* [5] core
top dataset in our analysis using depth estimate from foraminifera $\delta^{18}\text{O}_c$ projected on equilibrium
$\delta^{18}\text{O}_c$ (see methods). The comparisons of plankton tows with core tops allows us to test if the tow
derived $\text{Mg/Ca}-[\text{CO}_3^{2-}]$ sensitivity can be applied to climate archives.

The $\text{Mg/Ca}-[\text{CO}_3^{2-}]$ sensitivity for *N. pachyderma* from plankton tows (Fig. 1b) is consistent with
calibrations for closely related planktonic foraminifera *Neogloboquadrina incompta* [40], but steeper
than in temperate planktonic foraminifera [36]. This can be explained by a non-linear response of
$[\text{CO}_3^{2-}]$ on Mg incorporation at different temperatures. Specifically, the steeper response for subpolar
and polar foraminifera would imply that the $[\text{CO}_3^{2-}]$ effect is greater at low temperatures indicating
that the high sensitivity of Mg to low $[\text{CO}_3^{2-}]$ may be linked to the decreasing efficiency of Mg exclusion

under conditions less favourable for calcification. Along this line of argument [46]
 hypothesized that for low-Mg foraminifera like *N. pachyderma* the mechanism by which Mg is removed
 during calcification competes energetically with the need to modify the carbonate chemistry of
 seawater for calcification at low $[\text{CO}_3^{2-}]$. They propose that it may be more effective for foraminifera
 to prioritise the removal of H^+ over Mg^{2+} at low $[\text{CO}_3^{2-}]$ (this is also supported by [47]). Also, low
 metabolic/respiration rates hypothesized at low temperatures [48] may further reduce the energy
 available (required) to exclude Mg^{2+} and modify the carbonate chemistry for calcification, leading to
 the negative relationship between Mg/Ca and $[\text{CO}_3^{2-}]$ at low values.

 **Figure 1 Calibration.** In all panels, closed black squares refer to *N. pachyderma* collected from plankton tow samples, open
 black squares represent *N. pachyderma* from core tops [5] and open blue squares represent *N. incompta* from core tops [40].
 (a.) shows measured Mg/Ca values. (b.) shows normalized Mg/Ca values against water-column $[\text{CO}_3^{2-}]$ for tows and $\delta^{18}\text{O}_c$
 derived $[\text{CO}_3^{2-}]$ values for core tops. In yellow are the 95 % confidence intervals associated with the regression. (c.) shows
 152 Mg/Ca values corrected for $[\text{CO}_3^{2-}]$ seawater carbonate-ion concentration, in addition pink crosses show uncorrected Mg/Ca
 values for all *N. pachyderma* samples. In blue are the 95% confident intervals of fully propagated uncertainties, while in
 yellow, envelopes show the fully predicted uncertainties. (d.) shows $\delta^{18}\text{O}_c$ values vs water column $[\text{CO}_3^{2-}]$. Samples from
 plankton tows are corrected for a +2‰ offset to account for the difference between ontogenetic and crust calcite. In blue

are the 95% confidence intervals for the regression combining both tow and core top data, while the yellow and red shading
show the 95% confidence intervals for the tows and core top regressions respectively. In addition, function plots are shown
for other unicellular planktonic calcifying organisms [49, 50].

Our new Mg/Ca data from plankton samples provide an opportunity to directly quantify the carbonate
ion concentration effect on Mg/Ca measured in Arctic *N. pachyderma*. This is because the carbonate
ion concentration for the calcification environment can be determined reliably, and the sampled
environments cover the entire range of low carbonate ion conditions from 200 to less than 50 μM .
1b). The results reveal that the departure of the measured Mg/Ca from the theoretical Mg/Ca,
calculated assuming they reflect only ambient temperature, is systematically exponentially correlated
with $[\text{CO}_3^{2-}]$. When deriving $[\text{CO}_3^{2-}]$ values at the apparent calcification depths for the new and
previously published core top values included in this study (see methods), we find that the Mg/Ca
sensitivity for core top samples is within the 95% confidence intervals of the sensitivity derived for
plankton tow samples (Fig. 1b). This is important, as it demonstrates that the climate signal recorded
in the living foraminifera is transferred and preserved at the seafloor and in the climate archive.
Furthermore, it shows that the assumptions made with regards to the calcification environment for
core tops are generally valid. This being said, there is strong consensus that Mg/Ca values are lower
in crust versus ontogenetic calcite for *N. pachyderma* [30, 31, 51] and more recent findings show that
173 Mg/Ca of the outer crust of *N. pachyderma* is between 8.6-15% higher in living compared to dead
specimen recovered from the same plankton tows [52] in order to compare Mg/Ca values from tows
and core tops we therefore adjust uncorrected Mg/Ca values from plankton tow samples down by 15%
to account for the difference in Mg between ontogenetic and crust calcite. Given that core top
specimens are generally more encrusted than dead specimen from tows a 15% correction is a
conservative estimate of the true difference.

With plankton tow and core top samples combined, $\text{Mg}/\text{Ca}_{[\text{Corr}]}$ describe the following equation:
$\text{Mg}/\text{Ca}_{[\text{Corr}]} = 0.372 \pm 0.014 \exp^{(0.0936 \pm 0.0045T)}$, which shares the same sensitivity (within error) of the
predicted calibration equation, showing that $[\text{CO}_3^{2-}]$ is the sole or at least dominant non-thermal
control on the predicted Mg/Ca – temperature relationship (Fig. 1c). Our results also demonstrate that
the influence of $[\text{CO}_3^{2-}]$ on Mg/Ca measured on *N. pachyderma* can be mathematically quantified and
removed from temperature as for other planktonic foraminifera species (e.g., [41]), when $[\text{CO}_3^{2-}]$ is
known. Thus, for applications of the Mg/Ca paleothermometer on fossil material, an independent
$[\text{CO}_3^{2-}]$ proxy is required. Here, we investigate the influence of $[\text{CO}_3^{2-}]$ on oxygen isotopic
fractionation and evaluate its potential to serve as a proxy for the Mg/Ca correction scheme for *N.*
*pachyderma*. $\delta^{18}\text{O}_c$ is a traditional palaeoceanographic tracer combining the influence of $\delta^{18}\text{O}_{\text{sw}}$ of
seawater and temperature [53]. However, in addition to temperature, a dependence of oxygen isotope

fractionation on the carbonate chemistry of seawater has previously been demonstrated in planktonic
foraminifera [50, 54] and other unicellular planktonic calcifying organisms [49], suggesting the
existence of a universal carbonate ion effect on ~~stable~~ oxygen isotope ratios measured in unicellular
planktonic calcifying organisms.

To evaluate whether $\delta^{18}\text{O}_c/[\text{CO}_3^{2-}]$ values can be derived from $\delta^{18}\text{O}_c$ ~~in the past~~, we first tested if the
$\delta^{18}\text{O}_c/[\text{CO}_3^{2-}]$ sensitivity observed in material from plankton tows and core tops (Fig. 1d) is
compromised by interactions with $\delta^{18}\text{O}_{\text{sw}}$ and/or temperature in a generalized linear model including
$\delta^{18}\text{O}_c$, $\delta^{18}\text{O}_{\text{sw}}$, $[\text{CO}_3^{2-}]$, and temperature. We found that $\delta^{18}\text{O}_{\text{sw}}$ does not significantly contribute to the
total variance ($n=52$, $p>0.01$, Extended Data Table 5) and therefore **removed** it from considerations.
Furthermore, we **find** that temperature only weakly contributes to the total variance ($p=0.017$)
observed in the model. **Importantly, the interaction between $[\text{CO}_3^{2-}]$ and temperature is not**
**significant ($n=52$, $p=0.14$)** suggesting that while temperature contributes to the total variance in $\delta^{18}\text{O}_c$
it does not influence or interact with the $\delta^{18}\text{O}_c/[\text{CO}_3^{2-}]$ relationship. Nevertheless, we evaluate the
contribution of temperature and the potential of a non-linear effect of temperature on $\delta^{18}\text{O}_c$ further,
using a regression tree (Extended Data Figure S2). As expected, the regression tree shows that $[\text{CO}_3^{2-}]$
explains most of the variance and that temperature contributes to the variance in $\delta^{18}\text{O}_c$ only for values
above 5°C . When analysing the relationships between temperature, $[\text{CO}_3^{2-}]$, and $\delta^{18}\text{O}_c$ separately, **we**
**find that temperature and $[\text{CO}_3^{2-}]$ only covary above 5°C or $150\mu\text{mol.kg}^{-1}$ but not at colder**
**temperatures or lower concentrations (Extended Data Figure S3).** Similarly, the covariance between
temperature and $\delta^{18}\text{O}_c$ breaks down below $4\text{--}5^\circ\text{C}$ (also observed in [55]), while the relationship
between $\delta^{18}\text{O}_c$ and $[\text{CO}_3^{2-}]$ remains unchanged regardless of concentrations (Extended Data Figure
S3). These results thus confirm that the $\delta^{18}\text{O}_c/[\text{CO}_3^{2-}]$ sensitivity does not rely on a covariance with
temperature, which varies over space and time.

Following, we posit that the sensitivity or slope of $\delta^{18}\text{O}_c/[\text{CO}_3^{2-}]$ is regulated by the calcite saturation
product (K_{sp}), which in turn is determined by the conversion between (isotopically heavier) bicarbonate
and (isotopically lighter) carbonate **ion** in the calcification environment, that is a function of pH,
salinity, and to a lesser extent temperature [49]. We observe that the slopes of $\delta^{18}\text{O}_c/[\text{CO}_3^{2-}]$ for *N.*
*pachyderma* from tows and core tops are statistically the same at -0.0279 but offset by $\sim 2\text{‰}$ (Fig. 1d).
The slopes in our dataset are also steeper than reported for *Orbulina universa* (-0.002) or *Globigerina*
*bulloides* (e.g., -0.004) [50], but not outside of the slopes observed previously for other unicellular
planktonic calcifying organisms [49] (Fig. 1d). The steeper slopes for *N. pachyderma* suggest that the

relative contribution of the isotopically light $[\text{CO}_3^{2-}]$ to the calcifying fluid is low and that comparatively
less bicarbonate is converted into carbonate ion in the calcification environment when compared to
*O. universa*, or *G. bulloides*. This hypothesis agrees well with a different biomineralization processes
controlling the exclusion/uptake of Mg described earlier in *N. pachyderma*, where low cell energy due
to unfavourable calcification environments and low metabolic/respiration rates at cold temperatures
leads to low $[\text{CO}_3^{2-}]$ available for calcification. In addition, the release of CO_2 during respiration, and
the absence of photosynthetic activity from symbionts, further decrease the relative contribution of
isotopically light $[\text{CO}_3^{2-}]$ in favour for the isotopically heavier bicarbonate and potentially further
enhances the sensitivity of oxygen isotope fractionation in low $[\text{CO}_3^{2-}]$. The observed 2‰ offset in the
intercepts between tows and core tops agrees well with the average 2.1 ‰ offset observed between
ontogenetic calcite and a fully crusted *N. pachyderma* [56] which have also been linked to slow (e.g.,
crust) and fast (e.g. ontogenetic calcite) calcification rates in *N. pachyderma*. A caveat of this offset is
that specimens of a similar degree of encrustation must be analysed together to avoid the influence
of varying degrees of encrustation on the $\delta^{18}\text{O}_c/[\text{CO}_3^{2-}]$ relationship. Accordingly, the larger scatter
observed for the core top versus plankton tow dataset could potentially be explained by different
degrees of crust formation [56].

To explore the application potential of the $\delta^{18}\text{O}_c/[\text{CO}_3^{2-}]$ proxy to fossil material, we compared $[\text{CO}_3^{2-}]$
derived from $\delta^{18}\text{O}_c$ with CARINA [57] derived $[\text{CO}_3^{2-}]$ values for the Kozdon core top dataset (see
methods). Both methods provide $[\text{CO}_3^{2-}]$ values that are within a root mean square error (RMSE) of
$8.44 \mu\text{mol.kg}^{-1}$ of each other (adjusted $r^2=0.95$ $n=52$, $p<0.001$). Residuals are symmetrically distributed
around zero (Extended Data Figure S4). When $\delta^{18}\text{O}_c$ derived $[\text{CO}_3^{2-}]$ values are used to estimate
$\text{Mg}/\text{Ca}-[\text{CO}_3^{2-}]$ sensitivity, the scatter is much lower (when compared to CARINA) resulting in more
robust correlation statistics (adjusted $r^2=0.58$ vs $r^2=0.81$) and lower uncertainties. The reduced scatter
is most likely due to the uncertainties linked to assigning modern water column hydrography from
CARINA to pre-industrial/late Holocene core tops. By using $\delta^{18}\text{O}_c$ to derive $[\text{CO}_3^{2-}]$ these uncertainties
are removed. To further test the validity of this approach we also derived $[\text{CO}_3^{2-}]$ from $\delta^{18}\text{O}_c$ values
published in the core top dataset (paired Mg/Ca - $\delta^{18}\text{O}_c$ published in Meland *et al.* 2006 [4] and find
that the Mg/Ca sensitivity to $[\text{CO}_3^{2-}]$ is statistically identical with the tows and the Kozdon datasets
(Fig. 1b).

Finally, we combine all *N. pachyderma* datasets with a previously published dataset for *N. incompta*
[40]. When all datasets are combined we derived the following Mg/Ca - $[\text{CO}_3^{2-}]$ sensitivity $\text{Mg}/\text{Ca}_{[\text{CO}_3^{2-}]}$

$T = 3227.95 * [CO_3^{2-}]^{-1.53}$ (n=235, $r^2=0.87$, $p<0.01$). All uncertainties (including calibration uncertainties
accounting for covariance of error in calibration coefficients) are propagated through Monte-Carlo
simulation. [41]. Briefly, 1000 permutations are made selecting random values within the uncertainty
range for individual parameters ($\delta^{18}O_c$, $[CO_3^{2-}]$, Mg/Ca, and Temperature) and a calibration slope
applied for each. The median of these slopes is taken as our central calibration estimate with the 2σ
described by the 95th percentile of potential slopes. We also include the prediction interval for the
Monte Carlo Ensemble (Fig. 1c), see Methods for full details) Typical uncertainty in calibrated
temperatures, combining calibration uncertainty, and $[CO_3^{2-}]$ uncertainty, is ± 1.0 °C (± 1.2 °C at 0°C and
± 0.8 °C at 15°C, 2σ) if $\delta^{18}O_c$ are used to correct for $[CO_3^{2-}]$.

The combination of the Mg/Ca correction scheme and $\delta^{18}O_c$ derived $[CO_3^{2-}]$ values present a unique
opportunity to apply the correction to previously published Mg/Ca - $\delta^{18}O_c$ datasets and estimate SST
in glacial polar environments. In this way it allows us to reevaluate the magnitude and spatial
expression of marine polar amplification to glacial – interglacial climate forcing as recorded by Mg/Ca
and $\delta^{18}O_c$ values. Using previously published Mg/Ca - $\delta^{18}O_c$ datasets [32, 33, 58] we show that
uncorrected Mg/Ca based SSTs systematically underestimate maximum glacial cooling by 2 to $3^\circ \pm 1.0$
273 °C for four previous glacial maxima (Fig. 2, Extended Data Table 6). Furthermore, warming from glacial
to interglacial SST in the Labrador Sea at IODP Site U1305 is much higher based on Mg/Ca_[corr] values
for the past five glacial interglacial cycles increasing from 2.3–4.2 for uncorrected Mg/Ca values to 7.9–
11.0 ± 1.0 °C once corrected. Assemblage based (e.g., modern analogue technique; MAT) glacial-
interglacial SST gradients for the same site range between 7–9°C. These are similar but on average
warmer 1.5°C than our estimates, which is likely the result of low species diversity below 4 °C leading
to higher SST values during glacial maxima [59].

**Figure 2 Paleo application** of Mg/Ca correction scheme to four glacial-Interglacial periods including MIS1/2, Mis 5, MIS9 and
 MIS11 [32, 33, 58]. Purple lines show three-point running averages of corrected Mg/Ca based SSTs while grey lines show
 three-point running averages of uncorrected Mg/Ca based SST reconstructions. The blue shading shows the 95% confidence
 interval associated with Mg/Ca_[Corr] SST reconstructions. In pink are MAT based SST reconstructions as shown in Irali et al.
 [33] for MIS 5, 9, and 11. Note, MAT SST are not available for the last glacial interglacial transition.

A paleoceanographic map of corrected *N. pachyderma* Mg/Ca records covering the LGM and Late
 Holocene, demonstrates that the underestimated cooling (e.g., Δ SST) is most pronounced across mid-
 latitudes where our new estimates suggest cooling between $6-9 \pm 1.0^\circ\text{C}$ instead of milder $4-6^\circ\text{C}$ cooling
 (Fig. 3a). Our re-assessment suggests that polar amplification or high-latitude LGM cooling is much
 more extensive in magnitude and spatial expression than previously estimated, which raises
 questions about our understanding of heat transport under glacial boundary conditions. Furthermore,
 absolute Holocene and LGM SST values (Fig. 3b and 3c) are much closer to model-based
 reconstructions of LGM SST than previous estimates based on *N. pachyderma* (Extended Data Fig S5).
 Considering these examples, we believe that a systematic application of the calibration approach as
 presented in our study provides for the first time an opportunity to directly estimate past temperatures

in Polar Oceans, where low diversity of planktonic foraminifera prevented the use of transfer functions
and where dinocysts are rare.

**Figure 3 Data model intercomparison.** The top panel shows the difference in cooling between the Holocene and the LGM
(Δ SST) calculated using the isotope-enabled Community Earth System Model (iCESM) as shown in Tierney et al.[9] in the
background together with Δ SST calculated using corrected Mg/Ca values measured on *N. pachyderma* (symbols). For a list of
all datasets used please see Extended Data Table 7. The middle and bottom panel show Holocene and LGM SST respectively,
with the modelled (iCESM) SST in the background and corrected Mg/Ca based SST shown as symbols. Maps were produced
with *Ocean Data View*. Please note that modelled annual SST are likely too cold for a direct comparison with *N. pachyderma*-
based SST given their seasonal reproductive cycle. Maps displaying uncorrected Mg/Ca values against modelled paleodata
are shown in Extended Data Figure S5.

[revised manuscript text omitted]

For the core top datasets, calcification depth and hydrographic parameters were estimated by
comparing the measured foraminiferal $\delta^{18}\text{O}_c$ values to the expected equilibrium values of calcite
($\delta^{18}\text{O}_c$) precipitated from seawater. For the Kozdon dataset we selected temperature, salinity, and
carbonate chemistry using Carina [57], while paired hydrographic profiles were used for CE20009 core
tops. Since the dominant flux of *N. pachyderma* occurs in the summer, we only selected profiles
collected between June and October from stations as close to the core sites as possible for the Kozdon
dataset. Specific stations, locations and dates of profiles are detailed in Extended Data Table 8. To
obtain $\delta^{18}\text{O}_{\text{sw}}$ depth profiles, we used modelled water-column data from LeGrande and Schmidt [67].
We retrieved data from both the closest station to each core site and within a 0.5° radius to provide
locally averaged $\delta^{18}\text{O}_{\text{sw}}$ values. For the CE20009 core tops paired water column $\delta^{18}\text{O}_{\text{sw}}$ values are
available. **We then calculated site-specific $\delta^{18}\text{O}_{\text{sw}}$ - salinity linear relationships for each core.** Using the
$\delta^{18}\text{O}$ - paleotemperature equation developed by O'Neil *et al.* [68] we **determined expected $\delta^{18}\text{O}_c$**
**depth profiles (0–400 m) based on seasonal temperature, salinity, and $\delta^{18}\text{O}_{\text{sw}}$ data.** Calculations were
completed by subtracting **0.27‰**. The depth at which the expected equilibrium **$\delta^{18}\text{O}_c$ values matched**
**the measured foraminiferal $\delta^{18}\text{O}_c$ values (determined by linear interpolation) is the calcification depth,**
and the calcification temperature, salinity, and carbonate chemistry was determined based on

respective values recorded at this calcification depth. Except for two stations that yield anomalously
deep estimates, all calcification depths fall between 7-67 m water depth (Extended Data Table 8).

Analytical Methods:

To prepare foraminifera from plankton tows from MSM cruises for TE analysis, foraminifera tests were
gently crushed between two glass plates and transferred into acid-leached vials. The crushed
foraminiferal tests were cleaned using a cleaning protocol focussing on the removal of organic matter
(i.e., the oxidative step was repeated twice) followed by a final dissolution in dilute HNO₃. This
reduced cleaning protocol was adopted since clay or metal oxide removal are not required for tow
samples. Samples from MSM cruises were analysed using a Nu AttoM high-resolution double-focusing
inductively coupled plasma mass spectrometer at the Alfred Wegner Institute in Bremerhaven,
Germany. Five replications were carried out for each sample, where the average repeatability for
384 Mg/Ca among the five replicates was $0.95 \pm 0.47\%$ (relative standard deviation). Long-term
reproducibility of Mg/Ca analyses is monitored by regular analysis of the carbonate standard JcP-1,
yielding 3.94 ± 0.15 (2σ) mmol.mol⁻¹ over a period of about two years. Samples from cruise CE20009
were processed for elemental analysis at the University of Southampton. They were cleaned using
established methods (Barker et al., 2003), clay removed (for core top samples only) and oxidatively
cleaned with a solution of H₂O₂ (30% by weight) buffered with 0.1M NH₄OH (3.4% of peroxide in the
final oxidative mix). Samples were placed in a hot bath for 3 x 5 min for core tops and 3 x 20 min for
tows (with a fresh solution of oxidative mix each time for tows), separated by brief ultrasonication.
Samples were then weak acid-leached (in 0.0005 M HNO₃) for 30 seconds and dissolved in ~300 ul of
0.15 M HNO₃. An aliquot of 20 µl (diluted in 130 µl 0.5M HNO₃) was kept for elemental analysis.
Elemental analysis was performed on a Thermo Element Inductively Couple Plasma Mass
Spectrometer (ICPMS) and element-to-calcium rations measured with ⁴³Ca and ⁴⁸Ca and measured
against in house mixed element standards. Stable isotope analysis on MSM samples were performed
on a Finnigan MAT 252 mass spectrometer equipped with an automated carbonate preparation device
at MARUM, University of Bremen. Isotopic results were calibrated relative to the Vienna Pee Dee
belemnite (VPDB) using the NBS19 standard. The standard deviation of the laboratory standard was
lower than 0.07‰ for δ¹⁸O_c over the measuring period. Samples from CE20009 were analysed at
FARLAB, University of Bergen using a Thermo Scientific MAT 253 IRMS coupled to a Kiel IV automated
carbonate preparation device. Results are reported on the VPDB scale calibrated with NBS19, together
with NBS18, and internal standards CM12 and Riedal. Long term external precision (1 σ SD) is lower
than 0.08‰ for δ¹⁸O_c based on replicate analysis of the in house standard CM12.

**Paleodata correction:**

For all previously published datasets $\delta^{18}\text{O}_c$ data were corrected for glacial-interglacial sea-level
changes using sea-level reconstructions by Spratt and Lisiecki [69]. Here we use the commonly cited
value of 0.11‰/10m SL change for the correction of $\delta^{18}\text{O}_c$ [70-73]. We also applied a +10-15% offset
to $\text{Mg}/\text{Ca}_{[\text{meas}]}$ values that underwent a reductive cleaning step, (e.g., [32, 33]) or repeated acid leaches
[74] since these additional steps have been shown to lower Mg/Ca values in planktonic foraminifera
[75]. We then derived paleo $[\text{CO}_3^{2-}]$ values from $\delta^{18}\text{O}_c$ and corrected $\text{Mg}/\text{Ca}_{[\text{meas}]}$ using the correction
scheme presented here. For the Late Holocene – LGM compilation we included as many available
records from literature as possible. Extended Data Table 7 details core sites, time intervals and
references used for each estimate. For the calculation of mean glacial and interglacial values covering
the last five glacial-interglacial cycles from IODP site U1305 please see Extended Data Table 6.

**Calibration uncertainty calculations.**

For the *N.pachyderma* data, first the relationship between $\delta^{18}\text{O}$ and carbonate ion  assessed (Fig.
1d). We perform a Monte Carlo simulation (n=1000) on the slope of this relationship using the
analytical uncertainty on the oxygen isotope measurements (typically 0.2 at 2σ) and carbonate ion
from hydrographic data (GLODAP) with a prescribed uncertainty of 20 $\mu\text{mol}/\text{kg}$ (2σ). The error on this
slope is then propagated into the $\text{CO}_3^{2-}:\text{Mg}/\text{Ca}_{[\text{norm}]}$ relationship, where CO_3^{2-} as given by $\delta^{18}\text{O}$ is plotted
against normalised Mg/Ca ($\text{Mg}/\text{Ca}_{[\text{meas}]}/\text{Mg}/\text{Ca}_{[\text{pred}]}$) to give a $\text{Mg}/\text{Ca}_{[\text{corr}]}$ for non-thermal effects. Based
on the previous relationships, $\text{Mg}/\text{Ca}_{[\text{corr}]}$ has an uncertainty of 0.2 (2σ) which we then calibrate against
the in situ temperatures from CTD cruise and/or the CARINA database, the errors of which are related
to the depth estimation of the foraminifera but typically range from 0 – 3°C (minimum 0°C, maximum
10°C). e for the data from Meland et al. [4] we are unable to calculated an uncertainty so we use
an average of the uncertainties from Kozdon et al [5] (1.6°C at 2σ). For *N. incompta* data, all the steps
wed are identical to the *pachyderma* dataset except, as there are no $\delta^{18}\text{O}$ data 431 uncertainty is prescribed to be $\pm 0.8^\circ\text{C}$ (2σ) [40]. The $\text{Mg}/\text{Ca}_{[\text{corr}]}$ uncertainty is assumed to be the same
as for *N. pachyderma*. Both species are combined to make the final calibration.

The calibration Monte Carlo simulation is conducted on 1000 iterations of a slope using the
propagated $\text{Mg}/\text{Ca}_{[\text{corr}]}$ and the individual temperature uncertainties. We take the median slope of this
curve to define the $\text{Mg}/\text{Ca}_{[\text{corr}]}$ to temperature relationship, and the 68 and 95 % confidence intervals
(1 and 2σ respectively) to describe its uncertainties. These uncertainties represent the uncertainty in
the defined relationship, assuming it to be consistent and  temperature reconstruction
uncertainty of $\pm 1.2^\circ\text{C}$ at temperatures of around 0°C and $\pm 0.9^\circ\text{C}$ at temperatures of around 15°C.

These confidence intervals give a conservative uncertainty on the relationship between Mg/Ca_[corr] and
temperature and are used for any relative change calculations presented in this study. We also plot a
prediction uncertainty, which can be considered the uncertainty from a single future datapoint but is
not representative for a whole population of data.

**References:**

- 1. Zhu, J., C.J. Poulsen, and B.L. Otto-Bliesner, *High climate sensitivity in CMIP6 model not*
*supported by paleoclimate*. Nature Climate Change, 2020. **10**(5): p. 378-379.
- 2. Kucera, M., et al., *Reconstruction of sea-surface temperatures from assemblages of planktonic*
*foraminifera: multi-technique approach based on geographically constrained calibration data*
*sets and its application to glacial Atlantic and Pacific Oceans*. Quaternary Science Reviews,
2005. **24**(7): p. 951-998.
- 3. de Vernal, A. and C. Hillaire-Marcel, *Natural Variability of Greenland Climate, Vegetation, and*
*Ice Volume During the Past Million Years*. Science, 2008. **320**(5883): p. 1622-1625.
- 4. Meland, M.Y., et al., *Mg/Ca ratios in the planktonic foraminifer Neogloboquadrina pachyderma*
*(sinistral) in the northern North Atlantic/Nordic Seas*. Geochemistry, Geophysics, Geosystems,
2006. **7**(6).
- 5. Kozdon, R., et al., *Reassessing Mg/Ca temperature calibrations of Neogloboquadrina*
*pachyderma (sinistral) using paired $\delta^{44}/^{40}\text{Ca}$ and Mg/Ca measurements*. Geochemistry,
Geophysics, Geosystems, 2009. **10**(3).
- 6. Serreze, M.C. and R.G. Barry, *Processes and impacts of Arctic amplification: A research*
*synthesis*. Global and Planetary Change, 2011. **77**(1): p. 85-96.
- 7. Dai, A., et al., *Arctic amplification is caused by sea-ice loss under increasing CO₂*. Nature
communications, 2019. **10**(1): p. 121.
- 8. Walter Anthony, K., et al., *Methane emissions proportional to permafrost carbon thawed in*
*Arctic lakes since the 1950s*. Nature Geoscience, 2016. **9**(9): p. 679-682.
- 9. Tierney, J.E., et al., *Glacial cooling and climate sensitivity revisited*. Nature, 2020. **584**(7822):
p. 569-573.
- 10. Holland, M.M. and C.M. Bitz, *Polar amplification of climate change in coupled models*. Climate
dynamics, 2003. **21**(3-4): p. 221-232.
- 11. Manabe, S. and R.J. Stouffer, *Sensitivity of a global climate model to an increase of CO₂*
*concentration in the atmosphere*. Journal of Geophysical Research: Oceans, 1980. **85**(C10): p.
5529-5554.
- 12. Årthun, M., et al., *Skillful prediction of northern climate provided by the ocean*. Nature
communications, 2017. **8**(1): p. 1-11.
- 13. Årthun, M., T. Eldevik, and L.H. Smedsrud, *The role of Atlantic heat transport in future Arctic*
*winter sea ice loss*. Journal of Climate, 2019. **32**(11): p. 3327-3341.
- 14. Parmentier, F.-J.W., et al., *The impact of lower sea-ice extent on Arctic greenhouse-gas*
*exchange*. Nature climate change, 2013. **3**(3): p. 195-202.
- 15. Jansen, E., et al., *Past perspectives on the present era of abrupt Arctic climate change*. Nature
Climate Change, 2020. **10**(8): p. 714-721.
- 16. Vavrus, S.J., *The influence of Arctic amplification on mid-latitude weather and climate*. Current
Climate Change Reports, 2018. **4**(3): p. 238-249.
- 17. Boetius, A., et al., *Microbial ecology of the cryosphere: sea ice and glacial habitats*. Nature
Reviews Microbiology, 2015. **13**(11): p. 677-690.
- 18. Huang, J., et al., *Recently amplified arctic warming has contributed to a continual global*
*warming trend*. Nature climate change, 2017. **7**(12): p. 875-879.

- 19. Meredith, M., et al., *Polar regions. chapter 3, ipcc special report on the ocean and cryosphere*
*in a changing climate*. 2019.
- 20. Cowtan, K. and R.G. Way, *Coverage bias in the HadCRUT4 temperature series and its impact*
*on recent temperature trends*. Quarterly Journal of the Royal Meteorological Society, 2014.
**140**(683): p. 1935-1944.
- 21. Burls, N. and N. Sagoo, *Increasingly Sophisticated Climate Models Need the Out-Of-Sample*
*Tests Paleoclimates Provide*. Journal of Advances in Modeling Earth Systems, 2022. **14**(12): p.
e2022MS003389.
- 22. Fedorov, A.V., et al., *Tightly linked zonal and meridional sea surface temperature gradients over*
*the past five million years*. Nature Geoscience, 2015. **8**(12): p. 975-980.
- 23. Polyakov, I.V., et al., *Greater role for Atlantic inflows on sea-ice loss in the Eurasian Basin of the*
*Arctic Ocean*. Science, 2017. **356**(6335): p. 285-291.
- 24. Telford, R.J., *Limitations of dinoflagellate cyst transfer functions*. Quaternary Science Reviews,
2006. **25**(13-14): p. 1375-1382.
- 25. Bendle, J., A. Rosell-Melé, and P. Ziveri, *Variability of unusual distributions of alkenones in the*
*surface waters of the Nordic seas*. Paleoceanography, 2005. **20**(2).
- 26. Katz, M.E., et al., *Traditional and emerging geochemical proxies in foraminifera* Journal of
Foraminiferal Research, 2010. **40**(2): p. 165-192.
- 27. Katz, A., *The interaction of magnesium with calcite during crystal growth at 25–90 C and one*
*atmosphere*. Geochimica et Cosmochimica Acta, 1973. **37**(6): p. 1563IN31579-15781586.
- 28. Nuernberg, D., *Magnesium in tests of Neogloboquadrina pachyderma sinistral from high*
*northern and southern latitudes*. The Journal of Foraminiferal Research, 1995. **25**(4): p. 350-
368.
- 29. Hendry, K.R., et al., *Controls on stable isotope and trace metal uptake in Neogloboquadrina*
*pachyderma (sinistral) from an Antarctic sea-ice environment*. Earth and Planetary Science
Letters, 2009. **278**(1): p. 67-77.
- 30. Davis, C.V., et al., *Relationships Between Temperature, pH, and Crusting on Mg/Ca Ratios in*
*Laboratory-Grown Neogloboquadrina Foraminifera*. Paleoceanography, 2017. **32**(11): p. 1137-
1152.
- 31. Livsey, C.M., et al., *High-resolution Mg/Ca and $\delta^{18}O$ patterns in modern Neogloboquadrina*
*pachyderma from the Fram Strait and Irminger Sea*. Paleoceanography and Paleoclimatology,
2020. **35**(9): p. e2020PA003969.
- 32. Irvali, N., et al., *Evidence for regional cooling, frontal advances, and East Greenland Ice Sheet*
*changes during the demise of the last interglacial*. Quaternary Science Reviews, 2016. **150**: p.
184-199.
- 33. Irvali, N., et al., *A low climate threshold for south Greenland Ice Sheet demise during the Late*
*Pleistocene*. Proceedings of the National Academy of Sciences, 2020. **117**(1): p. 190-195.
- 34. Greco, M., et al., *Depth habitat of the planktonic foraminifera *Neogloboquadrina**
*pachyderma in the northern high latitudes explained by sea-ice and chlorophyll*
*concentrations*. Biogeosciences, 2019. **16**(17): p. 3425-3437.
- 35. Jonkers, L., et al., *Seasonal stratification, shell flux, and oxygen isotope dynamics of left-coiling*
**N. pachyderma* and *T. quinqueloba* in the western subpolar North Atlantic*. Paleoceanography,
2010. **25**(2).
- 36. Evans, D., et al., *Revisiting carbonate chemistry controls on planktic foraminifera Mg/Ca:*
*implications for sea surface temperature and hydrology shifts over the Paleocene–Eocene*
*Thermal Maximum and Eocene–Oligocene transition*. Climate of the Past, 2016. **12**(4): p. 819-
835.
- 37. Kisakürek, B., et al., *Controls on shell Mg/Ca and Sr/Ca in cultured planktonic foraminiferan,*
**Globigerinoides ruber* (white)*. Earth and Planetary Science Letters, 2008. **273**(3-4): p. 260-269.

- 38. Russell, A.D., et al., *Effects of seawater carbonate ion concentration and temperature on shell*
*U, Mg, and Sr in cultured planktonic foraminifera*. *Geochimica et Cosmochimica Acta*, 2004.
**68**(21): p. 4347-4361.
- 39. Gray, W.R., et al., *The effects of temperature, salinity, and the carbonate system on Mg/Ca in*
*Globigerinoides ruber (white): A global sediment trap calibration*. *Earth and Planetary Science*
*Letters*, 2018. **482**: p. 607-620.
- 40. Morley, A., et al., *Environmental Controls on Mg/Ca in Neogloboquadrina incompta: A Core-*
*Top Study From the Subpolar North Atlantic*. *Geochemistry, Geophysics, Geosystems*, 2017.
- 41. Gray, W.R. and D. Evans, *Nonthermal influences on Mg/Ca in planktonic foraminifera: a review*
*of culture studies and application to the Last Glacial Maximum*. *Paleoceanography and*
*Paleoclimatology*, 2019. **34**(3): p. 306-315.
- 42. Tierney, J.E., et al., *Bayesian calibration of the Mg/Ca paleothermometer in planktic*
*foraminifera*. *Paleoceanography and Paleoclimatology*, 2019. **34**(12): p. 2005-2030.
- 43. Foster, G.L. and J.W. Rae, *Reconstructing ocean pH with boron isotopes in foraminifera*. *Annual*
*Review of Earth and Planetary Sciences*, 2016. **44**: p. 207-237.
- 44. Darling, K.F., et al., *A resolution for the coiling direction paradox in Neogloboquadrina*
*pachyderma*. *Paleoceanography*, 2006. **21**(2).
- 45. Schiebel, R. and C. Hemleben, *Planktic foraminifers in the modern ocean*. 2017.
- 46. Evans, D., W. Müller, and J. Erez, *Assessing foraminifera biomineralisation models through*
*trace element data of cultures under variable seawater chemistry*. *Geochimica et*
*Cosmochimica Acta*, 2018. **236**: p. 198-217.
- 47. Zeebe, R.E. and A. Sanyal, *Comparison of two potential strategies of planktonic foraminifera*
*for house building: Mg²⁺ or H⁺ removal?* *Geochimica et Cosmochimica Acta*, 2002. **66**(7): p.
1159-1169.
- 48. Lombard, F., et al., *Modelling the temperature dependent growth rates of planktic*
*foraminifera*. *Marine Micropaleontology*, 2009. **70**(1): p. 1-7.
- 49. Ziveri, P., et al., *A universal carbonate ion effect on stable oxygen isotope ratios in unicellular*
*planktonic calcifying organisms*. *Biogeosciences*, 2012. **9**(3): p. 1025-1032.
- 50. Bijma, J., H. Spero, and D. Lea, *Reassessing foraminiferal stable isotope geochemistry: Impact*
*of the oceanic carbonate system (experimental results)*, in *Use of proxies in paleoceanography*.
1999, Springer. p. 489-512.
- 51. Jonkers, L., et al., *Encrustation and trace element composition of Neogloboquadrina dutertrei*
*assessed from single chamber analyses—implications for paleotemperature estimates*.
*Biogeosciences*, 2012. **9**(11): p. 4851-4860.
- 52. Hupp, B.N. and J.S. Fehrenbacher, *Geochemical differences between alive, uncrusted and dead,*
*crusted shells of Neogloboquadrina pachyderma: Implications for paleoreconstruction*.
*Paleoceanography and Paleoclimatology*, 2023. **38**(10): p. e2023PA004638.
- 53. Ravelo, A.C. and C. Hillaire-Marcel, *Chapter eighteen the use of oxygen and carbon isotopes of*
*foraminifera in paleoceanography*. *Developments in marine geology*, 2007. **1**: p. 735-764.
- 54. Spero, H.J., et al., *Effect of seawater carbonate concentration on foraminiferal carbon and*
*oxygen isotopes*. *Nature*, 1997. **390**(497-500).
- 55. Bauch, D., J. Carstens, and G. Wefer, *Oxygen isotope composition of living Neogloboquadrina*
*pachyderma (sin.) in the Arctic Ocean*. *Earth and Planetary Science Letters*, 1997. **146**(1-2): p.
47-58.
- 56. Kozdon, R., et al., *Intratrust oxygen isotope variability in the planktonic foraminifer N.*
*pachyderma: Real vs. apparent vital effects by ion microprobe*. *Chemical Geology*, 2009. **258**(3-
4): p. 327-337.
- 57. Key, R., et al., *The CARINA data synthesis project: introduction and overview*. *Earth System*
*Science Data*, 2010. **2**(1): p. 105-121.
- 58. Thornalley, D.J., I.N. McCave, and H. Elderfield, *Freshwater input and abrupt deglacial climate*
*change in the North Atlantic*. *Paleoceanography*, 2010. **25**(1).

- 59. De Vernal, A., et al., *Comparing proxies for the reconstruction of LGM sea-surface conditions*
*in the northern North Atlantic*. Quaternary Science Reviews, 2006. **25**(21-22): p. 2820-2834.
- 60. Masson-Delmotte, V., et al., *Past and future polar amplification of climate change: climate*
*model intercomparisons and ice-core constraints*. Climate Dynamics, 2006. **26**: p. 513-529.
- 61. Zhu, J., et al., *LGM paleoclimate constraints inform cloud parameterizations and equilibrium*
*climate sensitivity in CESM2*. Journal of Advances in Modeling Earth Systems, 2022. **14**(4): p.
e2021MS002776.
- 62. Dutton, A., et al., *Sea-level rise due to polar ice-sheet mass loss during past warm periods*.
*science*, 2015. **349**(6244): p. aaa4019.
- 63. Menviel, L.C., et al., *An ice–climate oscillatory framework for Dansgaard–Oeschger cycles*.
*Nature Reviews Earth & Environment*, 2020. **1**(12): p. 677-693.
- 64. Kageyama, M., et al., *Modeling the climate of the Last Glacial Maximum from PMIP1 to PMIP4*.
*Past Global Changes Magazine*, 2021. **29**(2): p. 80-81.
- 65. Mintrop, L., et al., *Alkalinity determination by potentiometry: Intercalibration using three*
*different methods*. 2000.
- 66. Dickson, A., C. Sabine, and J. Christian, *Guide to best practices for ocean CO2 measurements*,
*PICES Special Publication 3*. 2007, IOCCP report.
- 67. LeGrande, A.N. and G.A. Schmidt, *Global gridded data set of the oxygen isotopic composition*
*in seawater*. Geophysical Research Letters, 2006. **33**(12): p. L12604.
- 68. O'Neil, J.R., R.N. Clayton, and T.K. Mayeda, *Oxygen Isotope Fractionation in Divalent Metal*
*Carbonates*. The Journal of Chemical Physics, 1969. **51**(12): p. 5547-5558.
- 69. Spratt, R.M. and L.E. Lisiecki, *A Late Pleistocene sea level stack*. Climate of the Past, 2016. **12**(4):
p. 1079-1092.
- 70. Fairbanks, R.G., *A 17,000 year glacio-eustatic sea level record: influence of glacial melting rate*
*on the Younger Dryas event and deep-ocean circulation*. Nature, 1989. **342**: p. 637-642.
- 71. Schrag, D.P., G. Hampt, and D.W. Murray, *Pore fluid constraints on the temperature and oxygen*
*isotopic composition of the glacial ocean*. Science, 1996. **272**: p. 1930-1932.
- 72. Adkins, J.F., K. McIntyre, and D.P. Schrag, *The salinity, temperature, and $\delta^{18}O$ of the glacial*
*deep ocean*. Science, 2002. **298**(5599): p. 1769-1773.
- 73. Rohling, E., et al., *Sea-level and deep-sea-temperature variability over the past 5.3 million*
*years*. Nature, 2014. **508**(7497): p. 477-482.
- 74. Ezat, M.M., T.L. Rasmussen, and J. Groeneveld, *Reconstruction of hydrographic changes in the*
*southern Norwegian Sea during the past 135 kyr and the impact of different foraminiferal*
*Mg/Ca cleaning protocols*. Geochemistry, Geophysics, Geosystems, 2016. **17**(8): p. 3420-3436.
- 75. Barker, S., M. Greaves, and H. Elderfield, *A study of cleaning procedures used for foraminiferal*
*Mg/Ca paleothermometry*. Geochemistry Geophysics Geosystems, 2003. **4**: p. 8407.
- 76. Schlitzer, R., *Ocean Data View*. 2002, <http://ww.awi-bremenhaven.de/GEO/ODV,2002>.

**Acknowledgements.** This research was funded by MSCA-IF Project ARCTICO [838529] and the Marine
Institute of Ireland Research Programme 2014-2020 (PDOC/19/05/02) awarded to AM. In addition,
AM acknowledges Grant in Aid funding from the Marine Institute for research expedition CE20009 on
the RV Celtic Explorer with special thanks to the crew sailing under Master Anthony English and
Research Assistant Aedin McAlear for the analysis of Ocean ALK and DIC samples from the CE20009
survey. RC acknowledges funding for Project CE2COAST, funded by ANR (FR), BELSPO (BE), FCT (PT),
IZM (LV), MI (IE), MIUR (IT), Rannis (IS) and RCN (NO) through the 2019 "Joint Transnational Call on
Next Generation Climate Science in Europe for Oceans" initiated by JPI Climate and JPI Oceans.

**Author contributions.** AM designed the project, carried out palaeoceanographic sample preparation
of plankton tows from the Labrador Sea, compiled and analysed all datasets, wrote the manuscript,
and coordinated input of all co-authors. EDLV carried out palaeoceanographic sample preparation and
geochemical analysis of plankton tows and core tops from CE20009 and contributed to the writing of
the final manuscript. MR performed trace element and boron isotope analysis on *N pachyderma*
plankton tow samples from the Labrador Sea. JB supervised AM at the Alfred Wegner Institute and
oversaw trace element and boron isotope analysis. UN performed all stable isotope analysis of water
and foraminifera samples from CE20009 and contributed to the final version of this manuscript. GF
oversaw trace element and boron isotope analysis on *N. pachyderma* plankton tow samples from
CE20009 and contributed to the final version of this manuscript. TC performed uncertainty
calculations for the Mg/Ca correction and contributed to the final version of this manuscript. JM
collected all plankton tows during CE20009 and contributed to the final version of this manuscript. RC
oversaw Alkalinity and DIC analysis on seawater samples collected during CE20009 and contributed to
the final version of this manuscript. JB performed Alkalinity and DIC analysis on seawater samples
collected during CE20009 and contributed to the final version of this manuscript. MK advised AM
during project design, supervised her during her MSCA-IF at MARUM, University of Bremen, and
contributed to the final version of this manuscript.

**Competing interests.** Submission of a competing interests statement is required for all content of the
journal.

**Materials & Correspondence.** Correspondence and requests for materials should be addressed to AM

**Extended Data Figures**

**Extended Data Figure S1 Station Map.** Inverted triangle (red) show samples collected from plankton
 tows (e.g., MSM cruises in the Baffin Bay and Labrador Sea and CE cruise in the Nordic Sea), while black
 triangles show locations of the Kozdon dataset. In the Nordic Seas plankton tows are also paired with
 corresponding core tops. Temperature data shown is from the June-September WOA 2018 dataset at
 30m. The map was generated using Ocean Data View [76]

**Extended Data Figure S2 Regression Tree.** This plot describes the interaction of $[\text{CO}_3^{2-}]$ and
 temperature on $\delta^{18}\text{O}_c$ in the generalized model.

**Extended Data Figure S3.** The left panel shows the relationship between Temperature and $\delta^{18}\text{O}_c$, while
 the right panel shows the relationship between Temperature and $[\text{CO}_3^{2-}]$ for the combined tow and
 core top *N. pachyderma* datasets. The plankton tow dataset on *N. pachyderma* (left) and associated
 hydrographic data (derived using GIODAP (right) from Bauch et al. [55] is shown in green. The yellow
 shading highlights hydrographic conditions below 4°C.

**Extended Data Figure S4** Residual Plots. (Top) shows the regression and residuals for the $\delta^{18}\text{O}_c$ and
$[\text{CO}_3^{2-}]$ relationship. (Bottom) shows the regression and residuals for the relationship describing
$[\text{CO}_3^{2-}]$ derived from CARINA and $[\text{CO}_3^{2-}]$ derived from $\delta^{18}\text{O}_c$.

**Extended Data Figure S5** Uncorrected LGM (top) and Late Holocene (bottom) SST based on *N.*
*pachyderma* Mg/Ca calibrated using the calibration equation developed by Kozdon et al. 2009 [5]. For
datasets used please see Extended Data Table 7.

Extended Data Table 8. Hydrographic data, **satble** isotopes and Mg/Ca used for the reanalysis of Kozdon et al.

Site	Long	Lat	type	Core depth	Hydrographic data CARINA [T, S, ALK, DIC, Silicat		
					[m]	stations	date
HM71-17	-13.02	70.00	core top	1460	7; 9; 91	02/06/2002; 18/06/2002	-13.00; -11.82; - 14.02
HM80-43	-9.19	72.25	core top	2448	16; 489; 491	03/06/2002; 08/08/1998	-8.53; -11.00; - 7.05
23235-1	1.39	78.87	core top	2500	31; 32; 33	6/7/2002	2.42; 2.54; 1.85
HM94-30	-2.00	74.38	core top	3599	502; 510; 33; 36	15&18/08/1998; 01&02/08/1996	-2.99; -2.04; - 1.52; -3.01
HM94-12	-3.55	71.32	core top	1816	18	6/4/2002	-7.12
HM97-948	7.64	66.97	core top	1048	117, 118, 119	6/23/2002	6.29; 7.58; 8.83
HM52-18	-14.14	62.27	core top	1672	195; 197	10/7/2003	-17.84; -17.86
HM16132	-0.72	64.57	core top	2798	110; 126; 217	22&25/06/2002; 12/10/2003	0.5898; -1.62; 1.79
HM57-11	-8.30	67.12	core top	1617	100; 101; 102; 191	20/06/2002; 01/10/2003	-9.64; -8.47; - 7.98; -9.24
HM57-20	1.67	62.65	core top	750	217; 221; 223	12/10/2003; 13/10/2003	1.79; 3.41; 3.92
HM94-16	5.37	73.23	core top	2356	158; 497; 500; 27	26/09/2003; 14-18- 31/07/1996	6.98; 5.00; 4.94; 6.00
HM94-18	5.70	74.50	core top	2469	158; 500; 512; 513	19/08/1998	6.98; 4.94; 4.00; 6.96
HM49-15	-0.36	66.34	core top	3260	1009; 110; 111; 126	22/06/2002; 25/06/2002	-1.00; 0.59; 1.84; -1.62
HM16142	2.60	63.25	core top	1100	217; 221; 223	12&13/10/2003;	1.79; 3.41; 3.92
HM52-39	-6.79	65.57	core top	2305	102; 104; 105; 106	20&21/06/2002	-7.98; -6.63; - 6.04; -5.28
23261-2	13.11	72.18	core top	2224	498	8/12/1998	10.92
23259-3	9.30	72.02	core top	2518	498	8/12/1998	10.92
HM16130	-2.42	65.10	core top	3182	108; 126; 127; 128	21/06/2002; 25/06/2002	-2.67; -1.62; - 2.36; -2.36; -3.06
HM52-42	-2.80	66.34	core top	3104	108; 109; 125	21-22-25/06/2002	-2.67; -1.00; - 0.92
HM71-22	-3.61	69.34	core top	1833	81; 83; 84; 190	16-17/06/2002; 01/10/2003	-0.60; -3.44; - 4.93; -6.77
HM57-16	-4.37	67.28	core top	2816	104; 105; 106; 108	6/21/2002	-6.63; -6.04; - 5.28; -2.67

[5]

e, Phosphate]	Hydrographic data Legrande [d18Osw]			Calc depth	T	S	d18Osw
Lat	Stations	Long	Lat	[m]	[C]	[psu]	[‰]
68.68; 69.59; 70.01	597; 598; 599	-12.5; -13.3; -14.5	69.5; 70.5	36.40	0.57	34.75	0.10
71.75; 73.00; 73.00	742; 744; 746	-7.5; -9.5; -11.5	71.5; 73.5	9.22	0.50	34.00	-0.04
77.95; 78.37; 78.81	1251; 1252; 1253	0.5; 1.5; 2.5	78.5; 79.5	20.64	1.63	34.99	-0.33
73.99; 75.00	963; 964; 965	-1.5; -2.5; -3.5	74.5; 75.5	22.45	-0.12	34.59	0.11
72.52	746; 748; 750	-3.5; -5.5; -7.5	71.5; 73.5	7.02	0.88	34.64	0.09
66.00	405; 406; 407	6.5; 7.5; 8.5	66.5; 67.5	62.62	8.92	35.16	0.37
62.83	25; 27; 29	-14.5; 16.5; -18.5	61.5; 62.5	931.27	4.79	35.65	0.44
66.00; 65.12; 63.90	255; 256; 258	-1.5; -0.5; 1.5	64.5; 65.5	66.66	6.74	35.07	0.39
66.00; 68.67	389; 390; 391	-7.5; -8.5; -9.5	66.5; 68.5	35.10	4.12	34.66	0.25
63.90; 63.19; 62.96	116; 117; 118	1.5; 2.5; 3.5	62.5; 63.5	298.48	4.87	35.05	0.47
74.50; 73.00; 74.01; 74.00	900; 901; 902	4.5; 5.5; 6.5	73.5; 74.5	33.90	4.13	34.78	0.30
74.50; 74.01; 75.00; 75.02	971; 972; 973	4.5; 5.5; 6.5	74.5; 75.5	47.44	3.03	34.85	0.28
66.00; 65.12	326; 327; 328	-1.5; -0.5; 0.5	65.5; 66.5	33.33	8.38	35.09	0.38
63.90; 63.19; 62.96	116; 117; 118	1.5; 2.5; 3.5	62.5; 63.5	63.90	9.73	35.19	0.40
66.00	320; 321; 322	-7.5; -6.5; -7.5	65.5; 66.5	22.21	5.99	34.83	0.32
73.11	764; 765; 767	10.5; 11.5; 13.5	71.5; 73.5	37.36	6.86	35.04	0.32
73.11	833; 834; 835	8.5; 9.5; 10.5	72.5; 73.5	34.43	7.20	35.03	0.34
66.01; 65.12; 64.72; 64.28	253; 254; 255	-3.5; -2.5; -1.5	64.5; 65.5	31.33	7.84	34.97	0.38
66.00; 66.01; 65.49	324; 325; 326	-3.5; -2.5; -1.5	65.5; 66.5	21.87	8.82	34.97	0.37
70.00; 69.24	606; 608; 610	-5.5; -3.5; -1.5	69.5; 70.5	43.38	6.94	35.09	0.29
66.00; 66.01	392; 393; 394	-6.5; -5.5; -4.5	66.5; 67.5	20.12	6.96	34.88	0.31

Mg/Ca	CO3	d18Oc VPDB
[mmol/ mol]	[mmol/ kgSW]	[‰]
1.00	124.35	3.4
0.98	127.24	3.5
0.90	138.67	3.03
0.92	140.16	3.67
0.80	142.95	3.23
0.82	160.95	2.01
0.82	160.95	3.09
0.80	153.05	2.58
0.80	155.38	3.19
0.85	139.48	2.92
0.87	157.61	3.32
0.91	151.05	3.47
0.91	171.46	2.23
0.93	175.33	1.83
0.97	162.92	2.73
0.97	167.87	2.55
1.01	171.20	2.5
0.99	167.83	2.4
1.04	178.50	2.38
1.04	168.53	2.44
1.07	167.61	2.59

A solution for constraining past marine Polar Amplification

Morley, A.^{1,2*}, de la Vega, E.¹, Raitzsch, M.³, Bijma, J.⁴, Ninnemann, U.⁵, Foster, G.L.⁶, Chalk, T.B.⁷,
Meilland, J.⁸, Cave, R.R.⁹, Büscher, J. V.^{9,10}, and Kucera, M.⁸.

¹ University of Galway, School of Geography, Archaeology and Irish Studies, H91TK33 Galway, Ireland.

² ICIRAG – Irish Centre for Research in Applied Geosciences, Belfield, Dublin 4, Ireland.

³ Dettmer Group GmbH & Co. KG., 28195 Bremen, Germany

⁴ Alfred-Wegener-Institut, Helmholtz-Zentrum für Polar- und Meeresforschung, Am Handelshafen 12, 27570, Bremerhaven,
Germany

⁵ University of Bergen, Department of Earth Science and Bjerknes Centre for Climate Research, 5007, Bergen, Norway

⁶ School of Ocean and Earth Science, University of Southampton, National Oceanography Centre Southampton, Southampton,

[revised manuscript text omitted]

between $\delta^{18}\text{O}_c$ and $[\text{CO}_3^{2-}]$ remains unchanged regardless of concentrations (Extended Data Figure
S3). In summary, while temperature contributes weakly to the variance in $\delta^{18}\text{O}_c$ above 5°C it does not
influence the interaction between $\delta^{18}\text{O}_c$ and $[\text{CO}_3^{2-}]$. Furthermore, these results confirm that the
$\delta^{18}\text{O}_c/[\text{CO}_3^{2-}]$ sensitivity does not rely on covariance with temperature, which varies over space and
time.

Following, we posit that the sensitivity or slope of $\delta^{18}\text{O}_c/[\text{CO}_3^{2-}]$ is regulated by the calcite saturation
product (K_{sp}), which in turn is determined by the conversion between (isotopically heavier) bicarbonate
and (isotopically lighter) carbonate ions in the calcification environment, that is a function of pH,
salinity, and to a lesser extent temperature [50]. We observe that the slopes of $\delta^{18}\text{O}_c/[\text{CO}_3^{2-}]$ for *N.*
*pachyderma* from tows and core tops are statistically the same at -0.0279 but offset by $\sim 2\%$ (Fig. 1d).
The slopes in our dataset are also steeper than reported for *Orbulina universa* (-0.002) or *Globigerina*
*bulloides* (e.g., -0.004) [51], but not outside of the slopes observed previously for other unicellular
planktonic calcifying organisms [50] (Fig. 1d). The steeper slopes for *N. pachyderma* suggest that the
relative contribution of the isotopically light $[\text{CO}_3^{2-}]$ to the calcifying fluid is low and that comparatively
less bicarbonate is converted into carbonate ion in the calcification environment when compared to
*O. universa* or *G. bulloides*. This hypothesis agrees well with the biomineralization model controlling
the exclusion/uptake of Mg described earlier in *N. pachyderma*, where low cell energy due to
unfavourable calcification environments and low metabolic/respiration rates at cold temperatures
leads to low $[\text{CO}_3^{2-}]$ available for calcification [45, 46, 57]. In addition, the release of CO_2 during
respiration, and the absence of photosynthetic activity from symbionts, further decrease the relative
contribution of isotopically light $[\text{CO}_3^{2-}]$ in favour of the isotopically heavier bicarbonate and
potentially further enhances the sensitivity of oxygen isotope fractionation to varying $[\text{CO}_3^{2-}]$ in low
$[\text{CO}_3^{2-}]$ settings. The observed 2% offset in the intercepts between tows and core tops agrees well
with the average 2.1% offset observed between ontogenetic calcite and a fully crusted *N. pachyderma*
[58] which have also been linked to slow (e.g., crust) and fast (e.g. ontogenetic calcite) calcification
rates. A caveat of this offset is that specimens of a similar degree of encrustation must be analysed
together to avoid the influence of varying degrees of encrustation on the $\delta^{18}\text{O}_c/[\text{CO}_3^{2-}]$ relationship.
Accordingly, the larger scatter observed for the core top versus plankton tow dataset could potentially
be explained by different degrees of crust formation [58].

To explore the application potential of the $\delta^{18}\text{O}_c/[\text{CO}_3^{2-}]$ proxy to fossil material, we compared $[\text{CO}_3^{2-}]$
derived from $\delta^{18}\text{O}_c$ with CARINA [59] derived $[\text{CO}_3^{2-}]$ values for the Kozdon core top dataset (see

Methods). Both methods provide $[\text{CO}_3^{2-}]$ values that are within a root mean square error (RMSE) of
$8.44 \mu\text{mol.kg}^{-1}$ of each other (adjusted $r^2=0.95$ $n=52$, $p<0.001$). Residuals are symmetrically distributed
around zero (Extended Data Figure S4). When $\delta^{18}\text{O}_c$ derived $[\text{CO}_3^{2-}]$ values are used to estimate the
260 Mg/Ca - $[\text{CO}_3^{2-}]$ sensitivity, the scatter is much lower (when compared to CARINA) resulting in more
robust correlation statistics (adjusted $r^2=0.81$ vs $r^2=0.58$) and lower uncertainties. The reduced scatter
is most likely due to the uncertainties linked to assigning modern water column hydrography from
CARINA to pre-industrial/late Holocene core tops. By using $\delta^{18}\text{O}_c$ to derive $[\text{CO}_3^{2-}]$ these uncertainties
are removed. To further test the validity of this approach we also derived $[\text{CO}_3^{2-}]$ from $\delta^{18}\text{O}_c$ values
published in the core top dataset (paired $\text{Mg}/\text{Ca} - \delta^{18}\text{O}_c$) published in Meland *et al.* [20] and find that
the Mg/Ca sensitivity to $[\text{CO}_3^{2-}]$ is statistically identical with the tows and the Kozdon datasets (Fig.
1b).

[revised manuscript text omitted]

**Figure 2 Paleo application** of Mg/Ca correction scheme to four glacial-interglacial periods including MIS1, MIS5, MIS9 and
 MIS11 [29, 30, 61]. Datasets originate from Eirik Drift, Southeast of Greenland from IODP site U1305, and sister site MD99-
 2227. Purple lines show three-point running averages of corrected Mg/Ca-based SSTs while grey lines show three-point
 running averages of uncorrected Mg/Ca-based SST reconstructions. The blue shading shows the 95% confidence interval
 associated with Mg/Ca_{corr} SST reconstructions. In pink are MAT-based SST reconstructions as shown in Irvani et al. [30] for
 MIS 5, 9, and 11. Note, that MAT SSTs are not available for the last glacial-interglacial transition.

A paleoceanographic map of corrected *N. pachyderma* Mg/Ca records covering the LGM and Late
 Holocene, demonstrates that the underestimated cooling (e.g., Δ SST) is most pronounced across mid-
 latitudes where our new estimates suggest cooling between $6-9 \pm 1.0^\circ\text{C}$ instead of milder $4-6^\circ\text{C}$ cooling
 (Fig. 3a). Our re-assessment suggests that polar amplification or high-latitude LGM cooling was of a
 greater magnitude and more extensive spatial expression than previously estimated, which raises
 questions about our understanding of heat transport under glacial boundary conditions. Furthermore,
 absolute Holocene and LGM SST values based on corrected *N. pachyderma* Mg/Ca records (Fig. 3b and
 3c) are much closer to model-based reconstructions of LGM SSTs than previous estimates based on
 this species (Extended Data Fig. S5). Considering these examples, we believe that a systematic
 application of the calibration approach as presented in our study provides, for the first time, an

opportunity to directly estimate past temperatures in Polar Oceans, where low diversity of planktonic
foraminifera prevented the use of transfer functions and where dinocysts are rare.

336 **Figure 3 Data model intercomparison.** The top panel shows the difference in cooling between the Holocene and the LGM

[revised manuscript text omitted]

normalised Mg/Ca ($\text{Mg}/\text{Ca}_{[\text{meas}]}/\text{Mg}/\text{Ca}_{[\text{pred}]}$) to obtain a $\text{Mg}/\text{Ca}_{[\text{corr}]}$ for non-thermal effects alongside
its uncertainties. Based on the previous relationships (Fig. 1a and b), $\text{Mg}/\text{Ca}_{[\text{corr}]}$ has an uncertainty of
± 0.2 (2σ) which we then calibrate against the in-situ temperatures obtained from CTDs and/or the
CARINA database.

These errors are related to the depth estimation of the foraminifera from the isotope data but
typically range from 0 – 3°C (minimum 0°C, maximum 10°C, average $\pm 2.8^\circ\text{C}$ with a standard deviation
of 0.9°C). Note, for the data from Meland et al. [20] we are unable to calculate an uncertainty so we
use the average of the uncertainties from Kozdon et al. [27] which is 1.6°C at 2σ). For the *N. incompta*
data, all the steps to calculate the uncertainties are identical to the *N. pachyderma* dataset except, as
there are no $\delta^{18}\text{O}$ data, we assigned a temperature uncertainty of $\pm 0.8^\circ\text{C}$ (2σ) [39]. The $\text{Mg}/\text{Ca}_{[\text{corr}]}$
uncertainty is then assumed to be the same as for *N. pachyderma*. Both species are combined to make
the final calibration but using one or the other only gives consistent relationships within error,
strengthening the argument against a species-specific vital effect.

The calibration Monte Carlo simulation finally is conducted on 1000 iterations of a slope using the
propagated $\text{Mg}/\text{Ca}_{[\text{corr}]}$ and the individual temperature uncertainties (as above). We take the median
slope of this curve to define the $\text{Mg}/\text{Ca}_{[\text{corr}]}$ to temperature relationship, and the 68 and 95 %
confidence intervals (1 and 2σ respectively) to describe its uncertainties. These uncertainties
represent the uncertainty in the defined relationship, assuming it to be consistent and are associated
with a temperature reconstruction uncertainty of $\pm 1.2^\circ\text{C}$ at temperatures of around 0°C and $\pm 0.9^\circ\text{C}$ at
temperatures of around 15°C . These confidence intervals give a conservative uncertainty on the
relationship between $\text{Mg}/\text{Ca}_{[\text{corr}]}$ and temperature and are used for any relative change calculations
presented in this study. We also plot a prediction uncertainty, which can be considered the uncertainty
from a single future or additional data point but is overly conservative for the whole population of
data.

**Competing interests.** The authors declare no competing interests.

**Data Availability.** All new data and recalculated datasets shown here are available in the extended
data tables and Source Data tables provided with this paper.

**Code Availability.** The fully commented code for the error propagation calculation and input files are
available as an extended datafile provided with this paper.

**Materials & Correspondence.** Correspondence and requests for materials should be addressed to AM

**Acknowledgements.** This research was funded by MSCA-IF Project ARCTICO [838529], the Marine
Institute of Ireland Research Programme 2014-2020 (PDOC/19/05/02), and Science Foundation
Ireland Frontiers for the Future Project 21/FFP-P/10261, awarded to AM. In addition, AM
acknowledges Grant in Aid funding from the Marine Institute for research expedition CE20009 on the
RV Celtic Explorer with special thanks to the crew sailing under Master Anthony English and Research
Assistant Aedin McAlear for the analysis of Ocean ALK and DIC samples from the CE20009 survey. MR
acknowledges DFG (German Research Foundation) for funding through research grant no. RA 2068/4-
1. RC acknowledges funding for Project CE2COAST, funded by ANR (FR), BELSPO (BE), FCT (PT), IZM
(LV), MI (IE), MIUR (IT), Rannis (IS) and RCN (NO) through the 2019 "Joint Transnational Call on Next
Generation Climate Science in Europe for Oceans" initiated by JPI Climate and JPI Oceans

**Author contributions.** AM designed the project carried out palaeoceanographic sample preparation
of plankton tows from the Labrador Sea, compiled and analysed all datasets, wrote the manuscript,
and coordinated the input of all co-authors. EDLV carried out palaeoceanographic sample preparation
and geochemical analysis of plankton tows and core tops from CE20009 and contributed to the writing
of the final manuscript. MR performed trace element and boron isotope analysis on *N. pachyderma*
plankton tow samples from the Labrador Sea. JB supervised AM at the Alfred Wegner Institute and
oversaw trace element and boron isotope analysis. UN performed all stable isotope analyses of water
and foraminifera samples from CE20009 and contributed to the final version of this manuscript. GF
oversaw trace element analysis on *N. pachyderma* plankton tow samples from CE20009 and
contributed to the final version of this manuscript. TC performed uncertainty calculations for the
571 Mg/Ca correction and contributed to the final version of this manuscript. JM collected all plankton
tows during CE20009 and contributed to the final version of this manuscript. RC oversaw Alkalinity
and DIC analysis on seawater samples collected during CE20009 and contributed to the final version
of this manuscript. JB performed Alkalinity and DIC analysis on seawater samples collected during
CE20009 and contributed to the final version of this manuscript. MK advised AM during project design,
supervised her during her MSCA-IF at MARUM, University of Bremen, and contributed to the final
version of this manuscript.

**References:**

- 1. Holland, M.M. and C.M. Bitz, *Polar amplification of climate change in coupled models*. Climate
dynamics, 2003. **21**(3-4): p. 221-232.
- 2. Manabe, S. and R.J. Stouffer, *Sensitivity of a global climate model to an increase of CO2*
*concentration in the atmosphere*. Journal of Geophysical Research: Oceans, 1980. **85**(C10): p.
5529-5554.
- 3. Årthun, M., et al., *Skillful prediction of northern climate provided by the ocean*. Nature
communications, 2017. **8**(1): p. 1-11.
- 4. Årthun, M., T. Eldevik, and L.H. Smedsrud, *The role of Atlantic heat transport in future Arctic*
*winter sea ice loss*. Journal of Climate, 2019. **32**(11): p. 3327-3341.
- 5. Parmentier, F.-J.W., et al., *The impact of lower sea-ice extent on Arctic greenhouse-gas*
*exchange*. Nature climate change, 2013. **3**(3): p. 195-202.
- 6. Jansen, E., et al., *Past perspectives on the present era of abrupt Arctic climate change*. Nature
Climate Change, 2020. **10**(8): p. 714-721.
- 7. Vavrus, S.J., *The influence of Arctic amplification on mid-latitude weather and climate*. Current
Climate Change Reports, 2018. **4**(3): p. 238-249.
- 8. Boetius, A., et al., *Microbial ecology of the cryosphere: sea ice and glacial habitats*. Nature
Reviews Microbiology, 2015. **13**(11): p. 677-690.
- 9. Huang, J., et al., *Recently amplified arctic warming has contributed to a continual global*
*warming trend*. Nature climate change, 2017. **7**(12): p. 875-879.
- 10. Meredith, M., et al., *Polar regions. chapter 3, ipcc special report on the ocean and cryosphere*
*in a changing climate*. 2019.
- 11. Cowtan, K. and R.G. Way, *Coverage bias in the HadCRUT4 temperature series and its impact*
*on recent temperature trends*. Quarterly Journal of the Royal Meteorological Society, 2014.
**140**(683): p. 1935-1944.
- 12. Burls, N. and N. Sagoo, *Increasingly Sophisticated Climate Models Need the Out-Of-Sample*
*Tests Paleoclimates Provide*. Journal of Advances in Modeling Earth Systems, 2022. **14**(12): p.
e2022MS003389.
- 13. Fedorov, A.V., et al., *Tightly linked zonal and meridional sea surface temperature gradients over*
*the past five million years*. Nature Geoscience, 2015. **8**(12): p. 975-980.
- 14. Polyakov, I.V., et al., *Greater role for Atlantic inflows on sea-ice loss in the Eurasian Basin of the*
*Arctic Ocean*. Science, 2017. **356**(6335): p. 285-291.
- 15. Kucera, M., et al., *Reconstruction of sea-surface temperatures from assemblages of planktonic*
*foraminifera: multi-technique approach based on geographically constrained calibration data*
*sets and its application to glacial Atlantic and Pacific Oceans*. Quaternary Science Reviews,
2005. **24**(7): p. 951-998.
- 16. Weinelt, M., et al., *Paleoceanographic proxies in the northern North Atlantic*. The northern
North Atlantic: a changing environment, 2001: p. 319-352.
- 17. Telford, R.J., *Limitations of dinoflagellate cyst transfer functions*. Quaternary Science Reviews,
2006. **25**(13-14): p. 1375-1382.
- 18. Bendle, J., A. Rosell-Melé, and P. Ziveri, *Variability of unusual distributions of alkenones in the*
*surface waters of the Nordic seas*. Paleoceanography, 2005. **20**(2).
- 19. de Vernal, A. and C. Hillaire-Marcel, *Natural Variability of Greenland Climate, Vegetation, and*
*Ice Volume During the Past Million Years*. Science, 2008. **320**(5883): p. 1622-1625.
- 20. Meland, M.Y., et al., *Mg/Ca ratios in the planktonic foraminifer Neogloboquadrina pachyderma*
*(sinistral) in the northern North Atlantic/Nordic Seas*. Geochemistry, Geophysics, Geosystems,
2006. **7**(6).
- 21. Katz, M.E., et al., *Traditional and emerging geochemical proxies in foraminifera* Journal of
Foraminiferal Research, 2010. **40**(2): p. 165-192.
- 22. Katz, A., *The interaction of magnesium with calcite during crystal growth at 25–90 C and one*
*atmosphere*. Geochimica et Cosmochimica Acta, 1973. **37**(6): p. 1563IN31579-15781586.

- 23. Nuernberg, D., *Magnesium in tests of Neogloboquadrina pachyderma sinistral from high*
*northern and southern latitudes*. The Journal of Foraminiferal Research, 1995. **25**(4): p. 350-
368.
- 24. Hendry, K.R., et al., *Controls on stable isotope and trace metal uptake in Neogloboquadrina*
*pachyderma (sinistral) from an Antarctic sea-ice environment*. Earth and Planetary Science
Letters, 2009. **278**(1): p. 67-77.
- 25. Davis, C.V., et al., *Relationships Between Temperature, pH, and Crusting on Mg/Ca Ratios in*
*Laboratory-Grown Neogloboquadrina Foraminifera*. Paleoceanography, 2017. **32**(11): p. 1137-
1152.
- 26. Livsey, C.M., et al., *High-resolution Mg/Ca and $\delta^{18}O$ patterns in modern Neogloboquadrina*
*pachyderma from the Fram Strait and Irminger Sea*. Paleoceanography and Paleoclimatology,
2020. **35**(9): p. e2020PA003969.
- 27. Kozdon, R., et al., *Reassessing Mg/Ca temperature calibrations of Neogloboquadrina*
*pachyderma (sinistral) using paired $\delta^{44}Ca/40Ca$ and Mg/Ca measurements*. Geochemistry,
Geophysics, Geosystems, 2009. **10**(3).
- 28. Tierney, J.E., et al., *Glacial cooling and climate sensitivity revisited*. Nature, 2020. **584**(7822):
p. 569-573.
- 29. Irvani, N., et al., *Evidence for regional cooling, frontal advances, and East Greenland Ice Sheet*
*changes during the demise of the last interglacial*. Quaternary Science Reviews, 2016. **150**: p.
184-199.
- 30. Irvani, N., et al., *A low climate threshold for south Greenland Ice Sheet demise during the Late*
*Pleistocene*. Proceedings of the National Academy of Sciences, 2020. **117**(1): p. 190-195.
- 31. Emiliani, C., *Depth habitats of some species of pelagic foraminifera as indicated by oxygen*
*isotope ratios*. American Journal of Science, 1954. **252**(3): p. 149-158.
- 32. Tell, F., et al., *Upper-ocean flux of biogenic calcite produced by the Arctic planktonic*
*foraminifera Neogloboquadrina pachyderma*. Biogeosciences, 2022. **19**(20): p. 4903-4927.
- 33. Manno, C. and A. Pavlov, *Living planktonic foraminifera in the Fram Strait (Arctic): absence of*
*diel vertical migration during the midnight sun*. Hydrobiologia, 2014. **721**: p. 285-295.
- 34. Jonkers, L., et al., *Chamber formation leads to Mg/Ca banding in the planktonic foraminifer*
*Neogloboquadrina pachyderma*. Earth and Planetary Science Letters, 2016. **451**: p. 177-184.
- 35. Evans, D., et al., *Revisiting carbonate chemistry controls on planktic foraminifera Mg/Ca:*
*implications for sea surface temperature and hydrology shifts over the Paleocene–Eocene*
*Thermal Maximum and Eocene–Oligocene transition*. Climate of the Past, 2016. **12**(4): p. 819-
835.
- 36. Kisakürek, B., et al., *Controls on shell Mg/Ca and Sr/Ca in cultured planktonic foraminiferan,*
*Globigerinoides ruber (white)*. Earth and Planetary Science Letters, 2008. **273**(3-4): p. 260-269.
- 37. Russell, A.D., et al., *Effects of seawater carbonate ion concentration and temperature on shell*
*U, Mg, and Sr in cultured planktonic foraminifera*. Geochimica et Cosmochimica Acta, 2004.
**68**(21): p. 4347-4361.
- 38. Gray, W.R., et al., *The effects of temperature, salinity, and the carbonate system on Mg/Ca in*
*Globigerinoides ruber (white): A global sediment trap calibration*. Earth and Planetary Science
Letters, 2018. **482**: p. 607-620.
- 39. Morley, A., et al., *Environmental Controls on Mg/Ca in Neogloboquadrina incompta: A Core-*
*Top Study From the Subpolar North Atlantic*. Geochemistry, Geophysics, Geosystems, 2017.
- 40. Gray, W.R. and D. Evans, *Nonthermal influences on Mg/Ca in planktonic foraminifera: a review*
*of culture studies and application to the Last Glacial Maximum*. Paleoceanography and
Paleoclimatology, 2019. **34**(3): p. 306-315.
- 41. Tierney, J.E., et al., *Bayesian calibration of the Mg/Ca paleothermometer in planktic*
*foraminifera*. Paleoceanography and Paleoclimatology, 2019. **34**(12): p. 2005-2030.
- 42. Foster, G.L. and J.W. Rae, *Reconstructing ocean pH with boron isotopes in foraminifera*. Annual
Review of Earth and Planetary Sciences, 2016. **44**: p. 207-237.

- 43. Darling, K.F., et al., *A resolution for the coiling direction paradox in Neogloboquadrina*
*pachyderma*. *Paleoceanography*, 2006. **21**(2).
- 44. Schiebel, R. and C. Hemleben, *Planktic foraminifers in the modern ocean*. 2017.
- 45. Evans, D., W. Müller, and J. Erez, *Assessing foraminifera biomineralisation models through*
*trace element data of cultures under variable seawater chemistry*. *Geochimica et*
*Cosmochimica Acta*, 2018. **236**: p. 198-217.
- 46. Zeebe, R.E. and A. Sanyal, *Comparison of two potential strategies of planktonic foraminifera*
*for house building: Mg²⁺ or H⁺ removal?* *Geochimica et Cosmochimica Acta*, 2002. **66**(7): p.
1159-1169.
- 47. Jonkers, L., et al., *Encrustation and trace element composition of Neogloboquadrina dutertrei*
*assessed from single chamber analyses—implications for paleotemperature estimates*.
*Biogeosciences*, 2012. **9**(11): p. 4851-4860.
- 48. Hupp, B.N. and J.S. Fehrenbacher, *Geochemical differences between alive, uncrusted and dead,*
*crusted shells of Neogloboquadrina pachyderma: Implications for paleoreconstruction*.
*Paleoceanography and Paleoclimatology*, 2023. **38**(10): p. e2023PA004638.
- 49. Vázquez Riveiros, N., et al., *Mg/Ca thermometry in planktic foraminifera: Improving*
*paleotemperature estimations for G. bulloides and N. pachyderma left*. *Geochemistry,*
*Geophysics, Geosystems*, 2016. **17**(4): p. 1249-1264.
- 50. Ziveri, P., et al., *A universal carbonate ion effect on stable oxygen isotope ratios in unicellular*
*planktonic calcifying organisms*. *Biogeosciences*, 2012. **9**(3): p. 1025-1032.
- 51. Bijma, J., H. Spero, and D. Lea, *Reassessing foraminiferal stable isotope geochemistry: Impact*
*of the oceanic carbonate system (experimental results)*, in *Use of proxies in paleoceanography*.
1999, Springer. p. 489-512.
- 52. Ravelo, A.C. and C. Hillaire-Marcel, *Chapter eighteen the use of oxygen and carbon isotopes of*
*foraminifera in paleoceanography*. *Developments in marine geology*, 2007. **1**: p. 735-764.
- 53. Emiliani, C., *Pleistocene temperatures*. 1955.
- 54. Epstein, S., et al., *Carbonate-water isotopic temperature scale*. *Geological Society of America*
*Bulletin*, 1951. **62**(4): p. 417-426.
- 55. Spero, H.J., et al., *Effect of seawater carbonate concentration on foraminiferal carbon and*
*oxygen isotopes*. *Nature*, 1997. **390**(497-500).
- 56. Bauch, D., J. Carstens, and G. Wefer, *Oxygen isotope composition of living Neogloboquadrina*
*pachyderma (sin.) in the Arctic Ocean*. *Earth and Planetary Science Letters*, 1997. **146**(1-2): p.
47-58.
- 57. Lombard, F., et al., *Modelling the temperature dependent growth rates of planktic*
*foraminifera*. *Marine Micropaleontology*, 2009. **70**(1): p. 1-7.
- 58. Kozdon, R., et al., *Intratrust oxygen isotope variability in the planktonic foraminifer N.*
*pachyderma: Real vs. apparent vital effects by ion microprobe*. *Chemical Geology*, 2009. **258**(3-
4): p. 327-337.
- 59. Key, R., et al., *The CARINA data synthesis project: introduction and overview*. *Earth System*
*Science Data*, 2010. **2**(1): p. 105-121.
- 60. Darling, K.F., et al., *Molecular evidence links cryptic diversification in polar planktonic protists*
*to Quaternary climate dynamics*. *Proceedings of the National Academy of Sciences*, 2004.
**101**(20): p. 7657-7662.
- 61. Thornalley, D.J., I.N. McCave, and H. Elderfield, *Freshwater input and abrupt deglacial climate*
*change in the North Atlantic*. *Paleoceanography*, 2010. **25**(1).
- 62. Pflaumann, U., et al., *Glacial North Atlantic: Sea-surface conditions reconstructed by GLAMAP*
*2000*. *Paleoceanography*, 2003. **18**(3): p. 1065.
- 63. De Vernal, A., et al., *Comparing proxies for the reconstruction of LGM sea-surface conditions*
*in the northern North Atlantic*. *Quaternary Science Reviews*, 2006. **25**(21-22): p. 2820-2834.
- 64. Schlitzer, R., *Ocean Data View*. 2002, <http://ww.awi-bremenhaven.de/GEO/ODV,2002>.

- 65. Masson-Delmotte, V., et al., *Past and future polar amplification of climate change: climate*
*model intercomparisons and ice-core constraints*. Climate Dynamics, 2006. **26**: p. 513-529.
- 66. Zhu, J., et al., *LGM paleoclimate constraints inform cloud parameterizations and equilibrium*
*climate sensitivity in CESM2*. Journal of Advances in Modeling Earth Systems, 2022. **14**(4): p.
e2021MS002776.
- 67. Dutton, A., et al., *Sea-level rise due to polar ice-sheet mass loss during past warm periods*.
science, 2015. **349**(6244): p. aaa4019.
- 68. Menviel, L.C., et al., *An ice–climate oscillatory framework for Dansgaard–Oeschger cycles*.
Nature Reviews Earth & Environment, 2020. **1**(12): p. 677-693.
- 69. Kageyama, M., et al., *Modeling the climate of the Last Glacial Maximum from PMIP1 to PMIP4*.
Past Global Changes Magazine, 2021. **29**(2): p. 80-81.
- 70. Mintrop, L., et al., *Alkalinity determination by potentiometry: Intercalibration using three*
*different methods*. 2000.
- 71. Dickson, A., C. Sabine, and J. Christian, *Guide to best practices for ocean CO₂ measurements*,
*PICES Special Publication 3*. 2007, IOCCP report.
- 72. LeGrande, A.N. and G.A. Schmidt, *Global gridded data set of the oxygen isotopic composition*
*in seawater*. Geophysical Research Letters, 2006. **33**(12): p. L12604.
- 73. O'Neil, J.R., R.N. Clayton, and T.K. Mayeda, *Oxygen Isotope Fractionation in Divalent Metal*
*Carbonates*. The Journal of Chemical Physics, 1969. **51**(12): p. 5547-5558.
- 74. Friedman, I. and J.R. O'Neil, *Data of geochemistry: Compilation of stable isotope fractionation*
*factors of geochemical interest*. Vol. 440. 1977: US Government Printing Office.
- 75. Stangeew, E., *Distribution and Isotopic Composition of Living Planktonic Foraminifera N.*
*pachyderma (sinistral) and T. quinqueloba in the High Latitude North Atlantic*. 2001, Christian-
Albrechts Universität Kiel.
- 76. Pados, T., et al., *Oxygen and carbon isotope composition of modern planktic foraminifera and*
*near-surface waters in the Fram Strait (Arctic Ocean)—a case study*. Biogeosciences, 2015.
**12**(6): p. 1733-1752.
- 77. Volkmann, R. and M. Mensch, *Stable isotope composition ($\delta^{18}O$, $\delta^{13}C$) of living planktic*
*foraminifers in the outer Laptev Sea and the Fram Strait*. Marine Micropaleontology, 2001.
**42**(3-4): p. 163-188.
- 78. Mikis, A., et al., *Temporal variability in foraminiferal morphology and geochemistry at the West*
*Antarctic Peninsula: a sediment trap study*. Biogeosciences, 2019. **16**(16): p. 3267-3282.
- 79. Jonkers, L., et al., *Seasonal patterns of shell flux, $\delta^{18}O$ and $\delta^{13}C$ of small and large N.*
*pachyderma (s) and G. bulloides in the subpolar North Atlantic*. Paleoceanography, 2013. **28**(1):
p. 164-174.
- 80. Jonkers, L., et al., *Seasonal stratification, shell flux, and oxygen isotope dynamics of left-coiling*
*N. pachyderma and T. quinqueloba in the western subpolar North Atlantic*. Paleoceanography,
2010. **25**(2).
- 81. Jonkers, L., et al., *Variability in Neogloboquadrina pachyderma stable isotope ratios from*
*isothermal conditions: implications for individual foraminifera analysis*. Climate of the Past,
2022. **18**(1): p. 89-101.
- 82. Kohfeld, K.E., et al., *Neogloboquadrina pachyderma (sinistral coiling) as paleoceanographic*
*tracers in polar oceans: Evidence from Northeast Water Polynya plankton tows, sediment*
*traps, and surface sediments*. Paleoceanography, 1996. **11**(6): p. 679-699.
- 83. Simstich, J., M. Sarnthein, and H. Erlenkeuser, *Paired $\delta^{18}O$ signals of Neogloboquadrina*
*pachyderma (s) and Turborotalita quinqueloba show thermal stratification structure in Nordic*
*Seas*. Marine Micropaleontology, 2003. **48**(1): p. 107-125.
- 84. King, A.L. and W.R. Howard, *$\delta^{18}O$ seasonality of planktonic foraminifera from Southern Ocean*
*sediment traps: Latitudinal gradients and implications for paleoclimate reconstructions*.
Marine Micropaleontology, 2005. **56**(1-2): p. 1-24.

- 85. Greco, M., et al., *Depth habitat of the planktonic foraminifera < i> Neogloboquadrina*
*pachyderma in the northern high latitudes explained by sea-ice and chlorophyll*
*concentrations*. Biogeosciences, 2019. **16**(17): p. 3425-3437.
- 86. Barker, S., M. Greaves, and H. Elderfield, *A study of cleaning procedures used for foraminiferal*
*Mg/Ca paleothermometry*. Geochemistry Geophysics Geosystems, 2003. **4**: p. 8407.
- 87. Henehan, M.J., et al., *Evaluating the utility of B/C a ratios in planktic foraminifera as a proxy*
*for the carbonate system: A case study of G lobigerinoides ruber*. Geochemistry, Geophysics,
Geosystems, 2015. **16**(4): p. 1052-1069.
- 88. Spratt, R.M. and L.E. Lisiecki, *A Late Pleistocene sea level stack*. Climate of the Past, 2016. **12**(4):
p. 1079-1092.
- 89. Fairbanks, R.G., *A 17,000 year glacio-eustatic sea level record: influence of glacial melting rate*
*on the Younger Dryas event and deep-ocean circulation*. Nature, 1989. **342**: p. 637-642.
- 90. Schrag, D.P., G. Hampt, and D.W. Murray, *Pore fluid constraints on the temperature and oxygen*
*isotopic composition of the glacial ocean*. Science, 1996. **272**: p. 1930-1932.
- 91. Adkins, J.F., K. McIntyre, and D.P. Schrag, *The salinity, temperature, and $\delta^{18}O$ of the glacial*
*deep ocean*. Science, 2002. **298**(5599): p. 1769-1773.
- 92. Rohling, E., et al., *Sea-level and deep-sea-temperature variability over the past 5.3 million*
*years*. Nature, 2014. **508**(7497): p. 477-482.
- 93. Ezat, M.M., T.L. Rasmussen, and J. Groeneveld, *Reconstruction of hydrographic changes in the*
*southern Norwegian Sea during the past 135 kyr and the impact of different foraminiferal*
*Mg/Ca cleaning protocols*. Geochemistry, Geophysics, Geosystems, 2016. **17**(8): p. 3420-3436.

**Extended Data Figures**

**Extended Data Figure S1 Station Map.** Inverted triangles (red) show samples collected from plankton
tows (e.g., MSM cruises in the Baffin Bay and Labrador Sea and CE cruises in the Nordic Sea), while
black triangles show locations of the Kozdon dataset. In the Nordic Seas plankton tows are also paired
with corresponding core tops. The temperature data shown is from the June-September WOA 2018
dataset at 30m. The map was generated using Ocean Data View [64]

**Extended Data Figure S2** Regression Tree. This plot describes the interaction of $[\text{CO}_3^{2-}]$ and
 temperature on $\delta^{18}\text{O}_c$ in the generalized model.

**Extended Data Figure S3.** The left panel shows the relationship between Temperature and $\delta^{18}\text{O}_c$, while
 the right panel shows the relationship between Temperature and $[\text{CO}_3^{2-}]$ for the combined tow and
 core top *N. pachyderma* datasets. The plankton tow dataset on *N. pachyderma* (left) and associated
 hydrographic data (derived using GIODAP (right) from Bauch et al. [56] is shown in green. The yellow
 shading highlights hydrographic conditions below 4°C.

**Extended Data Figure S4** Residual Plots. (Top) shows the regression and residuals for the $\delta^{18}\text{O}_c$ and
 $[\text{CO}_3^{2-}]$ relationship. (Bottom) shows the regression and residuals for the relationship describing
 $[\text{CO}_3^{2-}]$ derived from CARINA and $[\text{CO}_3^{2-}]$ derived from $\delta^{18}\text{O}_c$.

**Extended Data Figure S5** Uncorrected LGM (top) and Late Holocene (bottom) SSTs based on *N.*
 *pachyderma* Mg/Ca calibrated using the calibration equation developed by Kozdon et al. 2009 [27].
 For datasets used please see Extended Data Table 7.